# Novel insights into deep groundwater exploration by geophysical estimation of hard rock permeability

**Muhammad Hasan** [1, 2, 3,*], **Lijun Su** [1, 2, 3, **]

[1] *State Key Laboratory of Mountain Hazards and Engineering Resilience, Institute of Mountain Hazards and Environment, Chinese Academy of Sciences, Chengdu 610299, China*

[2] *China-Pakistan Joint Research Center on Earth Sciences, CAS-HEC, Islamabad, Pakistan*

[3] *University of Chinese Academy of Sciences, Beijing 100049, China*

*Corresponding authors:

Muhammad Hasan: Email: mhasan@imde.ac.cn; ORCID: https://orcid.org/0000-0001-6804-7962; Phone Number: +86-13051361710

Lijun Su: Email: sulijun1976@163.com; ORCID: https://orcid.org/0000-0001-9972-4698

Corresponding authors' postal address: State Key Laboratory of Mountain Hazards and Engineering Resilience, Institute of Mountain Hazards and Environment, Chinese Academy of Sciences, Chengdu 610299, China

**Abstract**

Deep groundwater exploration in hard rock is a global challenge. An accurate measurement of hydraulic parameters is essential for both effective groundwater management and the prediction of future scenarios. The permeability (k) of an aquifer is typically measured in groundwater studies. Boreholes are the traditional means of measuring k. However, conventional approaches have a lot of flaws, such as being intrusive, expensive, time-consuming, useful only for areas with relatively uniform topographies, and only providing point-scale k measurements. Moreover, traditional approaches may not be able to do deep groundwater assessments. In contrast, geophysical technologies may assess subsurface hydrogeological conditions across large areas with minimal disruption to existing structures, in a shorter amount of time, and at a reduced cost. Several geophysical investigations previously used empirical methods to estimate the k parameter. These studies, however, used the VES (vertical electrical sounding) method to estimate k in a homogeneous setting at shallow depths, and only in 1D. It is difficult to quantify the aquifer potential in hard rock terrains using borehole or VES-based k due to the intrinsic heterogeneity of the terrain. For the first time, this work uses CSAMT (controlled-source audio-frequency magnetotellurics) method to estimate 2D and 3D k over 1 km depth in the exceedingly diverse environments of different rocks. These findings enable the scientific planning and management of deep groundwater resources in highly varied hard rock terrains where hydrogeological data is unavailable, resulting in a more accurate hydrogeological model compared to prior studies. This, in turn, decreases the necessity for expensive pumping tests and enables a more comprehensive evaluation of aquifer potential.

**Keywords:** Permeability (k); Geophysical methods; Hydraulic parameters; Groundwater; Hard rock; Hydrogeological uncertainty

## 1 Introduction

Metamorphic and igneous rocks make up the bulk of Earth's crust, which accounts for around a third of the planet's surface (Amiotte Suchet et al., 2003). The main focus of groundwater study in hard rock is the examination of underlying geological layers, faults, and fractures (Fernando and Pacheco, 2015; Hasan et al., 2021). Groundwater evaluation and monitoring relies heavily on categorizing rock mass according to its aquifer yield, or its capacity to store water (Majumdar and Das, 2011; Nwosu et al., 2013; Qian et al., 2024). The water-bearing rock's aquifer potential is dependent on a wide range of environmental variables. Several factors influence this, such as the rock type, its relationship and deformation, faults, mineral content, water penetration, rock-rock joints, and the rate of weathering (Dell'Oca et al., 2020; Abbas et al., 2022). Groundwater evaluations have a major challenge in determining the vertical and horizontal water-holding capacity of underlying rock over expansive areas (Courtois et al., 2010; Dewandel et al., 2004). Prior to beginning groundwater extraction, it is essential to accurately and thoroughly evaluate the aquifer potential linked to the different types of rock. Structural variability and a lack of data make it difficult to evaluate the water-carrying capabilities of geological layers (Robinson et al., 2016; Worthington et al., 2016; Zhu et al., 2017). Several groundwater and environmental problems may arise from a lack of understanding of hydrogeological uncertainty (Dewandel et al., 2004; Refsgaard et al., 2012; Lachassagne et al., 2021). Difficult issues in groundwater research include evaluating the condition of geological layers for continual groundwater assessments and minimizing costs without compromising effectiveness.

Rodell et al. (2009), Wada et al. (2010), Laghari et al. (2012), Wada et al. (2014), and Jasechko et al. (2024) all attest to the fact that groundwater resources around the globe are dwindling at an alarming rate. In order to effectively manage and utilize these precious assets, it

is crucial to do a thorough and precise assessment of groundwater resources. Consideration of hydraulic properties is crucial in groundwater evaluations. Permeability is the most popular aquifer measure and is mainly used to assess the water-holding capacity of rocks all over the world (Dewandel et al., 2004; Gerke et al., 2011; Allègre et al., 2016; Fiandaca et al., 2018; Mudunuru et al., 2022; Esmaeilpour et al., 2023; Yan et al., 2024; Carbillet et al., 2024). The aquifer potential of geological layers is usually determined by permeability (Zhang et al., 2004; Pellet et al., 2024). De Lima and Niwas (2000), Soupios et al. (2007), Hasan et al. (2021), and Yang and Zhang (2024) all state that borehole testing is the standard method for measuring aquifer parameters. While boreholes do enhance geological data, producing a comprehensive 2D analysis is a time-consuming and problematic process (Hubbard and Rubin, 2002; Niwas and De Lima, 2003). Borehole methods have a number of drawbacks, including being expensive and time-consuming, requiring large apparatus or machinery, being difficult to implement in hilly terrain, only providing localized information, not being able to image lateral geological structures, and not being able to evaluate the deep subsurface structures (Singh, 2005; Lin et al., 2018; Asfahani, 2023). Uncertainty in the estimation of groundwater resources may result from a lack of borehole data, as these limitations make it challenging to regularly execute a large number of drilling trials. Alternatively, there needs to be a way to reduce the number of expensive drilling while still precisely evaluating the groundwater storage capacity of the potential rock masses.

A number of prior groundwater investigations have made use of geophysical techniques (Bentley and Gharibi, 2004; Yadav and Singh, 2007; Fu et al., 2013; Vouillamoz et al., 2014; Robinson et al., 2016; Lin et al., 2018; Kouadio et al., 2020; Abbas et al., 2022; Kouadio et al., 2023; Zhang et al., 2024). A number of studies have shown that geophysical procedures

outperform drilling techniques in terms of speed, ease of use, cost, and lack of invasiveness (Hu et al., 2013; Lin et al., 2018; Di et al., 2020; Fusheng et al., 2022; Hasan et al., 2024). Additionally, they are capable of conducting thorough geological evaluations in both the vertical and horizontal planes (Fu et al., 2013; Hasan et al., 2021). These methods are superior to others when it comes to collecting hydrogeological data from various subterranean habitats (Niwas and De Lima, 2003; Wynn et al., 2016; Kouadio et al., 2023). Groundwater studies nowadays often include resistivity surveys. Resistivity methods offer a broader resistivity range compared to other geophysical parameters, which is a major advantage (Bentley and Gharibi, 2004; Camporese et al., 2011; Robinson et al., 2016). The three main methods for measuring resistivity are the controlled source audio-frequency magnetotellurics (CSAMT), vertical electoral soundings (VES), and electrical resistivity tomography (ERT) (Soupios et al., 2007; Di et al., 2020; Zhang et al., 2024). Niwas and De Lima (2003), Soupios et al. (2007), Majumdar and Das (2011), Nwosu et al. (2013), Hasan et al. (2021), and Asfahani (2023) are among the previous groundwater-based geophysical studies that primarily utilized the VES method to evaluate groundwater resources in a single dimension. It is unusual to evaluate aquifer yield at great depths in hard rock terrains using two- and three-dimensional hydraulic properties. Recent studies have demonstrated that CSAMT, which aims to gather extensive subsurface data at very deep depths using 2D/3D evaluations, is the most cost-effective and appropriate geophysical method for researching hard rock (Smith and Booker, 1991; Simpson and Bahr, 2005; Bai et al., 2010; Fu et al., 2013; Hu et al., 2013; Wang et al., 2015; Wynn et al., 2016; Di et al., 2020; Zhang et al., 2021; Kouadio et al., 2023; Hasan et al., 2024). Advantages of CSAMT over other geophysical research methods include its lower cost, its responsiveness to low-resistance rocks, and its ease of usage in challenging topographic circumstances (An et al., 2016; Kouadio et al.,

2020; Zhang et al., 2021). Compared to most geophysical technologies, including ERT, CSAMT's subsurface assessment capabilities are superior due to its depth capacity of up to one kilometer (Zonge and Hughes, 1988; Hasan et al., 2024). When combined with empirically based methodologies, CSAMT becomes an even more powerful tool for studying the incredibly diverse topographical features.

Several factors, such as the type of rock, fault, weathering degree, fluid content, permeability, pore-spacing, fracture, lithology, saturation, and joints, as well as the same structural heterogeneities, determine the geophysical and aquifer characteristics (Singh, 2005; Sinha et al., 2009; Hasan et al., 2021). Several prior studies utilized geophysical parameters in conjunction with hydraulic data or lithological logs to characterize underlying rock mass units hydrogeologically (De Lima and Niwas, 2000; Hubbard and Rubin, 2002; Niwas and De Lima, 2003; Singh, 2005; Soupios et al., 2007; Sinha et al., 2009; Majumdar and Das, 2011; Nwosu et al., 2013; Hasan et al., 2021; Asfahani, 2023). Resistivity methods provide an alternate option for aquifer parameter estimation by creating a beneficial relationship between electrical resistivity and the aquifer parameters (obtained from drilling tests). An innovative aspect of this work is its use of non-invasive geophysical techniques to create two- and three-dimensional k models in a diverse environment with a variety of rock types and significant depths. The planned study will necessitate the boring of a handful of boreholes at key spots all around the project site. A more trustworthy CSAMT study will allow us to evaluate the extensive research area. Then, by directly connecting geophysical and borehole data, k can be established for the entire researched site, even without drilling tests. Two- and three-dimensional k models are generated by applying the resulting equations to the full study area. This approach would reduce the need for costly

boreholes to obtain a thorough and complete evaluation of subsurface hydrogeological conditions.

No one had ever tried to estimate K using direct or indirect methods in such a heterogeneous context before this work, where a broad diversity of rock types are present at a depth of 1 kilometer. Volumetric measurements of 2D/3D k have never been obtained in hard rock exploration using a geophysical technique. Furthermore, no previous research has previously derived permeability using the CSAMT method in the same way as this one. Our more precise 2D and 3D k model predictions of complex hydrogeological circumstances surpass prior investigations, bridging the gap between dependable hydraulic models and limited borehole data. The primary goals of this study were as follows: (1) to rapidly predict two- and three-dimensional k models using geophysical methods; (2) to reliably assess the hydrogeological properties of rock formations for deep groundwater assessments in challenging geological settings; (3) to minimize costly boreholes and maximize the use of scarce drilling resources to collect hydrogeological data over large areas; (4) to decrease uncertainties in hydrogeological models; and (5) to promote the use of non-invasive geophysical techniques for hard rock groundwater investigations instead of costly drilling that can damage the rock.

## 2 Methods

Using existing borehole data and a non-invasive CSAMT method, this study aimed to estimate k for two- and three-dimensional evaluation of groundwater resources over the project area (Fig. 1a). A flowchart depicting the primary stages of this method is shown in Fig. 1b.

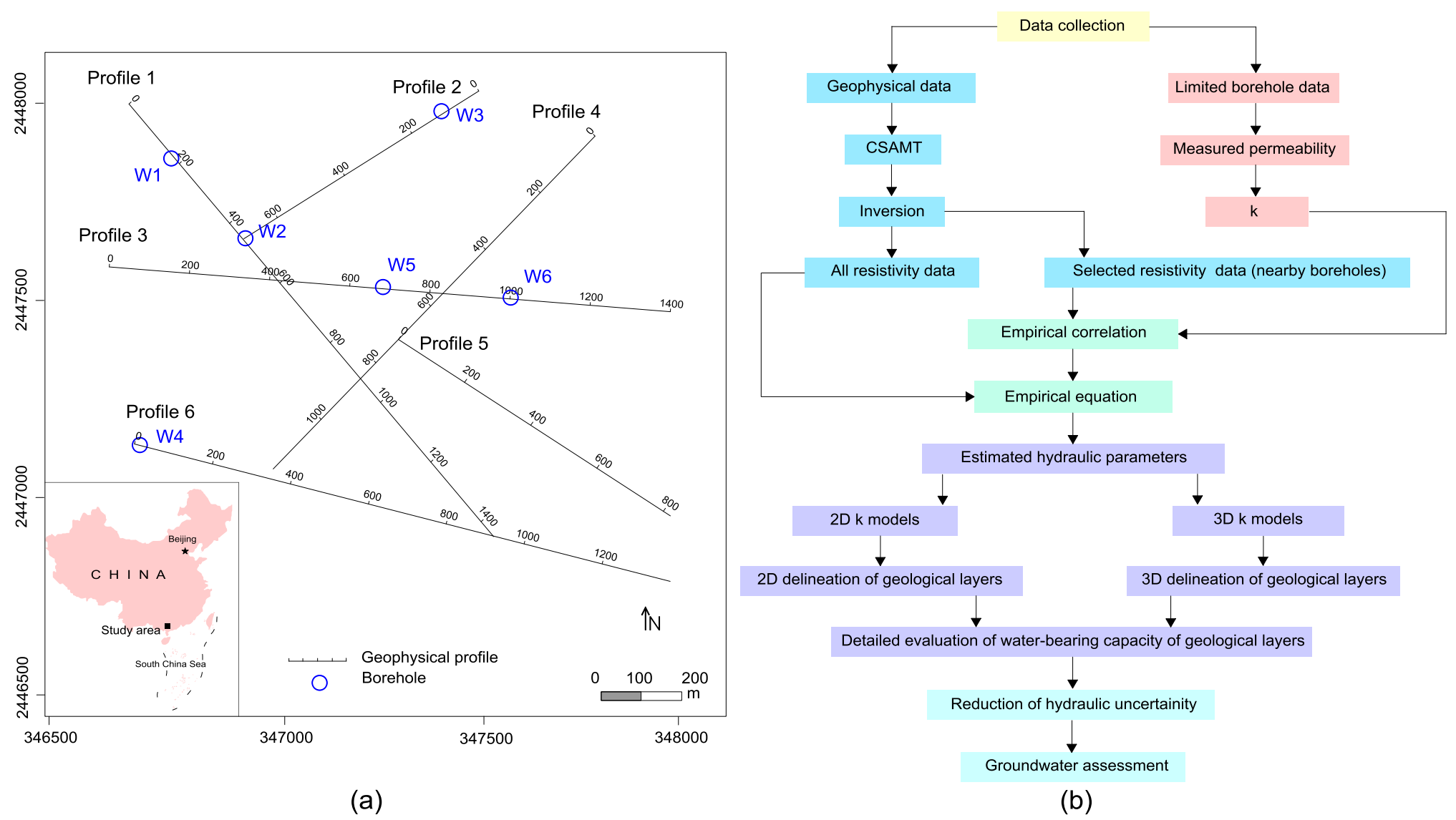

**Fig. 1. (a)** The location of the project site, with six boreholes W1–W6 (blue circles) and six CSAMT profiles 1-6 (black lines), **(b)** Flow diagram outlining the planned method for getting 2D and 3D k models for better, more thorough assessments of groundwater resources over large regions

### *2.1 Study area*

This study was conducted in the Jinji region of South China to explore deep underground water sources in a geologically diverse area (Fig. 1a). Due to its monsoon location, the study region experiences a concentration of precipitation in the summer, with an annual precipitation totaling 1981 mm. Rivers and other water features surround the Jinji area. The geomorphology of the project site is characterized by low, partly cut, and considerably depleted hills and mountains. The terrain in the north is somewhat flatter than that in the south. At a height of 39 to 447 meters above sea level, the region is famous for a number of characteristics, including a wide range of terrain slopes, from gentle to steep, abundant vegetation, and worn mountain rocks (Yang et al., 2021). Noteworthy among these are Mounts Dashishan, Qilongding, and Jixinshan. The southern

part of the research area features the 539.9-meter-high Xikeng peak, the landscape's highest point. The Yongkouwei River, which flows through the northeastern part of the site under study at an elevation of around 7.5 meters, cuts through it. Intruding rocks from the Indosinian, Caledonian, and Yanshanian eras are among the many geological formations and periods represented in the study region. Other layers from the Paleogene period are also present. The most common types of rock that have been discovered are sandstone, granite, and hornstone. The complex Kaiping concave fault and fold systems were the dominant geological features in the project region, which were developed as a result of magmatic processes and various structures (Qin, 2017). Emergence of joint fissured features symbolizes the various tectono-geological periods, with the local tectonic line corresponding with the faults strike, especially in the northeast orientation (Yang et al., 2021).

## 2.2 CSAMT survey

### 2.2.1 Principle

The application of CSAMT in the study of hard rock is extensive, as shown in numerous publications (Simpson and Bahr, 2005; Bai et al., 2010; Fu et al., 2013; Wang et al., 2015; Wynn et al., 2016; Di et al., 2020; Zhang et al., 2021; Kouadio et al., 2023; Hasan et al., 2024). For these kinds of studies, a faraway transmitter sends regulated electric signals into the earth, while a receiving station keeps an eye on the electric and magnetic fields (Zonge and Hughes, 1988; Zhang et al., 2021). There is a mathematical relationship between the reflected depth and the frequency in a subsurface structure where different fields have varied propagation depths (Borah and Patro, 2019). By taking advantage of the fact that various rocks have varying electrical conductivities, it tracks variations in the strength of the magnetic field and the main field

potential (Cagniard, 1953; Zonge and Hughes, 1988). The signal's frequency components are derived from the time series of the EM field fluctuations using Fourier transforms (Simpson and Bahr, 2005). A field source that is artificially regulated is utilized in CSAMT. Electrodes spaced one to two kilometers apart can be used to measure the electric dipole source's electromagnetic field component. We can set up the wires that will connect the batteries to the current electrodes and the transmitter. The average field source transmitter-receiver distances range from 5 to 10 km, though this can vary with depth of investigation (DOI) and geological factors. Dividing the magnitudes of the electric and magnetic fields by two orthogonal directions is one method to find the subsurface resistivity. Fu et al. (2013), Zhang et al. (2021), and Hasan et al. (2024) identified several parameters that impact the resistivity associated with subsurface geology. These factors include fault fragmentation, water saturation, lithological changes in stratigraphic structures, pore fluid, porosity, and rock types. The vertical resolution of 5–20% can be assessed by CSAMT when exploring depths ranging from 20 to 1000 meters. The propagation frequency and subsurface resistivity are the basis of DOI. According to Borah and Patro (2019), a lower frequency and higher resistivity typically result in a higher DOI. The distance between stations determines the lateral resolution; typically, this is between ten and two hundred meters. According to Simpson and Bahr (2005), increasing the distance between stations results in a more robust received signal. At every station, a portable receiver can be used to process, amplify, filter, and record the signal. In order to pick up sent signals, electrode pairs, which include magnetic-field sensors and short grounded dipoles, are utilized. Effective survey planning can reduce the impact of radio transmitters, metal fences, power lines, and other potential sources of inaccurate CSAMT data. Plan, three-dimensional, fence, and cross-sectional views are all potential ways to display the modeled resistivity data.

### 2.2.2 Data collection

The CSAMT data was acquired using six profiles (1–6) with a 50 meter interval between each station. About 1300 meters was the depth of investigation (DOI) in the CSAMT investigation. We took scalar measurements using the TM Mode. Bidirectional measurements are taken of magnetic and electric field: perpendicular to the measuring line and parallel to it. When doing EMAP observations, the measuring stations must be connected in a sequential fashion and 50 m away from the electrode. We used Gain mode X1 and a 50 Hz linear filter in our arrangement. While the emission current lowers to 2.6–4.5 A at 7680 Hz, it peaks at 12–18 A at 1 Hz. The CSAMT data was collected using a V8 multifunction receiver with a TXU-30 transmitter, which was manufactured in Phoenix, Canada. The TXU-30, a multi-function transmitter with a 30-kilowatt output, may enable an exclusively geophysical approach with transmission voltages up to 1000 V, currents up to 20 A at 1000 V, and currents up to 40 A at 500 V. This GPS-enabled transmitter is ideal for deep investigation since it is compatible with common household three-phase 220 volt alternators. There were 34 separate frequency points utilized, spanning the range of 1–7680 Hz. In addition to collecting data, the V8 multifunction receiver may monitor data sent by other secondary receiving units. The principal receiver's three channels and tracks make this possible. The distances between the transmitter and receiver ranged from 9.3 to 12.5 kilometers. The non-polarized electrode was used to record the electric field signals. The signals were picked up by the AMTC-30 inductive sensor, which is designed for high-frequency AMT/CSAMT magnets and operates between 10,000 Hz and 0.1 Hz. We acquired two orthogonal electric field components and three orthogonal magnetic field components at each site, and then we measured the tensor at each location. Here, the information was sourced from the American-made Trimble XH dual-frequency GPS receiver. Using the Hi-Tech V30GNSS

RTK apparatus, we measured the CSAMT lines for object recognition. Modern navigational aids allow for pinpoint accuracy on the order of sub-meter precision. The computer calculated the coordinate values of each survey line and survey point using the given direction and distance, and then sent them to the GPS or RTK. We found the measuring points of the survey lines using the RTK or GPS navigational capabilities. The distribution of inspection points was quite consistent when testing them for system quality within a 3–5% range along the measurement lines. The conclusions of the system quality assessment fulfilled the following design criteria: a root-mean-square (RMS) value below ±5%, a tolerance for consecutive points on the profile to have an error of less than 10, a tolerance for relative elevation of 1.67mm, and a tolerance for plane of 2.33mm. Because the experiment site was free of human and electrical interference, the data collected was of very high quality. The characteristics of the location were identified by the examination of the CSAMT data (An and Di, 2016; Hasan et al., 2024). Following the elimination of the skewed data, a curve analysis was conducted. The static corrections were made using a Hanning window spatial filtering method, which involved geological information and curve analysis. Correct data processing and interpretation were thus facilitated by the availability of high-quality geophysical data.

**2.2.3 Data processing**

For the data processing step, we used the CMTPro Version software produced by Phoenix Geophysics (Phoenix Geophysics CMTPro, 2020). This program combines source current, reference track data, and V8 data into CMT files, corrects electrode coordinates, automatically smoothes observed curves, and generates files in the AVG format, among other things. Fig. 2 shows a flow diagram of the CSAMT-SW method (Phoenix Geophysics CSAMT-SW, 2020) that was used to conduct the 2D inversion (Rodi and Mackie, 2001; Wang et al., 2015). Here are

the main components of the CSAMT-SW: 1. Data translation from AVG to D format; 2. Editing and creating CHK elevation files, and converting them to D format; 3. Manually checking the D file for corrupted sectors, filling in the gaps, removing near-field data, and skipping to certain spots; 4. Inversion outcomes from different static correction methods were very similar when compared; the D file was utilized for smoothing processing; 5. D, H, K, and Z are the four static correction results files for various correction approaches; 6. Converting to text files using BOSTICK inversion and near-field correction; 7. Finite (limited) layers containing data indicating resistivity in depth sections can be created using quasi-2D inversion and the CSAMT global field model (ID), which mixes near and transition fields, applied directly to the measured data from CSAMT. We stored the output as * _BOS.DAT and * _BSS.DAT, respectively, for data in the D file after applying Bostick inversion (Fusheng et al., 2022). The modified data is stored in D files, according to the specifications of the 2D inversion model of CSAMT, in a newly created * _M. DMT text file. Once the maximum number of iterations or RMS error is reached, in this case, 5 iterations, the resulting models are fitted with the observed observations using the inversion approach. A trustworthy 2D resistivity model (Zhang et al., 2021) of CSAMT was produced, considering the local geology and dataset quirks, by employing the most appropriate processing and inversion procedures to reduce model errors. By showing changes in resistivity, the final inversion model improved our knowledge of the subsurface geological characteristics.

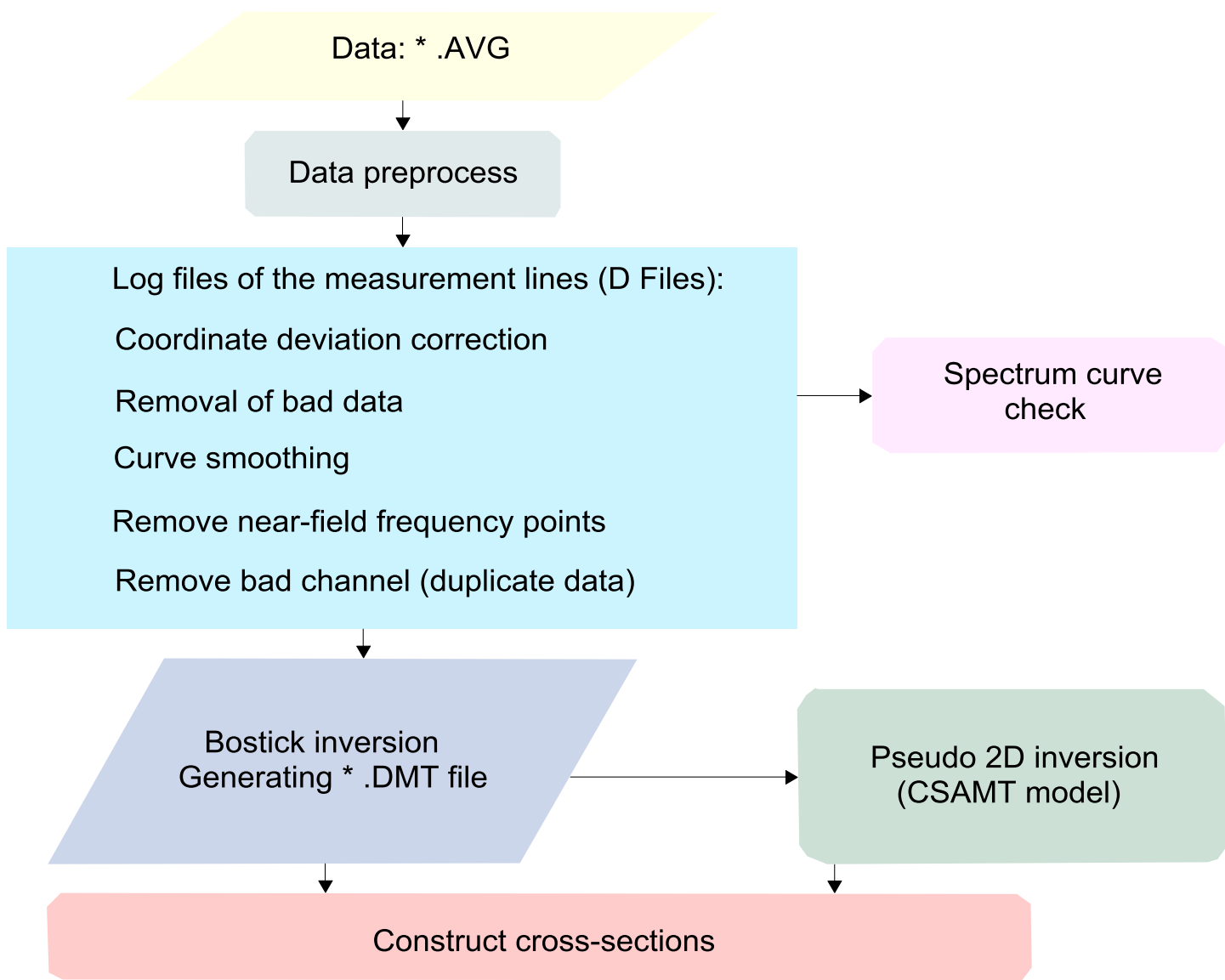

**Fig. 2.** Displaying the procedure of 2D inversion of CSAMT data by the use of Bostick inversion

### 2.3 Estimation of permeability (k)

In groundwater studies, permeability (k) is frequently employed to identify the groundwater flow characteristics in aquifers (Allègre et al., 2016; Fiandaca et al., 2018; Mudunuru et al., 2022; Esmaeilpour et al., 2023; Carbillet et al., 2024). Porous materials, such as rocks or unconsolidated materials are characterized by their capacity to permit water to permeate through them. This property is measured by their permeability. An essential factor regulating subsurface flow at any depth is permeability, or the ease with which fluids can pass through porous materials. The permeability of an aquifer is traditionally determined by expensive drilling and laboratory tests. Fluid content, flaws, saturation level, rock composition, faults, compaction, deformation, joints, and cracks are some of the factors that affect permeability (k) (Dewandel et al., 2004; Yan et al., 2024).

A large number of researchers have found substantial evidence that geophysical and hydrological features are correlated (De Lima and Niwas, 2000; Hubbard and Rubin, 2002; Niwas and De Lima, 2003; Singh, 2005; Soupios et al., 2007; Sinha et al., 2009; Majumdar and Das, 2011; Asfahani, 2023). The first step in establishing these connections is to calculate the permeability using drilling data collected at certain points. The next step is to integrate the electrical resistivity (using geophysical data) and permeability (measured in boreholes) in order to derive the empirical equations. After that, we put all of the resistivity readings from the six profiles into the resulting equation to get the overall site permeability. This makes it possible to find the site's complete permeability, even when a borehole is inaccessible. In order to provide a one-dimensional estimate of permeability, most of the earlier empirical geophysical approaches relied on vertical electrical sounding (VES), which was most common in uniform settings at shallow depths. As a result, no one had ever used k to probe deep underground water, especially in hard rock formations. We used a CSAMT-based empirical approach to estimate two and three dimensional k across a vast area with different rock formations at great depths for the first time in our recent research.

Initially, 37 k measurements were obtained from six boreholes (W1, W2, W3, W4, W5, and W6) at varying depths ranging from 10 to 200 meters (Fig. 3a). In the second stage, 37 borehole-derived k values were empirically correlated with 37 resistivity values from the chosen CSAMT soundings. P1-5 represents the fifth sounding at 200 meters on surveyed line 1 with well W1; P1-9 denotes the ninth sounding at 400 meters along profile 1 with well W2; P2-3 indicates the third sounding at 100 meters with profile 2 and well W3; P6-1 signifies the first sounding at 0 meters on surveyed line 6 with well W4; P3-15 refers to the fifteenth sounding at 700 meters along profile 3 with well W5; and P3-21 corresponds to the twenty-first sounding at

1000 meters on surveyed line 3 with well W6. In the third stage, the empirical integration of the selected observations (37 data sets) of CSAMT-based resistivity and borehole-based k was utilized to formulate the subsequent equation (Fig. 3b):

$$k = 15.345(e)^{-0.002(\rho)} \tag{1}$$

where k stands for permeability, which is measured in m/d units, and ρ signifies the true or inverted resistivity, denoted in Ωm. Using comprehensive resistivity data from six geophysical surveyed lines, Eq. (1) was used to predict permeability (k) over the entire area. In this way, we were able to assess the water-retaining capacity of three rock types: granite, hornstone, and sandstone. This allowed us to conduct a thorough evaluation of groundwater resources from 0 to 1300 meters below the surface across three potential aquifers: low potential aquifer (LPA) connected to granite, medium potential aquifer (MPA) contained inside hornstone and high potential aquifer (HPA) linked to sandstone). Finally, two- and three-dimensional models were created using the Geosoft and SKUA-GOCAD software tools for the k parameter, a predicted hydrogeological feature that extends throughout all 1-6 geophysical profiles (Webring, 1981; Mira Geoscience Ltd, 1999; Hasan et al., 2024).

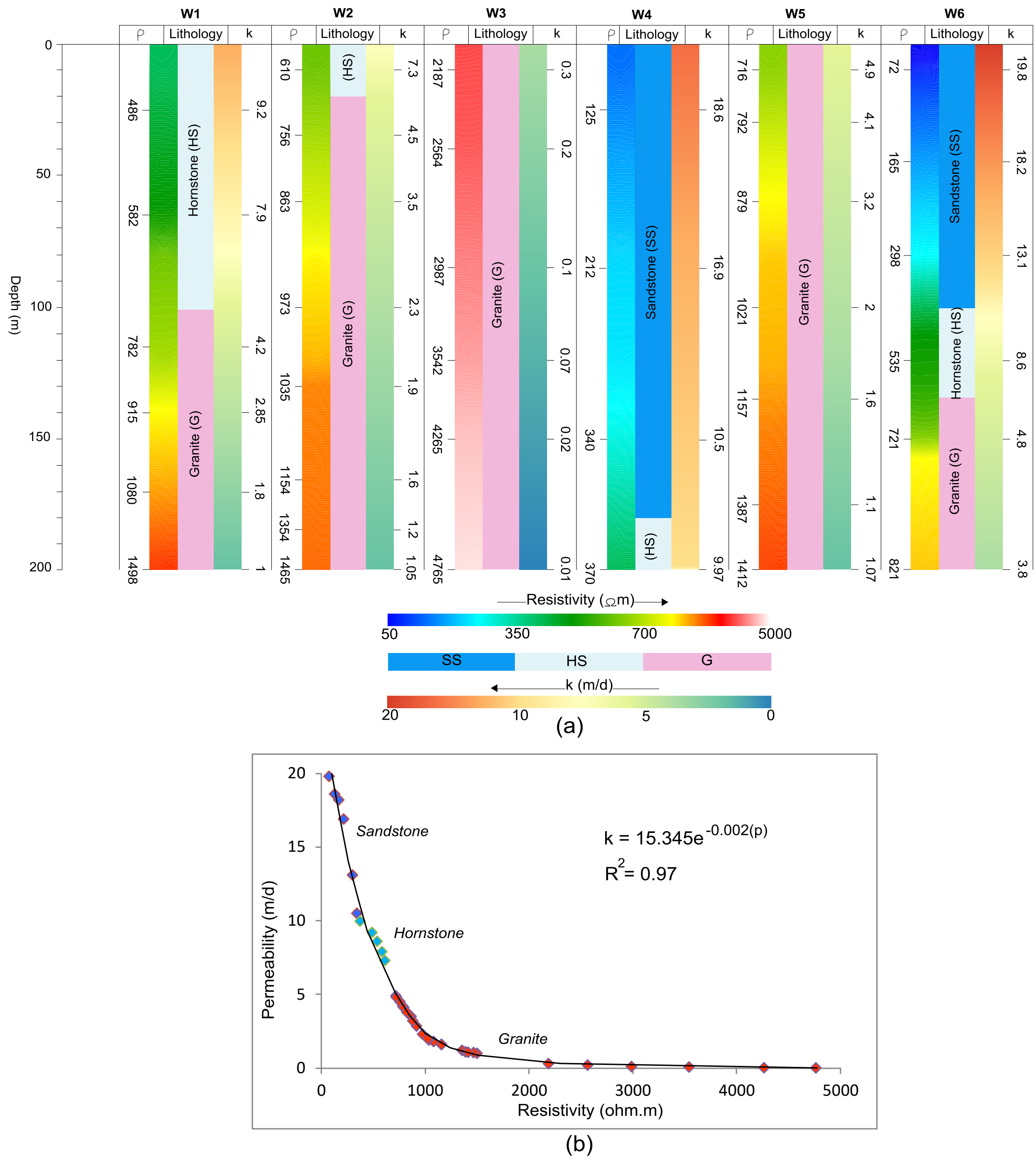

**Fig. 3. (a)** The evaluation of hornstone (HS), sandstone (SS), and granite (G) carried out by presenting 37 resistivity-k data points at depths ranging from 10 to 200 m using 6 drilled tests (W1–W6) and associated resistivity (ρ) from CSAMT soundings; **(b)** Using a total of 37 data points, the geophysical-borehole correlation for the predicted k

## 3 Results

### 3.1 Geophysical-borehole correlation

Table 1 displays the combined data from six boreholes and six CSAMT profiles that were utilized to stratify the underground formation into three distinct layers based on the electrical resistivity and permeability (k) ranges. Data from borehole and CSAMT-based resistivity measurements, as well as the research area's geological context, were used to build the subsurface hydrogeological models. Hornstone, sandstone, and granite are the three separate geological layers that make up these models. When sandstone, hornstone, and granite were being evaluated, the following factors were considered: sandstone's resistivity must be less than 350 Ωm and the k range must be between 10 to 20 m/d; hornstone's resistivity must be between 350 and 700 Ωm and the k range must be between 5 and 10 m/d; and granite's resistivity must be greater than 700 Ωm and k between 0 and 5 m/d. Based on our evaluations of the subsurface hydrogeological model's aquifer potential zones, we found that sandstone contains the high potential aquifer (HPA), hornstone contains medium potential aquifer (MPA), and granite has low potential aquifer (LPA). Aquifers with the largest yields or the best water-bearing capacity are indicated by sandstone, whereas aquifers with the lowest yields or the worst water-bearing capacities are denoted by granite. Groundwater development is best facilitated by sandstone in the study area, whereas groundwater extraction is most hindered by granite.

**Table 1**

Integrating the separate ranges of electrical resistivity and permeability (k) allows for a comprehensive evaluation of groundwater in different types of hard rock

| Resistivity | k | Type of rock | Aquifer potential |
|---|---|---|---|

| (Ωm) | (m/d) | | |
|---|---|---|---|
| < 350 | 10–20 | Sandstone | High potential aquifer (HPA) |
| 350–700 | 5–10 | Hornstone | Medium potential aquifer (MPA) |
| >700 | 0–5 | Granite | Low potential aquifer (LPA) |

### 3.2 2D groundwater assessments

Using geophysical-borehole correlation as its basis, Eq. (1) efficiently converts two-dimensional CSAMT models into two-dimensional k models, as shown in Fig. 4. Fig. 5 and 6 show that, in contrast to the limited drill experiments, geophysical-based 2D k models allow for an accurate and comprehensive assessment of the groundwater resources in hard rock across the whole research area, from 0 to 1300 meters deep. Line 1 of the survey has had the following geological layers marked out for the purpose of groundwater evaluation: A sandstone layer of high potential aquifer, 85 to 305 meters thick, is visible between 245 and 380 meters of distance, at depths ranging from 205 to 400 meters. From 0 to 1300 meters below surface, at concentrations of 0 to 525 meters and 1185 to 1445 meters away, the remaining portion of the profile is composed of a medium potential aquifer embedded in sandstone. Distances of 0–285 m within 290–790 m depth, 385–1185 m between 0–1300 m depth, and 1305–1450 m within 390–745 m depth were used to assess granite aquifers with poor potential. Along profile 2, the geological layers that were employed for groundwater assessment are described as follows: A hornstone layer 140–380 meters thick encloses a medium potential aquifer 490–1105 meters below ground, more precisely between 145–215 meters and 290–645 meters distance. We did not detect any sandstone associated with the high potential aquifer along this profile. Along this profile, granite from low potential aquifers predominates, with the exception of the zones evaluated by medium potential

hornstone aquifers; granite is located at 0–700 m distance between 0–1300 m depths. Along profile 3, the following geological layers have been characterized for the purpose of groundwater evaluation: A hornstone-containing medium-potential aquifer is evaluated at depths between 0 and 1300 meters and within a range of 0 to 1400 meters distance. Sandstone-associated high potential aquifers are located between 0 and 250 meters distance and between 0 and 1190 meters depths; 905 and 1065 meters away between 0 and 205 meters deep; and 1040 and 1390 meters distance and between 490 and 1305 meters depths. Distances of 80–1015 m between 0–590 m depths, 395–845 m between 915–1300 m depth, and 1100–1300 m between 200–500 m depth are used to assess the possible aquifers contained beneath granite. Here is the breakdown of the geological layers in Profile 4 for the purpose of groundwater assessment: The hornstone medium potential aquifer is checked at distances of 0–105 m and depths of 0–340 m. There is a layer about 290 m thick hornstone at depths ranging from 0 to 1300 m between 340 and 645 m distances, with depths of 0–300 m between 595 and 790 m profile spread, and 0–345 m deep between 1015 and 1145 m distance. No sandstone that could contain a high-potential aquifer is being investigated along this profile. The low potential aquifer associated with granite is delineated at most portions of the profile at 0–1145 m distance between 0–1300 m depths, excluding the areas with medium potential aquifer of hornstone. The geological layers that were considered for the groundwater assessment along profile 5 are as follows: Between 190 and 845 meters beneath the granite, there is hornstone associated with a medium-potential aquifer, which is located between 390 and 1325 meters below the surface. Two small sandstone patches of high yield aquifer can also be seen along this profile. One is at a distance of 290 m, between 790 and 960 m depth, while the other is at 815 m, between 1045 and 1135 m depth. Within the depth range of 0–1300 meters, granite from low-potential aquifers is assessed at a distance of 0–190

meters, and between 0 and 1025 meters, at a distance of 790–815 meters. Here are the geological layers that can be used for groundwater assessment along profile 6: To assess the high potential aquifer linked to sandstone, distances of 0–190 m between depths of 0–490 m and 1245–1345 m between depths of 215–1225 m are utilized. Distances of 0–690 m within depths of 390–1300 m and 790–1360 m within 0–1190 m depths are used to evaluate low yield aquifer granite. Between 0 and 1300 meters depth and 0 and 1350 meters distance, the hornstone of the medium potential aquifer dominates the rest of the profile. In the southeastern and northwest regions, there are a lot of medium to high potential aquifers, according to the results of the integrated 2D k models shown in Fig. 5 and 6. On the other hand, in the central areas, groundwater resources are scarce or nonexistent.

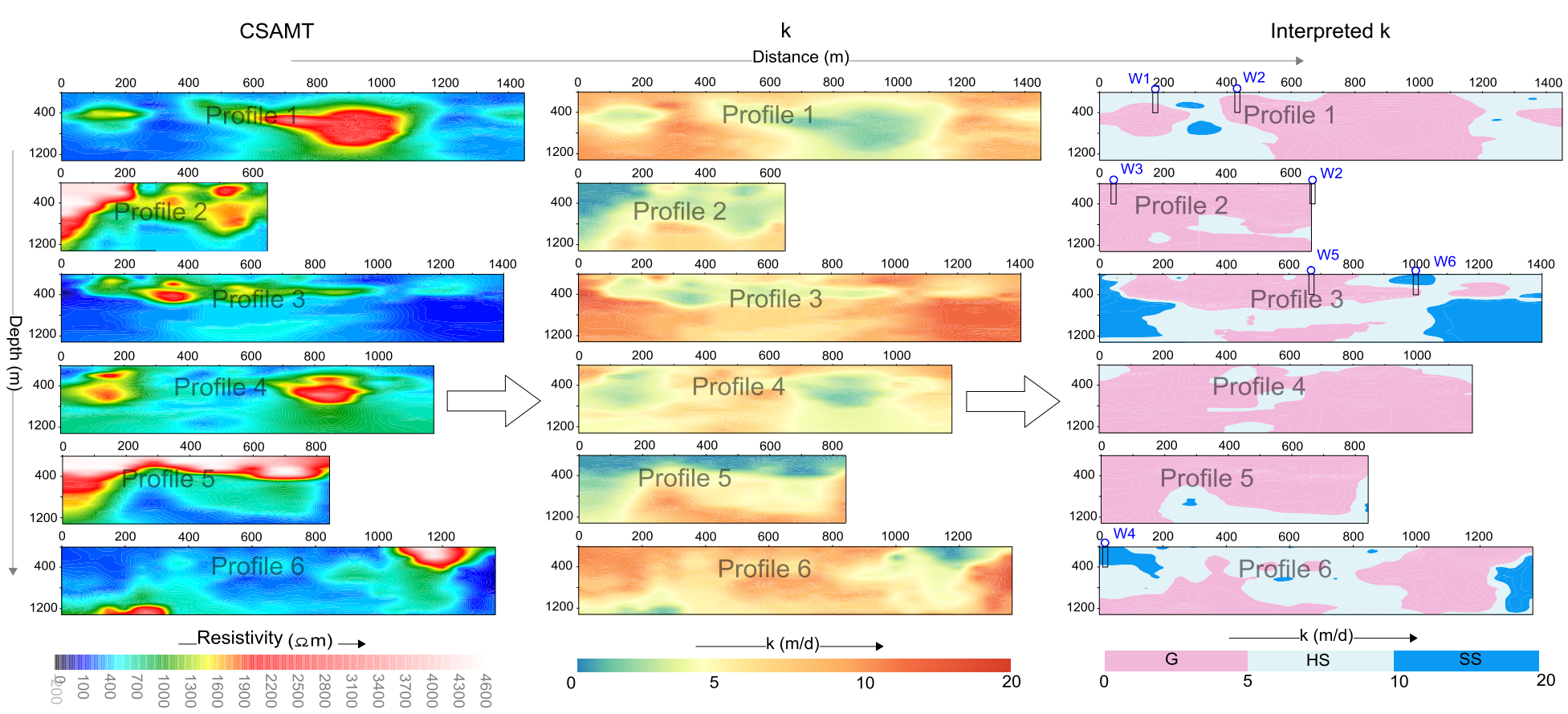

**Fig. 4.** The transformation of 2D CSAMT models (for profiles 1–6) into 2D k models, along with the interpretation of these models via geophysical-borehole correlation, facilitates groundwater assessment through high potential aquifer (HPA), medium potential aquifer (MPA), and low potential aquifer (LPA) associated with sandstone (SS), hornstone (HS), and granite (G), respectively

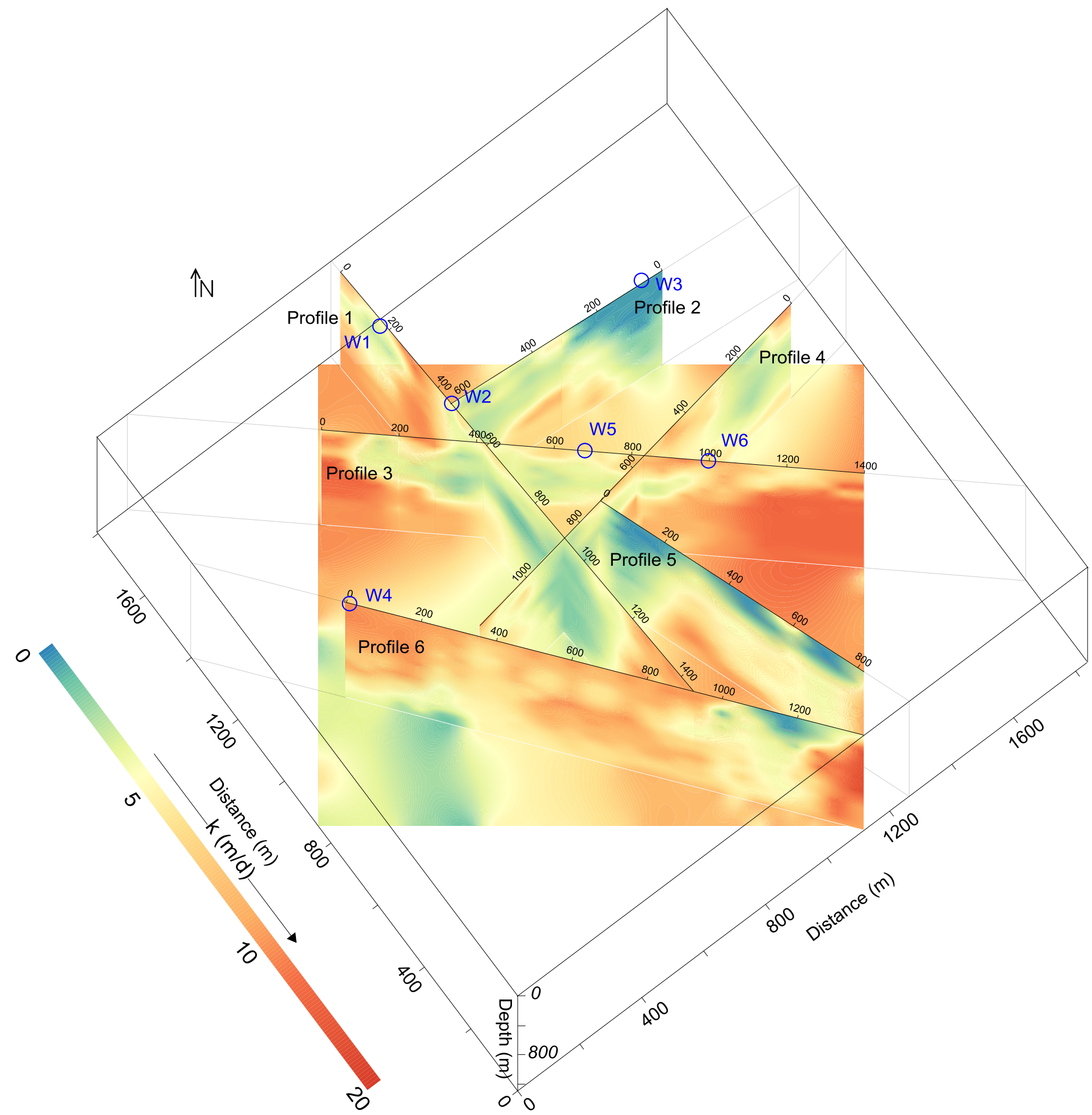

414

415 **Fig. 5.** The integrated 2D k models derived from the incorporation of geophysical and drilling data, with k

416 represented on a color bar spanning from green to red

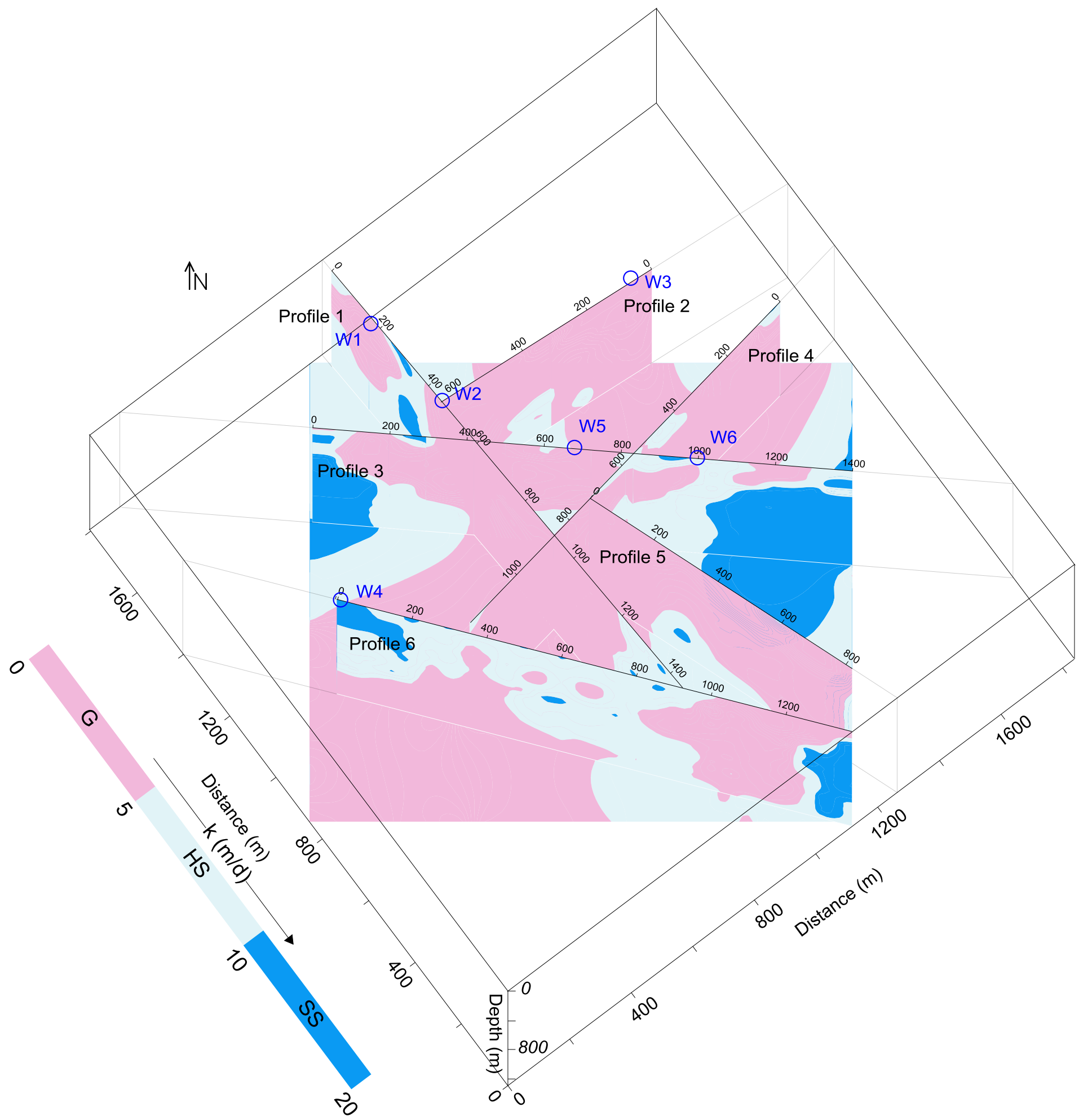

Fig. 6 Analysis of 2D k models (derived from designated k ranges) for three groundwater potential aquifers: low potential aquifer (LPA), medium potential aquifer (MPA), and high potential aquifer (HPA), associated with three geological formations: granite (G), hornstone (HS), and sandstone (SS), respectively

## 3.3 3D groundwater assessments

A thorough assessment of the water-bearing capacity of the rock mass for groundwater evaluation was conducted using the 3D k external visualization depicted in Fig. 7 (a, b). The granite of low potential aquifer was evaluated at the ground surface along profile 1 at distances

of 85–215 m and 385–1175 m, surveyed line 2 at 0–655 m, CSAMT line 3 at 0–45 m, 95–175 m, 265–585 m, 605–845 m, and 1145–1315 m, line 4 at 90–390 m, 490–615 m, and 745–1115 m, line 5 at 0–815 m, and surveyed line 6 at 1045–1345 m. A medium potential aquifer within hornstone was identified along profile 1 at distances of 0–95 m, 190–260 m, 295–415 m, and 1185–1425 m; along profile 3 at 40–105 m, 215–275 m, 580–605 m, 850–910 m, 1010–1155 m, and 1310–1410 m; along profile 4 at 45–90 m, 390–490 m, 590–685 m, and 1115–1185 m; and along line 6 at 90–190 m, 215–275 m, 315–485 m, 505–605 m, and 635–1045 m. The sandstone with significant aquifer potential was assessed across many locations: profile 1 at distances of 265–310 m, line 3 at 235–255 m and 915–1010 m, profile 4 within 0–45 m, and profile 6 at 0–90 m, 210–25 m, 275–305 m, 515–525 m, and 605–635 m. Fig. 7 indicates that elevated aquifer yield is predominantly concentrated in the southern regions.

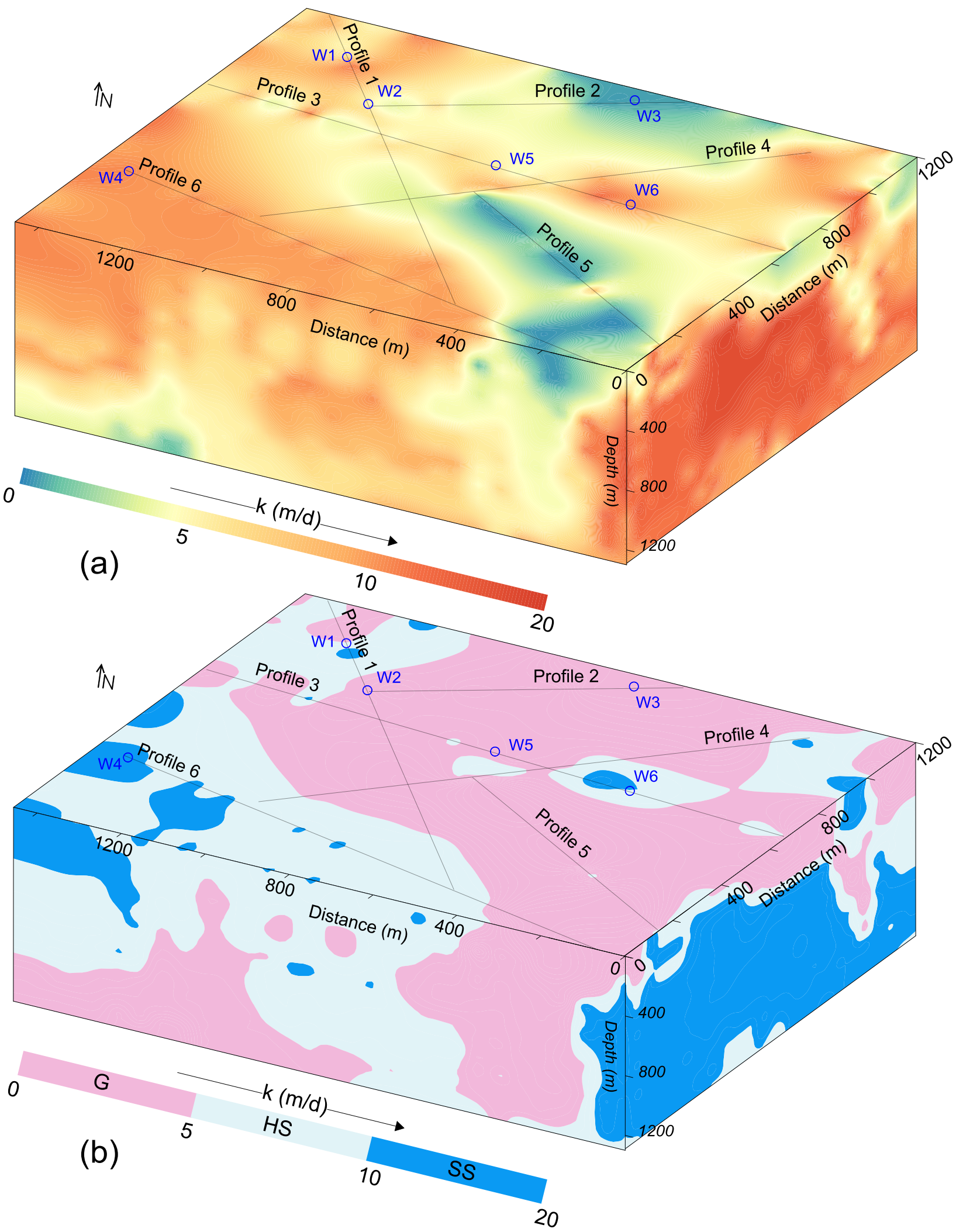

**Fig. 7.** The 3D k models, generated from the correlation of CSAMT and borehole data (with k represented on a color scale ranging from green to red), correspond to three groundwater potential aquifers: low potential aquifer (LPA), medium potential aquifer (MPA), and high potential aquifer (HPA), associated with three geological strata: granite (G), hornstone (HS), and sandstone (SS), respectively, for **(a)** the external view of the 3D k model, and **(b)** the analysis of the 3D k model from an external perspective

Fig. 8 (a, b) presents a comprehensive evaluation of the rock mass's aquifer potential for groundwater assessment using a 3D internal perspective. At a subterranean depth of 1300 m, the

low aquifer yield of granite was assessed using profile 1 across a distance of 515–1215 m, profile 2 across 0–290 m, profile 3 across 390–690 m, profile 4 across 0–1145 m, profile 5 across 0–195 m and 565–595 m, and profile 6 across lengths of 0–695 m and 1075–1115 m. Hornstone associated with a medium potential aquifer was identified by profile 1 at intervals of 0–545 m and 1215–1445 m, profile 2 at 295–675 m, profile 3 at 175–395 m, 445–815 m, and 915–1035 m, profile 5 at 205–565 m and 610–815 m, and surveyed line 6 at 685–1080 m and 1110–1355 m. An aquifer with high potential, situated within sandstone, was evaluated along profile 3 at intervals of 0–205 m and 1010–1400 m, as well as along line 5 at 810–815 m. Medium to high potential aquifers, located at a depth of 1300 m, are predominantly found in the southeastern and northwestern regions, whilst the central areas are primarily characterized by low potential aquifers. Fig. 8 illustrates the results of the 3D K analysis, indicating that the northeastern and southwestern regions are primarily composed of granite with negligible aquifer yield. The water retention capacity of the rock mass is enhanced when observed from an aerial perspective. This enables a precise assessment of the aquifer potential of geological strata for thorough groundwater analysis via 3D k modeling.

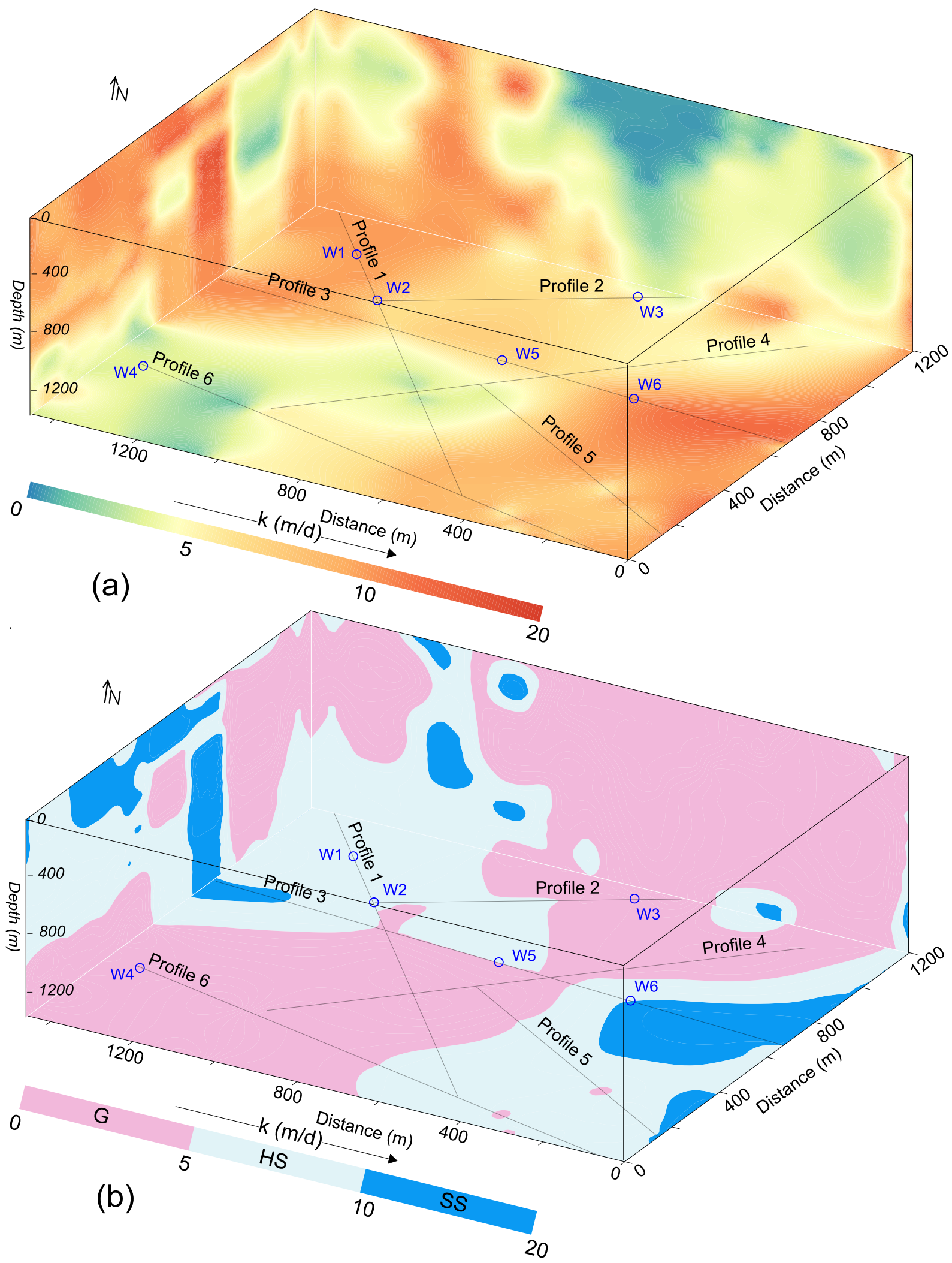

**Fig. 8.** The 3D k models, obtained from the correlation of CSAMT and borehole data (with k represented on a color scale ranging from green to red), illustrate three groundwater potential aquifers: low potential aquifer (LPA), medium potential aquifer (MPA), and high potential aquifer (HPA), associated with three geological strata: granite (G), hornstone (HS), and sandstone (SS), respectively, for **(a)** the internal view of the 3D k model, and **(b)** the analysis of the 3D (internal perspective) k model

### 3.4 Groundwater assessments via depths

Because of the scarcity of borehole data, it is not possible to use the observed k (borehole-based k) to determine the water-bearing capacity of rock masses located below 200 m deep. The evaluation of hard rock groundwater resources was made efficient, precise, and comprehensive by creating a high link between drilling and CSAMT data. Because of this, k could be quickly and accurately determined up to depths of 1300 m. From 2D/3D groundwater yield insights, we were able to derive anticipated k values at 0, 200, 600, 1000, and 1300 m depths (Fig. 9). Evaluation of groundwater at a depth of 1300 meters was based on the following criteria: The southwest and northeastern regions are assessed for granite, which constitutes over 45% of the subsoil in low potential aquifer locations. Near the granite formation in the northwest and southeast, we looked into hornstone, which comprised 40% of the medium potential aquifer. In the eastern region, subsurface assessments were conducted on high-yield sandstone for over 15% of the total. For groundwater evaluation at a depth of 1000 m, the following criteria were used to understand the subsurface: The subsoil around the high-potential aquifers in the southeast consisted of 14% sandstone. Near the granite in the southeast and northwest areas, 38% of the hornstone belonged to a medium-potential aquifer. There were three boundaries, in the middle, to the northeast, and to the southwest, in the subsurface, which was 52% granite and had a poor aquifer yield. We examined the hydrogeological conditions at 600 meters below ground using the following criteria: in the central and northern areas, a low-yielding granite aquifer constituted 55% of the subsurface; in the western areas, hornstone was more common, accounting for 32% of the subsurface and indicating a medium-yielding aquifer; and in the southeastern regions, sandstone was the most studied, constituting 13% of the subsoil and indicating a high-yielding aquifer. To assess the hydrogeological conditions at a depth of 200 meters, the following criteria were used: Granite with a low potential aquifer constituted 64% of the total in the center and

northern parts. Hornstone with a medium yield aquifer comprised 26% of the underground in the southern regions. Research in the west focused on sandstone that made up 10% of the subsoil and had a high potential aquifer. Surface measurements taken at a depth of 0 meters allowed us to determine the following hydrogeological conditions: While a medium-potential aquifer is contained within 22% of the southwesterly surface's hornstone, 69% of the subsurface in the central sections is granite. The sandstone, which is primarily located in the southwest, contains a high-potential aquifer and is studied on 9% of the surface. Fig. 9 shows that as we descend then thickness of the granite from low yield aquifers decreases. Midway through, when depth drops to 600 to 700 m, groundwater conditions are at their worst. In the northwest, southeast, and southwest areas, there are rock masses that could represent aquifers, especially at depths lower than 700 meters.

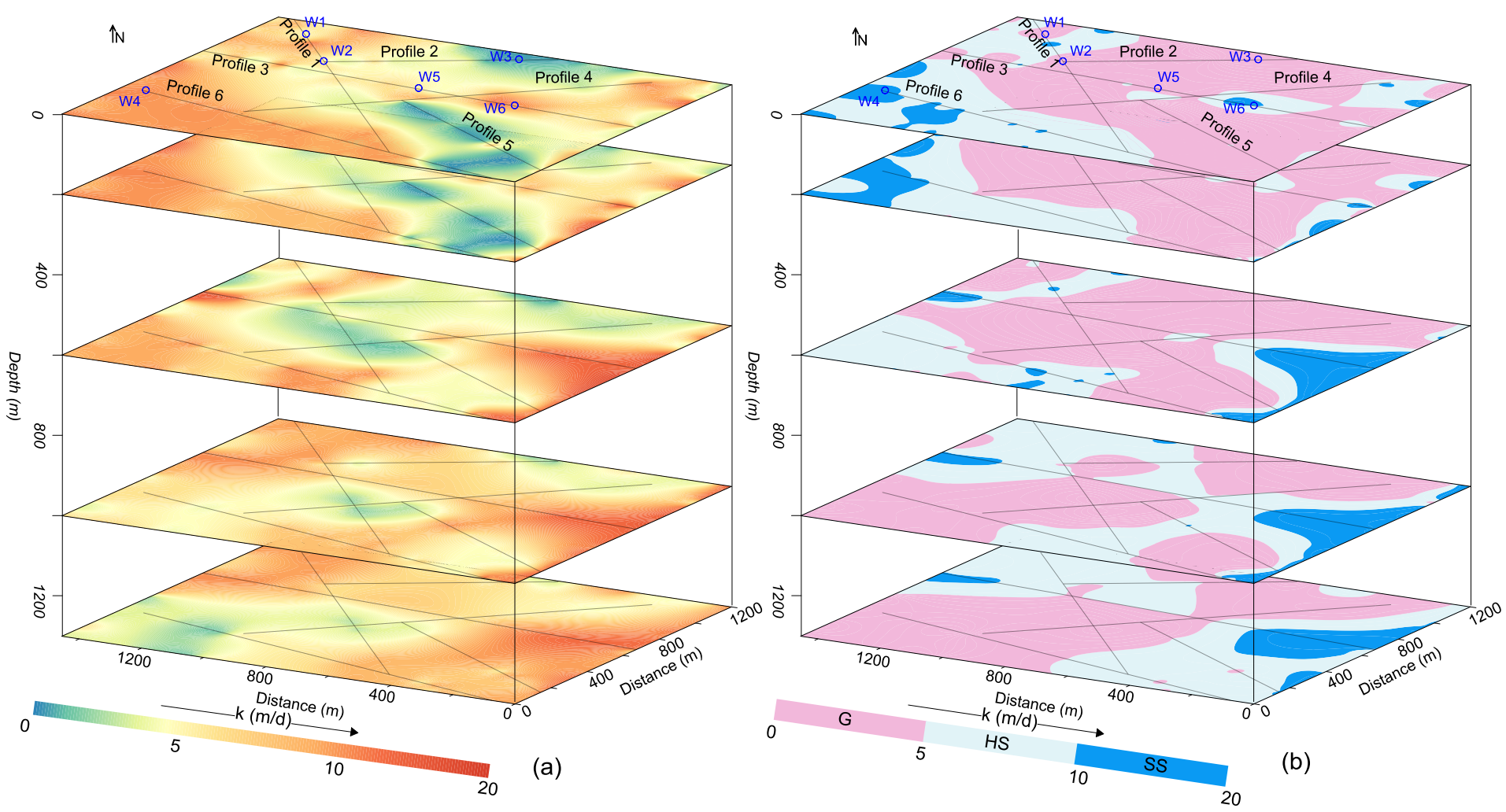

**Fig. 9**. **(a)** Geophysical k imaging at various depths (0, 200, 600, 1000, and 1300 m) is represented by K on a color bar that goes from green to red, **(b)** Assessment of geophysical-derived k (using specified k ranges) at different depths for various types of aquifers: low potential aquifer (LPA) granite (G), medium potential aquifer (MPA) hornstone (HS), and high potential aquifer (HPA) sandstone (SS)

### 3.5 Comparison of the predicted and measured k

Groundwater evaluations across the project area can benefit from the precise and systematic evaluation of water-bearing yield provided by the k generated using CSAMT. Based on the k results (Fig. 4–9), it can be seen that granite is studied in the central, northeastern, and southwestern areas, whereas in the southeastern, western, and northwest regions, hornstone is typically identified as a type of sandstone. Sandstone undergoes extensive analysis in the eastern areas but just minor appraisal in the western sections. It is challenging to evaluate the water-bearing capacity of rock masses using data acquired from boreholes because the resulting incompatible mapping of subsurface geological layers is problematic. Drilling results do match the CSAMT k at a few spots at 200 m deep close to the boreholes. Thus, measured k (obtained via drills) compared to predicted k makes groundwater potential estimations across big regions uncertain.

Table 2 presents the percentage of matching for the selected metrics, ascertained by juxtaposing the drill-k with the CSAMT-k. We compared the predicted k with the borehole-based k for several selected data points and observed the following % agreement: Upon experimentally connecting the first well, W1, to the fifth sounding along surveyed line 1 (P1-5), the percentage match at depths of 25 m and 115 m is 85% and 88%, respectively, as per Eq. (1). Following the application of Eq. (1), the percentage matching for sounding 9 along profile 1 (P1-9) and well number two (W2) at a depth of 10 meters and 130 meters, respectively, is 90% and 100%. The percentage matching is 70 and 50, respectively, when W3 (well number three) and P2-3 (third sounding on line 2), which have depths of 85 and 150 meters, are merged, as per Eq. (1). We obtain %matching values of 82 and 85, respectively, with depths of 10 and 45 meters, by plugging the data from W4 (the fourth well) and P6-1 (the first sounding at line 6) using Eq. (1).

We may acquire %matching of 84 and 85 for depths of 60 and 200 m, respectively, by integrating W5, the fifth well, and P3-15, the fifteenth sounding at line 3, using Eq. (1). Additionally, at a depth of 80 and 120 meters, the integration of W6 (well number six) and P3-21 (the 21st sounding on line 3) using Eq. (1) results in a percentage matching of 95 and 85, respectively. The aforementioned comparison between the obtained and projected k indicates a lower degree of inaccuracy or strong matching. The comparison also demonstrates that predicted and measured k values generally fall into the same aquifer potential zone, even for data points with low %matching.

**Table 2**

The percentage matching comparison between drill-k and CSAMT-k for the chosen data points

| CSAMT data points (selected) | | | Drilling data | | | %Matching |
|---|---|---|---|---|---|---|
| CSAMT sounding number | Resistivity (Ωm) | Predicted k' using Eq. (1) | Borehole name | Depth (m) | Measured k | k' vs k |
| P1-5 | 486 | 7.8 | W1 | 25 | 9.2 | 85 |
| P1-5 | 782 | 3.7 | W1 | 115 | 4.2 | 88 |
| P1-9 | 610 | 6.6 | W2 | 10 | 7.3 | 90 |
| P1-9 | 1035 | 1.9 | W2 | 130 | 1.9 | 100 |
| P2-3 | 2987 | 0.07 | W3 | 85 | 0.1 | 70 |
| P2-3 | 4265 | 0.01 | W3 | 150 | 0.02 | 50 |
| P6-1 | 72 | 16.2 | W4 | 10 | 19.8 | 82 |
| P6-1 | 165 | 15.5 | W4 | 45 | 18.2 | 85 |
| P3-15 | 879 | 2.7 | W5 | 60 | 3.2 | 84 |
| P3-15 | 1412 | 0.91 | W5 | 200 | 1.07 | 85 |

| | | | | | | |
|---|---|---|---|---|---|---|
| P3-21 | 298 | 12.5 | W6 | 80 | 13.1 | 95 |
| P3-21 | 535 | 7.3 | W6 | 120 | 8.6 | 85 |

## 4 Discussions

Groundwater research is seeing a rise in the use of geophysical technology. Groundwater evaluations have shown promise in prior studies when geophysical and drilling data are combined. We can determine a rock mass's water-bearing potential by looking at its hydraulic properties. Drilling boreholes to measure permeability (k) is the best and most feasible way to measure hydraulic parameters for groundwater evaluations. Through the use of geophysical methods, this study is the first to indirectly obtain 2D/3D k at depths more than 1 km in a context with a diverse range of rocks.

This work introduces CSAMT, a novel geophysical method for assessing the water-bearing capacity of rock masses, which allows for more precise groundwater evaluation even when sufficient borehole data is unavailable. This paves the way for a comprehensive assessment of hard rock groundwater at depths above 1 kilometer by means of the anticipated hydraulic parameter k. In light of the varied terrain in southern China, our approach provides a versatile empirical correlation based on the region's massive geophysical dataset and limited drill data. The lithologies and rocks were categorized using the same set of k values as were utilized in the aquifer models. The rocks are classified into three groups based on different aquifer potential zones: fresh granite zone of low potential aquifer (LPA), sandstone zone of high potential aquifer (HPA), and hornstone zone of medium potential aquifer (MPA), which lies in the middle zone between the two possible aquifers. The calculations can be used to find the overall water-bearing capacity of the rock formation in these particular geological settings since they are based on

resistivity-k measurements of hornstone, granite, and sandstone. The precise parameter ranges are defined by taking into account the local environment and the rock's composition. The precise rock mass class of a possible aquifer can be determined using the well-established flexible equations, taking into account the hydrogeological conditions of the location. Any geological context can benefit from the generalized equations that can be derived using the suggested method. Because it is relative, a rock unit's k-resistivity range could change depending on are to area. Drilling five or more boreholes across the entire area, with at least five measurements collected from the rock unit in each, usually yields a reliable empirical equation. The reliability of the empirical equation is greatly affected by the ranges of k-resistivity and the quantity of geophysical-borehole datasets. A more precise calculation of k is possible with the use of more datasets in correlation analysis. With real and estimated k matching rates over 80%, most datasets showed an outstanding level of accuracy (Table 2). The established equation provides a poor fit between the anticipated and actual k, especially for very high resistivity and low k values. For instance, the anticipated and calculated k values are in the same LPA zone, even though there was only a 70% and 50% match between W3 and P2-3 at 85 and 150 meters of depth, respectively. All lithologies and rock types, including granite, hornstone, and sandstone, are represented in the resistivity-k ranges used for correlation analysis across the project territory. In order to get trustworthy results, the researchers may have used the drill locations as a proxy for the rock unit characteristics of the whole study site. It may be more accurate to use distinct formulae for each kind of rock unit to find k rather than using a single formula to evaluate several geological layers. While there is sufficient drilling data for each rock mass unit, separate equations may be more effective. Since this correlation is the basis for the expected k, the locations of the surveyed lines are crucial to its correct calculation. As a result, regions near

geophysical profiles yield somewhat more accurate findings from two- and three-dimensional k models than regions far from these profiles. We can estimate k in similar geological conditions using the resultant equation when drilling data is not available.

CSAMT is heavily utilized for reducing the effect of weak natural signals and for exploring subterranean structures. However, there are a number of factors that might influence resistivity measurements, including transmission devices, electrical lines, metal obstructions, etc., and this can cause results and interpretations to be unclear. Good CSAMT survey design, however, mitigates these effects and yields accurate results, as demonstrated in this study. Data of a high quality could be collected since the project site was free of electrical and human interference. With a resistivity of 28 Ωm and a k value of 20 m/d, the sandstone rock mass was found to have the highest water-holding capacity. However, the rock mass (granite) was found to have a minimal water-bearing capability of 0.01 m/d when the resistivity value was tested at 5000 Ωm. When comparing the geophysical k to the drilling k, the latter provides a more precise and comprehensive evaluation of the rock mass's water-bearing capacity while the former decreases the variability in the anticipated hydrogeological model. Due to a lack of boring trials, hydrogeological models used to evaluate groundwater in highly heterogeneous hard rock are thus profoundly flawed. When hydrogeological models are accurate but borehole data is insufficient, geophysical approaches can help fill the gap.

## 5 Conclusions

We present new ways to study deep groundwater using non-invasive equipment. To evaluate the water-bearing capacity of rock masses for deep groundwater evaluation across broad, diverse hard rock regions without drilling data, this research applies CSAMT for the first time ever to indirectly estimate two and three dimensional permeability (k) values. In groundwater research,

boreholes are the most common method for measuring hydraulic parameters. But drilling methods have a lot of downsides and cost a lot of money. Compared to traditional methods, this study's evaluation of the hydrogeological properties of rock masses is more thorough and precise, and it does so with fewer boreholes. We derived the flexible equation that determines k by examining CSAMT-drilling data from multiple areas in order to evaluate the water-holding capacity of rock formations in varied settings. For groundwater assessment across the entire study area, k was determined using a well-established equation, which allowed for a comprehensive evaluation of the rock mass's water-bearing capacity. With a resistivity below 350 Ωm and a k range of 10 to 20 m/d, the sandstone of a high potential aquifer (HPA) was studied in a three-layer hydrogeological model. The hornstone of a medium potential aquifer (MPA) was assessed with a resistivity rise ranging from 350 to 700 Ωm and a k range of 5 to 10 m/d. The resistivity value of the granite from the low potential aquifer (LPA) was greater than 700 Ωm, and the k values were between 0 and 5 m/d. As the resistivity decreases and the k parameter rises, the results show that the rock mass can hold more water. The assumption that sandstone is the rock mass with the greatest water-bearing capacity and granite is the rock mass with the least amount of groundwater was previously thought to underpin the expectations for deep groundwater resources in hard rock. Our 2D/3D k models forecasted that deep groundwater extraction would take place in the center regions at depths below 700 m and in the neighboring areas around granite at depths between 0 and 1300 m. The k-models have been found to have a strong relationship with the hydrogeology and local geology. Based on our findings, this approach has the potential to be a less costly substitute for costly drills in order to obtain more accurate hydrogeological modeling maps than what is now available. When investigating hard rock's groundwater, geophysical methods can evaluate the rock's water-retention capacity rapidly

and thoroughly, filling the gap between good hydrogeological models and inadequate drilling data. By utilizing groundwater hydrogeological principles to refine empirical equations, future research could improve the elucidation of aquifer properties. This method would be more useful in groundwater applications since it would enhance our knowledge of how geophysical and aquifer characteristics interact with one another.

**Code availability**

Software application or custom code supports the published claims and complies with field standards

**Data availability**

Data available on request from the corresponding author

**Author contributions**

MH conceptualized the research goals and developed the methodology. MH and LS found the funding for the project. MH developed the code and prepared its visualization, and LS provided programming support and analysis tools. MH prepared the original draft.

**Declaration of competing interest**

The authors declare that they have no conflict of interest.

**Acknowledgements**

The authors wish to acknowledge the institutions that facilitated the research for this study: the State Key Laboratory of Mountain Hazards and Engineering Resilience, Institute of Mountain

Hazards and Environment, Chinese Academy of Sciences, and China-Pakistan Joint Research Center on Earth Sciences, CAS-HEC, Islamabad, Pakistan.

## Financial support

This research was financially supported by the National Natural Science Foundation of China's Research Fund for International Young Scientists (RFIS-I) (Grant No. 42350410442).

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
