# Peer review of "Novel insights into deep groundwater exploration by geophysical estimation of 1 hard rock permeability 2 Muhammad Hasan 1, 2, 3, \*, Lijun Su 1, 2, 3, \*\* 1 State Key Laboratory of Mountain Hazards and Engineering Resilience, Institute of Mountain 4 Hazards and Environment, Chinese Acade"

_EGUsphere, 2024_

## Community Comment (CC2)

**Journal of Hydrology**

**Geophysical prediction of 2D and 3D hydraulic conductivity in deep hard-rock aquifers**
**--Manuscript Draft--**

| Manuscript Number: | HYDROL65101 |
| --- | --- |
| Article Type: | Research paper |
| Keywords: | Hydraulic conductivity (K);  Controlled-source audio-frequency magnetotellurics (CSAMT);  Hydraulic parameters;  Groundwater;  Hard rock;  Hydrogeological uncertainty |

| Abstract: | Future scenario prediction and efficient groundwater management depend on an accurate estimation of hydraulic parameters. One of the most common aquifer parameters studied in groundwater investigations is hydraulic conductivity (K). Conventionally, K is measured via boreholes. Traditional methods, on the other hand, have a number of drawbacks, including as the fact that they are intrusive, costly, time-consuming, and only provide point-scale K measurements; they are also not applicable to regions with very varied topographies. Besides, deep groundwater assessment might not be possible using conventional methods. Contrarily, geophysical methods can evaluate subsurface hydrogeological conditions across vast areas with less effort and without invasiveness, as well as at lower cost and in less time. K has previously been estimated through a number of empirically based geophysical investigations. However, the VES (vertical electrical sounding) approach was employed in these investigations to estimate only 1D K, primarily at shallow depths in a homogenous context. Because hard rock terrains are inherently heterogeneous, accurately assessing the aquifer potential associated with weathered layers and fractures/faults using borehole/VES-based K is problematic. To this end, this work employs the CSAMT (controlled-source audio-frequency magnetotellurics) approach for the first time to estimate 2D and 3D K over a depth of 1 km. In the extremely varied contexts of various rocks, the suggested approach evaluates the water-bearing capacity of geological layers and gives a more thorough and precise evaluation of groundwater potential. Compared with the past studies, these results provide a more accurate hydrogeological model, which in turn reduces the need for costly pumping tests and allows for a more thorough assessment of aquifer potential, which is essential for the scientific planning and management of groundwater resources in areas with very varied hard rock terrains where hydrogeological data is unavailable. |
| --- | --- |

**Highlights**

- For the first time, a non-invasive CSAMT method is proposed for 2D/3D K prediction

- K was first ever predicted over 1 km depth in heterogeneous setting

- Our approach reduces many boreholes for K prediction over large area

- This research, compared with past studies, provides deep groundwater assessment

**Geophysical prediction of 2D and 3D hydraulic conductivity in deep hard-**

**rock aquifers**

**Muhammad Hasan** [a, b, c,*], **Lijun Su** [a, b, c, **], **Peng Cui** [a, b, c, d], **Yanjun Shang** [c, e]

[a] *State Key Laboratory of Mountain Hazards and Engineering Resilience, Institute of Mountain*

*Hazards and Environment, Chinese Academy of Sciences, Chengdu 610299, China*

[b] *China-Pakistan Joint Research Center on Earth Sciences, CAS-HEC, Islamabad, Pakistan*

[c] *University of Chinese Academy of Sciences, Beijing 100049, China*

[d] *Institute of Geographic Sciences and Natural Resources Research, CAS, Beijing, China*

[e] *Key Laboratory of Shale Gas and Geoengineering (KLSGG), Institute of Geology and*

*Geophysics, Chinese Academy of Sciences, 100029 Beijing, P.R. China*

*Corresponding authors:

Muhammad Hasan: Email: mhasan@imde.ac.cn; ORCID: https://orcid.org/0000-0001-6804-

7962; Phone Number: +86-13051361710

Lijun Su: Email: sulijun1976@163.com; ORCID: https://orcid.org/0000-0001-9972-4698

Corresponding authors' postal address: State Key Laboratory of Mountain Hazards and

Engineering Resilience, Institute of Mountain Hazards and Environment, Chinese Academy of

Sciences, Chengdu 610299, China

Co-authors' emails:

Peng Cui: Email: pengcui@imde.ac.cn

Yanjun Shang: Email: jun94@mail.iggcas.ac.cn

**Abstract:** Future scenario prediction and efficient groundwater management depend on an accurate estimation of hydraulic parameters. One of the most common aquifer parameters studied in groundwater investigations is hydraulic conductivity (K). Conventionally, K is measured via boreholes. Traditional methods, on the other hand, have a number of drawbacks, including as the fact that they are intrusive, costly, time-consuming, and only provide point-scale K measurements; they are also not applicable to regions with very varied topographies. Besides, deep groundwater assessment might not be possible using conventional methods. Contrarily, geophysical methods can evaluate subsurface hydrogeological conditions across vast areas with less effort and without invasiveness, as well as at lower cost and in less time. K has previously been estimated through a number of empirically based geophysical investigations. However, the VES (vertical electrical sounding) approach was employed in these investigations to estimate only 1D K, primarily at shallow depths in a homogenous context. Because hard rock terrains are inherently heterogeneous, accurately assessing the aquifer potential associated with weathered layers and fractures/faults using borehole/VES-based K is problematic. To this end, this work employs the CSAMT (controlled-source audio-frequency magnetotellurics) approach for the first time to estimate 2D and 3D K over a depth of 1 km. In the extremely varied contexts of various rocks, the suggested approach evaluates the water-bearing capacity of geological layers and gives a more thorough and precise evaluation of groundwater potential. Compared with the past studies, these results provide a more accurate hydrogeological model, which in turn reduces the need for costly pumping tests and allows for a more thorough assessment of aquifer potential, which is essential for the scientific planning and management of groundwater resources in areas with very varied hard rock terrains where hydrogeological data is unavailable.

**Keywords:** Hydraulic conductivity (K); Controlled-source audio-frequency magnetotellurics (CSAMT); Hydraulic parameters; Groundwater; Hard rock; Hydrogeological uncertainty

**1. Introduction**

Hard rock, including igneous and metamorphic rocks, makes up around 20–35% of Earth's surface (Amiotte Suchet et al., 2003; Gao et al., 2024). Groundwater research in hard rock primarily involves the evaluation of subsurface geological layers, faults, and fractures (Fernando and Pacheco, 2015; Hasan et al., 2021). An important part of groundwater monitoring and assessment is classifying the aquifer yield (water-retaining capability) of rock mass (Majumdar and Das, 2011; Nwosu et al., 2013; Rao et al., 2022). Many factors naturally determine the aquifer potential of rocks that contain water. These include the type of rock, its association and deformation, fractures, the amount of water that can penetrate, the joints between rocks, the mineral composition, the rate of weathering, and faults (Maréchal et al., 2004; Slater, 2007; Vassolo et al., 2019). The main challenge in groundwater assessments is finding a way to measure the capacity of subsurface rocks to hold water both horizontally and vertically over large areas (Courtois et al., 2010; Dewandel et al., 2014). Groundwater extraction cannot proceed without first conducting a precise and comprehensive assessment of the aquifer potential associated with the various rock masses (Rao et al., 2022). Evaluating the water-carrying potential of geological layers is uncertain due to structural variability and a lack of data (Lachassagne et al., 2001; Misstear et al., 2009; Worthington et al., 2016). Ignorance of hydrogeological uncertainty may lead to a number of groundwater and environmental issues (Dewandel et al., 2004; Refsgaard et al., 2012; Lachassagne et al., 2021). Assessing the state of the geological layers for continuous groundwater assessments and reducing expenses without sacrificing efficacy are challenging tasks in groundwater research.

Groundwater resources are rapidly depleting over the world (Rodell et al., 2009; Wada et al., 2010; Laghari et al., 2012; Wada et al., 2014; Nguyen et al., 2022; Jasechko et al., 2024).

Therefore, it is critical to conduct an accurate and comprehensive evaluation of groundwater resources in order to manage and make use of these valuable reserves. Groundwater evaluations rely heavily on hydraulic characteristics. The most often adopted aquifer measure, hydraulic conductivity, is primarily world widely used to evaluate the rocks' capacity to hold water (Sale,

2001; Chandra et al., 2008; Camporese et al., 2011; Niwas and Celik et al., 2012; Fu et al., 2015;

Dewandel et al., 2017; Trinh et al., 2018; Ferris et al., 2020; Minutti et al., 2020; Asfahani, 2023;

Leal et al., 2023; Cui et al., 2024; Gao et al., 2024). Hydraulic conductivity is typically used to determine the aquifer potential of geological layers (Chen et al., 2001; Attwa et al., 2014; Bréard

Lanoix et al., 2020). Aquifer parameters are generally measured via borehole testing (De Lima and Niwas., 2000; Yao et al., 2013; Oli et al., 2022; Zoorabadi et al., 2022; Yang and Zhang,

2024). While boreholes do provide improved geological information, the process of creating a thorough 2D study is laborious and fraught with serious drawbacks (Hubbard and Rubin, 2002;

Niwas and De Lima, 2003; Gao et al., 2024). Borehole methods are costly and time-consuming, necessitate large apparatus or machinery, are challenging to execute in higher landscapes, only offer localized information, are unable to image lateral geological structures, and cannot assess the deep subsurface structures (Singh, 2005; Roques et al., 2018; Hasan et al., 2021). These limitations make it challenging to regularly conduct a sizable number of drilling experiments, which implies that a lack of borehole data could cause uncertainty in the evaluation of groundwater resources. Alternatively, to significantly reduce the number of expensive boreholes and precisely estimate the groundwater potential of the prospective rock masses, a cost-effective approach is needed.

Geophysical techniques were used in a number of previous groundwater investigations (da Silva et al., 2004; Porsani et al., 2005; Chambers et al., 2006; Yadav and Singh, 2007; Francese et al., 2009; Parks et al., 2011; An et al., 2012; Fu et al., 2013; Vouillamoz et al., 2014; Robinson et al., 2016; McLachlan et al., 2017; Lin et al., 2018; Kouadio et al., 2020; Abbas et al., 2022; Kouadio et al., 2023; Zhang et al., 2024). Geophysical practices are faster, easier to use, less expensive, and non-invasive than drilling techniques (Rashid et al., 2012; Loperte et al., 2016; Gao et al., 2024). They can also provide comprehensive vertical and horizontal geological evaluations (Cassidy et al., 2014; Soro et al., 2017; Hasan et al., 2021). When it comes to gathering hydrogeological data from diverse habitats below ground, these techniques are head and shoulders above the competition (An et al., 2012; Wynn et al., 2016; Kouadio et al., 2023). Nowadays, resistivity surveys are frequently carried out in various groundwater investigations. A significant advantage of resistivity methods over other geophysical methods is that they provide a wider resistivity range than other geophysical parameters (Niwas and De Lima, 2003; Bentley and Gharibi, 2004; Robinson et al., 2016). The principal resistivity methods include the vertical electoral soundings (VES), electrical resistivity tomography (ERT) technique, and the controlled source audio-frequency magnetotellurics (CSAMT) method (Soupios et al., 2007; Di et al., 2020; Gao et al., 2024). VES method was mostly used in previous groundwater-based geophysical studies to assess groundwater resources only in one dimension (Chandra et al., 2008; Majumdar and Das, 2011; Niwas and Celik, 2012; Nwosu et al., 2013; Attwa et al., 2014; McLachlan et al., 2017; Hasan et al., 2021; Asfahani , 2023). In hard rock terrains, it is rare to assess aquifer yield using two and three dimensional hydraulic characteristics at large depths. Recent research has shown that CSAMT is the best geophysical approach for studying hard rocks in terms of both cost and suitability that aim to collect comprehensive subsurface data at extremely deep depths via 2D/3D evaluations (Smith and Booker, 1991; Simpson and Bahr, 2005; Bai et al., 2010; An et al., 2012; Fu et al., 2013; Hu et al., 2013; Wang et al., 2015; Wynn et al., 2016; Di et al., 2020; Zhang et al., 2021; Kouadio et al., 2023; Hasan et al., 2024). When compared to other geophysical research methods, CSAMT has several advantages, including being more affordable, responsive to rocks with low resistance, and easier to use in difficult topographic situations (An et al., 2016; Kouadio et al., 2020; Zhang et al., 2021). With a depth capability of up to one kilometer, CSAMT provides more comprehensive subsurface assessments than the majority of geophysical methods, including ERT (Zonge and Hughes, 1988; Hasan et al., 2024). Thus, CSAMT is an effective instrument for investigating the vastly different topographical features, and it works better when applied with empirically based methods.

Geophysical and aquifer characteristics are determined by the same structural heterogeneities and several factors, including type of rock, fault, weathering degree, fluid content, permeability, pore-spacing, fracture, lithology, saturation, and joints (Purvance and Andricevic, 2000; Sinha et al., 2009; Sikandar and Christen, 2012; Hasan et al., 2021; Gao et al., 2024). For the hydrogeological characterization of subsurface rock mass units, a number of earlier researchers were able to successfully connect hydraulic data or lithological logs with geophysical parameters (De Lima and Niwas, 2000; Purvance and Andricevic, 2000; Chen et al., 2001; Sale, 2002; Hubbard and Rubin, 2002; Niwas and De Lima, 2003; Singh, 2005; Slater, 2007; Soupios et al., 2007; Chandra et al., 2008; Sinha et al., 2009; Majumdar and Das, 2011; Niwas and Celik, 2012; Sikandar and Christen, 2012; Nwosu et al., 2013; Attwa et al., 2014; Hasan et al., 2021; Oli et al., 2022; Rao et al., 2022; Asfahani, 2023; Gao et al., 2024). By establishing a useful connection involving electrical resistivity and the aquifer parameters (derived from drilling tests), resistivity methods can provide an alternate means of estimating hydraulic parameters. This study is groundbreaking because it uses non-invasive geophysical technique to generate two and three dimensional K models in a very varied environment with several types of rocks and considerable depths. The proposed research will need the drilling of only a small number of boreholes at strategic locations across the project area. We can then assess the vast research field with a more reliable CSAMT study. Then, even in the absence of drilling tests, K may be determined throughout the entire investigated site by directly correlating geophysical and borehole data. By applying the resultant equations to the entire research region, two and three dimensional K models are produced. This strategy would cut down on the pricey boreholes required to achieve a comprehensive and detailed assessment of subsurface hydrogeological conditions.

Before this work, no one had ever attempted to estimate K in a context as heterogeneous as this, with a wide variety of rock types present at a depth of 1 kilometer, using either direct or indirect approaches. Never before has a geophysical approach been employed in hard rock exploration to acquire volumetric measurements of 2D/3D K. In addition, no other study has ever used the CSAMT approach to derive any hydraulic parameter as this one has. We bridge the gap between reliable hydraulic models and limited borehole data with our more accurate 2D and 3D

K model estimates of complicated hydrogeological situations, outperforming previous investigations. We set out to do this study primarily to: (1) quickly predict two and three dimensional K models using geophysical methods; (2) accurately estimate the hydrogeological characteristics of rock masses for groundwater evaluations at great depths in difficult geological contexts; (3) reduce expensive boreholes and make the most efficient use of scarce drilling resources in order to collect hydrogeological data over large area; (4) reduce uncertainties in hydrogeological models and (4) encourage the use of non-invasive geophysical techniques for hard rock groundwater investigations as an alternative to expensive drilling.

**2. Study Area**

This study was carried out in the Jinji region of South China for deep groundwater exploration within a very diverse geological environment (Fig. 1). Precipitation at the study region is mostly concentrated in the summer due to its monsoon location; the annual precipitation totals 1965 mm. The Jinji region is surrounded by rivers and other bodies of water.

Low, somewhat cut, and significantly depleted hills and mountains characterize the geomorphology of the project site. The northern landscape is somewhat lower in elevation than its southern counterpart. The area is renowned for several things: a variety of terrain slopes, from mild to steep, lush vegetation, and weathered mountain rocks at an elevation of 43 to 438 meters above sea level (Yang et al., 2021). Mounts Dashishan, Qilongding, and Jixinshan are among the most notable. The southern portion of the study site is home to the summit of Xikeng, which stands at 549.8 meters and serves as the highest point of the landscape. The northeast section of the site under investigation is traversed by the Yongkouwei River, which flows through it at an elevation of around 7.5 meters. The study area has a variety of geological formations and periods, including the Jurassic, Permian, Carboniferous, Devonian, and even some Paleogene layers, as well as intrusive rocks from the Indosinian, Caledonian, and Yanshanian periods. Hornstone, granite, and sandstone are the main lithologies that have been found (Fig. 1a). Due to the influence of magmatic processes and different structures, the complex Kaiping concave fault and fold systems formed the main geological characteristics in the project area (Qin, 2017). With the local tectonic line coinciding with the faults strike, primarily in the northeast orientation, the emergence of joint fissured features represents the numerous tectono-geological phases (Yang et al., 2021). The location of the research area, which includes geophysical profiles, simplified geological conditions, and drilling tests, is shown in Fig. 1.

[Figure]

**Fig. 1.** Project site location with a simplified geological background of granite, hornstone, and sandstone, six CSAMT profiles (black lines), and six boreholes (blue circles), including three potential aquifers namely high potential aquifer (HPA), medium potential aquifer (MPA), and low-negligible potential aquifer (L-NPA)

**3. Methods**

The current work attempted to estimate K for two and three dimensional assessment of groundwater resources throughout the project territory using available borehole data in conjunction with a non-invasive CSAMT approach. Fig. 2 is a flowchart that summarizes the main steps of this approach.

[Figure]

**Fig. 2.** Flowchart providing an overview of the proposed approach for obtaining 2D and 3D K models for more precise and comprehensive evaluations of groundwater resources across extensive areas

*3.1. CSAMT survey*

The use of CSAMT in hard rock research is widespread (Simpson and Bahr, 2005; Bai et al., 2010; An et al., 2012; Fu et al., 2013; Wang et al., 2015; Wynn et al., 2016; Di et al., 2020; Zhang et al., 2021; Kouadio et al., 2023; Hasan et al., 2024). Such investigations involve the controlled electric signals that are sent into the earth from the transmitter site, which is located distant from the receiver, and the receiving station monitors the electric and magnetic fields (Zonge and Hughes, 1988; Zhang et al., 2021). In a subsurface structure where different fields have different propagation depths, a mathematical relationship exists between the reflected depth and the frequency (Borah and Patro, 2019). Using the fact that different rocks have different electrical conductivities, it monitors changes in the main field potential and magnetic field strength (Cagniard, 1953; Zonge and Hughes, 1988). The frequency components of the signal are obtained by applying Fourier transforms once the time series of the EM field variations has been grabbed (Simpson and Bahr, 2005). When doing CSAMT, an artificially controlled field source is used. It is possible to measure the electromagnetic field component of an electric dipole source with electrodes placed 1–2 kilometers apart. We can place the transmitter and any necessary connections between the batteries and the current electrodes. Field source transmitter-receiver distances are typically 5–10 km, depending on geological conditions and DOI (depth of investigation). One way to determine the resistivity of the subsurface is to divide the electric and magnetic field magnitudes by two orthogonal directions. A number of factors influence the subsurface geology-related resistivity, including fault fragmentation, water saturation, lithological changes in stratigraphic structures, pore fluid, porosity, and rock kinds (Fu et al., 2013; Zhang et al., 2021; Hasan et al., 2024). From 20 to 1000 meters below the surface, CSAMT can evaluate geological features with a vertical resolution of 5–20%. DOI is based on the subsurface resistivity and propagation frequency. Higher DOI is usually caused by greater resistivity and reduced frequency (Borah and Patro, 2019). The lateral resolution is contingent upon the station spacing, which generally ranges from 10 to 200 meters. A stronger received signal is achieved with larger station-spacing sizes (Simpson and Bahr, 2005). Use of a portable receiver allows for signal processing, amplification, filtering, and recording at each station. Electrode pairs, which consist of short grounded dipoles and magnetic-field sensors, are used to detect transmitted signals. The influence of radio transmitters, metal fences, power lines, and other obstacles on the accuracy of CSAMT data can be mitigated through well-planned surveys.

Possible representations of modeled resistivity data include plan, 3D, fence, and cross-sectional views.

Along six profiles (XKWT1–XKWT6), a total of 122 sounding stations and 5,825 meters of profile length were utilized to acquire CSAMT data. Every station was 50 meters away from the others. The DOI in the CSAMT survey was 1300 meters. Scalar measurements were taken using the TM Mode. These measurements take the electric and magnetic fields in two directions:

parallel to the measuring line and perpendicular to it. The measuring stations must be 50 m distant from the electrode and linked consecutively when conducting EMAP observations. Our setup included a 50 Hz linear filter and Gain mode X1. For 1 Hz, the emission current peaks between 12 and 18 A, while for 7680 Hz, it drops to between 2.6 and 4.5 A. For the purpose of gathering CSAMT data, a Phoenix, Canada-made V8 multifunction receiver with TXU-30

transmitter was employed. An exclusively geophysical approach with transmission voltages up to

1000 V, currents up to 20 A at 1000 V, and currents up to 40 A at 500 V can all be supported by the TXU-30 multi-function transmitter with a 30 kilowatt output. Deep exploration is a natural fit for this GPS-enabled transmitter because it works with standard domestic three-phase 220 volt alternators. A working frequency range of 1–7680 Hz was employed over 34 distinct frequency points. The V8 multifunction receiver can do more than just gather data; it can also keep tabs on the data from other secondary receiving units. In order to accomplish this, the primary receiver features three channels and three tracks. Transmitter and receiver distances varied between 9.3

and 12.5 kilometers. In order to capture the electric field signals, the non-polarized electrode was employed. The AMTC-30 inductive sensor received the signals, which operates between 10,000

Hz and 0.1 Hz and is designed for high-frequency AMT/CSAMT magnets. A tensor measurement was carried out at each site following the acquisition of two orthogonal electric field components and three orthogonal magnetic field components. In this case, the data came from the US firm Trimble's GPS receiver (XH dual-frequency). We quantified the CSAMT lines of objects recognition with the help of the Hi-Tech V30GNSS RTK equipment. These days, with the use of GPS, pinpoint accuracy can reach sub-meter levels. Using the specified direction and distance, the computer determined and transmitted the coordinate values of every survey line and survey point to the GPS or RTK. Using either the RTK or GPS navigational capability, the survey lines' measuring points were located. Testing the points along the measurement lines for system quality within a 3–5% range revealed a pretty uniform distribution of inspection points.

The following design requirements were met by the results of the system quality check: an RMS

value below ±5%, an error tolerance of adjacent points on the profile of less than 10, a relative elevation tolerance of 1.67mm, and a plane tolerance of 2.33mm. The data acquired was of exceptional quality because there was no human or electrical interference at the project location.

By analyzing the CSAMT data, we were able to determine the site's features (An et al., 2012;

Hasan et al., 2024). After the skewed data was removed, a curve analysis was carried out.

Geological information and curve analysis were used to make the static corrections, which were done using a Hanning window spatial filtering method. So, having high-quality geophysical data made it easier to get correct data processing and interpretation.

     The CMTPro Version software developed by Canadian Phoenix was employed for the data processing phase (Phoenix Geophysics CMTPro, 2020). Bad measuring point curves are removed, electrode coordinates are corrected, observed curves are automatically smoothed, CMT

files are compiled from source current, reference track data, and V8 data, and files in the AVG

format are generated, among other things, by this software. The 2D inversion (Rodi and Mackie,

2001; Wang et al., 2015) was executed using the CSAMT-SW algorithm, as shown in Fig. 3, which is a flow diagram of the algorithm (Phoenix Geophysics CSAMT-SW, 2020). These are the CSAMT-SW's key parts: 1. An AVG file to D format data conversion; 2. Changing and producing new CHK elevation files as well as importing existing ones into D files; 3. Inspecting the D file by hand for damaged sectors, interpolating them, erasing near-field data, and jumping to certain locations; 4. Various static correction approaches yielded inversion results that were nearly identical when compared; the D file was used for smoothing processing; 5. The four results (D files) of static correction for various correction processes are D, H, K, and Z; 6.

Employing BOSTICK inversion and near-field correction to convert to text files; 7. By directly applying the CSAMT global field model (ID), which combines transition and near fields, to the measured data from CSAMT, quasi-2D inversion can be used to create finite (limited) layers with data representing resistivity in depth sections. For data in D file, we applied Bostick inversion (Fusheng et al., 2022) and saved the output as * _BOS.DAT and * _BSS.DAT, respectively. In addition, a newly produced * _M. DMT text file is used to hold the transformed data formatted in D files, in compliance with the requirements of the 2D inversion model of

CSAMT. We utilize the inversion method to fit the derived models with the observed measurements after we reach the maximum number of iterations or the maximum RMS error, which in this study are 5 iterations. By utilizing the most suitable processing and inversion algorithms, the errors in the models were minimized, and a reliable 2D resistivity model (Zhang et al., 2021) of CSAMT was generated, taking into account the local geology and dataset peculiarities. Our understanding of the subsurface geological features was enhanced by the final inversion model, which demonstrated changes in resistivity.

[Figure]

Data: * .AVG

Data preprocess

Log files of the measurement lines (D Files):

Coordinate deviation correction

Removal of bad data

Curve smoothing

Remove near-field frequency points

Remove bad channel (duplicate data)

Spectrum curve check

Bostick inversion
Generating * .DMT file

Pseudo 2D inversion
(CSAMT model)

Construct cross-sections

**Fig. 3.** Displaying the procedure of 2D inversion of CSAMT data by the use of Bostick inversion

*3.2. Estimation of hydraulic conductivity (K)*

In groundwater studies, hydraulic conductivity (K) is frequently employed to estimate the amount of water that can be extracted from underground aquifers (Chandra et al., 2008;

Camporese et al., 2011; Niwas and Celik et al., 2012; Fu et al., 2015; Dewandel et al., 2017;

Trinh et al., 2018; Minutti et al., 2020; Leal et al., 2023; Cui et al., 2024). Traditionally, costly borehole experiments are used to determine hydraulic conductivity, a crucial aquifer characteristic (Niwas and Celik et al., 2012; Hasan et al., 2021; Gao et al., 2024). Water's ability to flow easily through rock mass's pore spaces or cracks is known as hydraulic conductivity. A

number of elements influence hydraulic conductivity (K), such as the amount of fluid present, defects, saturation level, rock composition, faults, compaction, deformation, and joint and cracks (Purvance and Andricevic, 2000; Sinha et al., 2009; Sikandar and Christen, 2012; Hasan et al.,

2021).

There is strong evidence of a correlation between geophysical and hydrological characteristics, according to numerous researchers (Sale, 2002; Hubbard and Rubin, 2002; Niwas and De Lima, 2003; Slater, 2007; Soupios et al., 2007; Sinha et al., 2009; Majumdar and Das,

2011; Sikandar and Christen, 2012; Attwa et al., 2014; Oli et al., 2022; Rao et al., 2022;

Asfahani, 2023; Gao et al., 2024). The process of these correlations begins with the determination of hydraulic conductivity from drilling data at specific point-locations. Then, the empirical equations are obtained by integrating the hydraulic conductivity from boreholes with electrical resistivity from geophysical data. Next, hydraulic conductivity for the entire researched site is determined by applying all resistivity values from six profiles to the resulting equation.

This allows for a determination of hydraulic conductivity over the entire site, even in cases when a borehole cannot be accessed. Previous empirical geophysical methods mostly utilized VES

(vertical electrical sounding) for 1D prediction of hydraulic conductivity, predominantly in homogeneous environments at shallow depths. Consequently, deep groundwater investigation using K had not been previously conducted, particularly in hard rock formations. Consequently, our recent research first time ever employed a CSAMT-based empirical approach to estimate two and three dimensional K across an extensive area characterized by varied rock formations at large depths.

In the first step, 37 measurements from six boreholes (ZK1, ZK2, ZK3, ZK4, ZK5, and

ZK6) were collected at various depths between 10 and 200 meters (Fig. 4a). 6 K values (2.6, 1.8,

0.91, 0.82, 0.64, and 0.41 m/d) were derived from ZK1 at depths of 25, 65, 115, 140, 170, and

200 m. At depths of 10, 35, 60, 100, 130, 165, 185, and 200 meters, eight K values (i.e., 1.5, 0.95,

0.85, 0.76, 0.7, 0.47, 0.38, and 0.24 m/d) were derived from ZK2. Six K values (i.e., 0.96, 0.9,

0.77, 0.6, 0.3, and 0.05 m/d) were derived from ZK3 at depths of 10, 40, 85, 120, 150, and 200 m.

At depths of 25, 85, 150, and 200 meters, four K values (i.e., 22, 14, 2.2, and 0.99 m/d) were derived from ZK4. Seven K values (i.e., 0.97, 0.93, 0.79, 0.72, 0.61, 0.5, and 0.39 m/d) were derived from ZK5 at depths of 10, 30, 60, 100, 135, 175, and 200 m. And, six K values (i.e., 15, 10, 5.5, 2.18, 0.96, and 0.88 m/d) were derived from ZK6 at depths of 10, 45, 80, 120, 150, and 200 m. The second stage involves: 37 CSAMT-derived observations (i.e., 6 values from $5^{th}$ sounding XKWT1-5 along CSMAT profile XKWT1 at 200 m distance: 486, 582, 782, 915, 1080, and 1472 Ωm; 8 values from $8^{th}$ sounding XKWT3-8 along geophysical profile XKWT3 at 350 m distance: 610, 756, 863, 973, 1035, 1354, 1490, and 1775 Ωm; 6 values from $2^{nd}$ sounding XKWT4-2 along surveyed line XKWT4 at 50 m distance: 735, 802, 942, 1186, 1661, and 2654 Ωm; 4 values from $15^{th}$ sounding XKWT4-15 along CSAMT profile XKWT4 at 700 m distance: 75, 163, 537, and 710 Ωm; 7 resistivity values from $15^{th}$ sounding XKWT5-15 along geophysical line XKWT5 at 700 m distance: 716, 792, 879, 1021, 1157, 1310, and 1490 Ωm; 6 values from $21^{st}$ sounding XKWT5-21 along profile XKWT5 at 1000 m distance: 142, 223, 326, 535, 721, and 821 Ωm) in line with the measured K (obtained from drill tests) were acquired at the aforementioned depth (Fig. 4a). In the third stage, the empirical integration of the picked observations (37 data sets) of CSAMT-based resistivity and borehole-based K was used to derive the following equation (Fig. 4b):

$$K = 47194(\rho)^{-1.617} \tag{1}$$

where K is the hydraulic conductivity, measured in m/d units, and ρ, represented in Ωm, denotes the true or inverted resistivity. Eq. (1) was utilized to predict the hydraulic conductivity (K) over entire area using an extensive resistivity data from six geophysical surveyed lines. By this way, we were able to estimate the water-retaining ability of three rock types (granite, hornstone, and sandstone) for a comprehensive evaluation of groundwater resources across three potential aquifers (low-negligible potential aquifer (L-NPA), medium potential aquifer (MPA), and high potential aquifer (HPA)) throughout the entire study area with 0–1300 m depth. Lastly, the K

parameter, a predicted hydrogeological feature that stretches throughout all XKWT1–XKWT6

geophysical profiles, was modeled in two and three dimensions using the Geosoft and SKUA-

GOCAD software programs (Webring, 1981; Mira Geoscience Ltd, 1999; Hasan et al., 2024).

[Figure]

**Fig. 4. (a)** The presentation includes 37 resistivity-K data points at depths ranging from 10 to 200 m, derived from 6 drilled tests (ZK1–ZK6) and corresponding resistivity (ρ) measurements from CSAMT soundings. The soundings are identified as follows: XKWT1-5 for sounding number 5 along profile XKWT1, XKWT3-8 for sounding 8 along profile XKWT3, XKWT4-2 for sounding 2 along profile

XKWT4, XKWT4-15 for sounding 15 along profile XKWT4, XKWT5-15 for sounding 15 along profile

XKWT5, and XKWT5-21 for sounding number 21 along profile XKWT5. Different rocks include hornstone (HS), sandstone (SS), and granite (G); **(b)** Using a total of 37 data points, the geophysical- borehole correlation for the predicted K

**4. Results**

*4.1. Geophysical-borehole correlation*

The combined data from six CSAMT profiles and six boreholes, which were used to divide the subterranean formation into several different geological strata according to the various ranges of hydraulic conductivity (K) and electrical resistivity, is displayed in Table 1. The subsurface hydrogeological models were constructed using data from the CSAMT-based resistivity and borehole-based K and the geological settings of the study area. These models include three distinct geological layers, namely hornstone, sandstone, and granite. The following conditions were taken into account when evaluating sandstone, hornstone, and granite: sandstone with a resistivity below 350 Ωm and a K range of 5 to 25 m/d, hornstone with a resistivity that goes from 350 to 700 Ωm and K ranges from 1 to 5 m/d, and granite with a resistivity over 700

Ωm and a K range of 0 to 1 m/d. We rated the different aquifer potential zones in the subsurface hydrogeological model as follows: sandstone has a high potential aquifer (HPA), hornstone contains a medium potential aquifer (MPA), and granite includes a low to negligible potential aquifer (L-NPA). Sandstone indicates aquifers with the highest yields or rock masses with the best water-bearing capacities, while granite denotes rock masses with the lowest yields or rock masses with the worst water-bearing capacities. Accordingly, in the research region, sandstone presents the most favorable conditions for developing groundwater, while granite presents the worst cases for groundwater extraction.

**Table 1**

Using the distinct value ranges of electrical resistivity and hydraulic conductivity (K) to integrate them for a thorough groundwater assessment in hard rock of various types

| Resistivity (Ωm) | K (m/d) | Type of rock | Aquifer potential |
| --- | --- | --- | --- |
| < 350 | 5–25 | Sandstone | High potential aquifer (HPA) |
| 350–700 | 1–5 | Hornstone | Medium potential aquifer (MPA) |
| >700 | 0–1 | Granite | Low-negligible potential aquifer (L-NPA) |

*4.2. 2D groundwater assessment*

Eq. (1), which is based on geophysical-borehole correlation (Fig. 4), efficiently transforms two-dimensional CSAMT models into two-dimensional K models and displays the results in Fig. 5. Using geophysical-based 2D K models, we can precisely and thoroughly assess the groundwater resources in hard rock across the complete study area, 0–1300 meters deep, in comparison to the limited drill tests (Fig. 6 and 7). For XKWT1 surveyed line, the following geological layers have been delineated for groundwater assessment: A high potential aquifer contained in sandstone was assessed at a distance of 240–380 meters and between 200 and 850 depths. The medium potential aquifer hold by hornstone was assessed for 0–240 m distance within 0–395 m and 800–1300 m depth, for 240–530 m apart and 0–1300 m deep, for 1200–1300 m apart and 0–1300 m deep, for 1200–1450 m distance within 0–400 m and 800–1300 m depths. Granite-related low potential aquifers were identified by distances of 0–290 m and depths of 300–800 m, 380–1220 m and depths of 0–1300 m, and 1300–1450 m and depths of 400–750 m. The following is a description of the geological layers used for groundwater assessment along profile XKWT2: No sandstone with significant aquifer potential was assessed along this profile.

Hornstone of moderate potential aquifer was defined in depths of 0–300 m and 850–1300 m within a distance of 200–450 m, at 0–200 m depth within 0–50 m distance, and at 800–1300 m depth within 0–200 m distance. This profile is predominantly characterized by granite of low potential yield, extending from 0 to 1300 meters in depth and 0 to 450 meters in distance. The characterization of geological strata for groundwater evaluation along profile XKWT3 is as follows: We looked at a high-potential groundwater in sandstone at distances of 0–150 m and depths of 0–250 m, 60–100 m and depths of 775–825 m, 450–510 m and depths of 0–170 m, and 500–540 m and depths of 700–715 m. Hornstone with a medium possible yield was defined by profile lines that went from 0 to 540 meters apart and from 0 to 390 meters deep; from 650 to 950 meters apart and from 1100 to 1300 meters deep; and from 350 to 650 meters apart and from 700 to 1300 meters deep. One layer of granite rock with a depth of 0–800 meters at a distance of 0–650 meters, and another layer with a depth of 850–1250 meters at a distance of 0–400 meters, is used to assess the low potential aquifer. Profile XKWT4's geological layer delineation for groundwater assessment is as follows: High aquifer yield sandstone was assessed at a depth of 0–280 meters and a distance of 650–700 meters. Medium aquifer yield hornstone was defined at 600–900 m depth and 0–150 m distance, 430–500 m distance and 0–150 m depth, and 400–650 distance and 500–1300 m depth. One granite layer with a depth of 0–850 meters at a distance of 0–700 meters and another granite layer with a depth of 900–1300 meters at a distance of 0–400 meters are used to assess the low aquifer yield found within granite. The following is a description of the geological layers used for groundwater assessment along profile XKWT5: High potential sandstone was analyzed at distances of 0–190 m and depths of 0–1200 m, 910–1060 m and depths of 0–200 m, and 1065–1400 m and depths of 500–1300 m. Medium yield hornstone was primarily identified by distances of 175–380 m and depths of 650–1300 m, 390-

800 m and depths of 600–1000 m, 800–1400 m and depths of 0–400 m, and 800–1050 m and depths of 400–1300 m. Granite of the low-yield aquifer is assessed for distances of 90–1010 m and depths of 0–600 m, 1000–1300 m depth and 400–850 m distance, as well as 1100–1300 m distance and 200–500 m depth. The geological layers for groundwater assessment along profile

XKWT6 are delineated as follows: No sandstone has been assessed along this profile. Hornstone of medium potential aquifer is evaluated at 0–100 m distance between 0–350 m depth, 350–500

m distance for 0–360 m depth, 345–655 m distance between 450–850 m depth, 400–520 m distance from 1100 to 1300 m depth, 590–800 m distance with 0–280 m depth, and 980–1150

distance for 0–360 m depth. Low-yield granite predominantly characterizes this profile at depths of 0–1300 m and distances of 0–1150 m. The findings from the integrated 2D K models illustrated in Fig. 6 and 7 point out that the aquifer-bearing capacity of geological rock units generally rises with depth, predominantly exhibiting medium to high potential aquifers in the eastern, western, and partially southern regions, while the central areas exhibit the least or worst occurrence of groundwater resources.

[Figure]

**Fig. 5.** The conversion of 2D CSAMT models (for six profiles XKWT1–XKWT6) into 2D K models and the interpretation of these 2D K models, using geophysical-borehole correlation, enable groundwater evaluation through high potential aquifer (HPA), medium potential aquifer (MPA), and low-negligible potential aquifer (L-NPA) connected to sandstone (SS), hornstone (HS), and granite (G), respectively

[Figure]

**Fig. 6.** The integrated 2D K models obtained from geophysical-drilling incorporation (K indicated on a color bar ranging from blue to grey) for **(a)** 0–200 m depth, **(b)** 0–600 m depth, **(c)** 0–1000 m depth, and

**(d)** 0–1300 m depth

[Figure]

**Fig. 7** Interpretation of 2D K models (obtained from specific K ranges) for three groundwater potential aquifers: low-negligible potential aquifer (L-NPA); medium potential aquifer (MPA); high potential aquifer (HPA), linked to three geological strata: granite (G), hornstone (HS), and sandstone (SS), respectively, at depths of **(a)** 0–200 m, **(b)** 0–600 m, **(c)** 0–1000 m, and **(d)** 0–1300 m

*4.3. 3D groundwater assessment*

A comprehensive evaluation of the water-bearing capacity of rock mass for groundwater assessment was accomplished by the 3D K external visualization illustrated in Fig. 8 (a, b). Granite of low potential aquifer was assessed at the ground surface along profile XKWT1 at distances of 400–500 m and 700–1100 m, along profile XKWT2 at 50–200 m, along profile XKWT3 at 0–30 m, 200–250 m, 350–450 m, and 600–650 m, along profile XKWT4 at distances of 0–420 m and 500–600 m, along profile XKWT5 at distances of 120–210 m, 290–815 m, and 1150–1300 m, and along profile XKWT6 at distances of 100–500 m and 800–1000 m. Medium aquifer yield within hornstone was identified along profile XKWT1 at distances of 0–400 m, 500–695 m, and 1100–1450 m; XKWT2 at 0–50 m and 200–450 m; XKWT3 at 30–220 m, 250–350 m, 400–450 m, and 500–550 m; along profile XKWT4 at 420–500 m and 600–650 m; along profile XKWT5 at 0–120 m, 200–220 m, 260–280 m, 590–610 m, 800–910 m, 1060–1150 m, and 1300–1400 m; and along profile XKWT6 at 0–100 m, 500–800 m, and 1000–1150 m. The sandstone with significant aquifer potential was evaluated in various locations, including profile XKWT1 at 250–300 m, profile XKWT3 at 430–500 m, profile XKWT4 at 650–700 m, profile XKWT5 at both 250–300 m and 900–980 m, and profile XKWT6 at 0–30 m. A low-potential aquifer of granite predominates on the surface, especially in the center. The remaining regions are assessed based on the medium potential aquifer of hornstone. Sandstone with high aquifer production is assessed in limited areas encircled by hornstone

Fig. 8 (c, d) provides a thorough analysis of the rock mass's water-bearing capacity for groundwater assessment using a 3D K internal viewpoint. At a subterranean depth of 1300 m, a low aquifer yield of granite was evaluated using profile XKWT1 over a distance of 550–1180 m, profile XKWT2 over 0–100 m, profile XKWT3 over 0–110 m, profile XKWT4 over 0–400 m, profile XKWT5 over 400–800 m, and profile XKWT6 over distances of 0–400 m and 550–1150

m. The medium aquifer yield within hornstone was delineated by profile XKWT1 at distances of

0–550 m and 1200–1450 m, profile XKWT2 at 100–450 m, profile XKWT3 at 70–650 m, profile

XKWT4 at 350–705 m, surveyed line XKWT5 at 200–405 m and 850–1050 m, and XKWT6 at

400–550 m. The high-potential aquifer contained within sandstone was assessed exclusively along profile XKWT5 at distances 0–200 m and 1000–1400 m. Medium to high potential aquifers, at 1300 m depth, predominantly occupy the eastern and western regions, while the center portions are primarily characterized by low potential aquifers. Fig. 8 presents the findings of the 3D K analysis, revealing that the interior predominantly consists of granite with minimal aquifer yield. The water-retaining capacity of the rock mass increases when viewed from above.

This facilitates an accurate appraisal of the water-bearing capacity of geological layers for comprehensive groundwater evaluation using 3D K modeling.

[Figure]

**Fig. 8.** The 3D K models derived from the correlation of CSAMT and borehole data (with K displayed on a color scale transitioning from blue to grey) for three groundwater potential aquifers: low-negligible potential aquifer (L-NPA); medium potential aquifer (MPA); high potential aquifer (HPA), linked to three geological strata: granite (G), hornstone (HS), and sandstone (SS), respectively, for **(a)** The outside view of 3D K model, **(b)** Analysis of the 3D (external perspective) K model, **(c)** The inner outlook of 3D K model, and **(d)** Analysis of the 3D (internal perspective) K model

*4.4. Groundwater assessment via depths and 2D/3D insights*

Given the limited data collected from boreholes, the water-bearing capacity of rock masses below 200 m depth cannot be evaluated using the observed K (borehole-based K). An efficient, precise, and comprehensive evaluation of hard rock groundwater resources was achieved by establishing a strong correlation between drilling and CSAMT data. This allowed for the determination of K up to depths of 1300 m, while also saving time. As shown in Fig. 9, predicted K values at depths of 0, 300, 600, 900, and 1300 m were obtained by 2D/3D

groundwater yield insights. The following were the criteria for evaluating groundwater at a depth of 1300 meters: Granite, which makes up almost half of the subsoil in areas with low potential aquifer, is evaluated in the southwest and central regions. Hornstone, which made up 28% of the medium potential aquifer, was investigated in the western and eastern parts around the granite formation. Nearly a quarter of the subsurface assessments in the eastern region were carried out on high-yield sandstone. The following criteria were utilized to gain an understanding of the subsurface for groundwater evaluation at a depth of 900 m: Sandstone made up 21% of the high- potential-aquifer subsoil in the eastern areas. The eastern and western regions revealed 27%

hornstone of a medium potential aquifer surrounding granite. The subsurface, which had a low aquifer yield, was 52% granite, with boundaries in the center, north, and southwest. At 600

meters below the surface, we used these criteria to interpret the hydrogeological conditions: In the central, northern, and western areas, 55% of the subsurface was found to be a low-yielding aquifer of granite; in the southwestern and eastern areas, hornstone was more prevalent, making up 25% of the subsurface and suggesting a medium-yielding aquifer; and in the eastern regions, sandstone was mostly studied, making up 20% of the subsoil and suggesting a high-yielding aquifer. These criteria were utilized to analyze the hydrogeological conditions at a depth of 300

meters: In the central and northwest sections, granite with a low potential aquifer made up 63%

of the total. In the southern and southeastern regions, hornstone with a medium yield aquifer made up 27% of the underground. In the southwestern and eastern areas, sandstone with a high potential aquifer, accounting for 10% of the subsoil, was studied. The hydrogeological conditions shown below were ascertained from the surface at a depth of 0 m: In the northern and central regions, 65% of the subsurface is composed of granite, while 26% of the surface is hornstone and contains a medium-potential aquifer. The sandstone, which is mostly found in the inner and southwestern regions, is examined on 9% of the surface and contains a high-potential aquifer. As seen in Fig. 9, the thickness of low yield aquifer granite diminishes as one move downwards. The conditions for the occurrence of groundwater are worst in the middle parts, dropping down to 600 or 700 m. Rock masses with significant potential aquifer, particularly below 700 m depth, are located in the western and eastern regions.

[Figure]

**Fig. 9**. **(a)** In 2D and 3D perspectives of groundwater occurrence, K on a color bar that increases from blue to grey represents geophysical-based K imaging at different depths (0, 300, 600, 900, and 1300 m),

**(b)** Analysis of geophysical-derived K (utilizing specific K ranges) at varying depths via 2D and 3D

insights for fresh granite (G) of low-negligible potential aquifer (L-NPA), hornstone (HS) of medium potential aquifer (MPA), and sandstone (SS) of high potential aquifer (HPA)

*4.5. Comparison of the predicted and measured K*

The K derived using CSAMT provides an accurate and methodical assessment of water- bearing yield for groundwater evaluation throughout the project area. The K results (Fig. 5–9)

indicate that granite is analyzed in the central parts; hornstone is largely identified between granite and sandstone in the eastern, southern, and western regions. In the eastern regions, sandstone is thoroughly analyzed, whereas in the western sections, it is partially appraised. Data collected from boreholes produce incompatible mapping of subsurface geological layers, which makes it difficult to assess the water-bearing capacity of rock masses. The drilling results do coincide with the CSAMT K in a small number of locations at depths of 200 meters near the drills. Consequently, groundwater potential assessments across large regions are rendered imprecise by measured K (obtained via drills), in comparison to the predicted K.

Table 2 shows the %matching for the selected measurements, which were determined by comparing the drill-K with the CSAMT-K. For a few chosen data points, we compared the predicted K with the borehole-based K and found the following percentage matching: Applying

Eq. (1) for depths of 25 m and 115 m, respectively, yields a %matching of 81 and 92 when ZK1

(well number one) is empirically coupled to XKWT1-5 (5th sounding along CSAMT profile

XKWT1). A percentage matching of 100 and 90, at 10 and 130 m depth, respectively, are produced by the combination of XKWT3-8 (sounding number 8 of XKWT3) and ZK2 (well number two). The integration of ZK3 (well number three) with XKWT4-2 (second sounding on line XKWT4), at depths of 85 and 150 meters, results in matching percentages of 95 and 97, respectively. The amalgamation of ZK4 (well number four) and XKWT4-15 (15th sounding of

XKWT4) results in matching percentages of 67 and 86, at respective depths of 25 and 85 meters.

The integration of well number five (ZK5) and the fifteenth sounding at line XKWT5 (XKWT5-

15) produces %matching of 96 and 92, respectively, at depths of 60 and 200 m. In addition, a %matching of 75 and 84, with depths of 80 and 120 m respectively, are produced by combining ZK6 (well number six) and XKWT6-21 (21$^{st}$ sounding at XKWT6). A lesser degree of error or strong matching is shown by the aforementioned comparison between the obtained and predicted K. The comparison also shows that, even for data points with poor %matching, predicted and observed K values typically fall into the same aquifer potential zone.

**Table 2**

The percentage matching between the drill-K and the CSAMT-K for the selected measurements

| CSAMT data points (selected) | | | Drilling data | | | %Matching |
|---|---|---|---|---|---|---|
| CSAMT sounding number | Resistivity (Ωm) | Predicted K' (m/d) using Eq. (1) | Borehole name | Depth (m) | Measured K (m/d) | K' vs K |
| XKWT1-5 | 486 | 2.1 | ZK1 | 25 | 2.6 | 81 |
| XKWT1-5 | 782 | 0.99 | ZK1 | 115 | 0.91 | 92 |
| XKWT3-8 | 610 | 1.5 | ZK2 | 10 | 1.5 | 100 |
| XKWT3-8 | 1035 | 0.63 | ZK2 | 130 | 0.7 | 90 |
| XKWT4-2 | 942 | 0.73 | ZK3 | 85 | 0.77 | 95 |
| XKWT4-2 | 1661 | 0.29 | ZK3 | 150 | 0.3 | 97 |
| XKWT4-15 | 75 | 33 | ZK4 | 25 | 22 | 67 |
| XKWT4-15 | 163 | 12 | ZK4 | 85 | 14 | 86 |
| XKWT5-15 | 879 | 0.82 | ZK5 | 60 | 0.79 | 96 |
| XKWT5-15 | 1490 | 0.35 | ZK5 | 200 | 0.38 | 92 |

| XKWT5-21 | 326 | 4.1 | ZK6 | 80 | 5.5 | 75 |
| XKWT5-21 | 535 | 1.83 | ZK6 | 120 | 2.18 | 84 |

**5. Discussion**

The application of geophysical technologies is increasingly prevalent in groundwater research. Previous investigations indicated that groundwater evaluation could benefit from integrating geophysical and drilling data. The water-bearing potential of a rock mass can be estimated using a number of hydraulic characteristics. Hydraulic conductivity (K), typically evaluated via boreholes, is the most reliable and practical hydraulic parameter utilized in groundwater assessments. This study is the first to employ geophysical approaches to indirectly acquire 2D/3D K at depths greater than 1 km in a context with a wide diversity of rocks.

In this paper, we present CSAMT, a new geophysical approach for more accurate groundwater evaluation in the lack of adequate borehole data, by evaluating the water-bearing capacity of rock masses. It opens the door to a thorough evaluation of deep hard rock groundwater using the predicted hydraulic parameter K. Based on the diverse topography of South China, our methodology offers a flexible empirical correlation using its huge geophysical dataset and sparse borehole data. The rocks and lithologies were classified according to a specific set of K values used in the aquifer models. The rocks are categorized into three groups according to distinct aquifer potential zones: fresh granite with a low-negligible potential aquifer (L-NPA), sandstone with a high potential aquifer (HPA), and hornstone with a medium potential aquifer (MPA) between the two potential aquifers. These computations are applicable in such specific geological conditions to determine the total water-bearing capacity of the rock formation, as they are based in resistivity-K measurements of hornstone, granite, and sandstone. Depending on the local environment and the composition of the rock, the exact parameter ranges are determined. Based on the area hydrogeological circumstances, the exact rock mass class of a potential aquifer can be calculated using the well-established flexible equations. The proposed method allows for the derivation of generalized equations that are applicable in any geological setting. The K-resistivity range of a rock unit is relative and might vary from one location to another. In most cases, a solid empirical equation can be derived by drilling five or more boreholes across the entire area, each of which should have at least five measurements taken from the rock unit. Both the quantity of geophysical-borehole observations and the range of K- resistivity have a significant impact on the validity of the empirical equation. By incorporating more datasets into correlation analysis, K can be more accurately computed. According to Table

2, the majority of the datasets demonstrated an impressive level of accuracy with matching rates exceeding 80% between the actual and estimated K. Particularly for extremely low resistivity and high K values, the established equation gives poor matching between the predicted and observed K. For example, at a depth of 25 meters, there was only a 67% match between ZK4 and

XKWT4-15; nonetheless, the calculated and predicted K values are in the same HPA zone.

Throughout the project region, the resistivity-K ranges utilized for correlation analysis encompass all lithologies and rock types, including granite, hornstone, and sandstone. The positions of the boreholes may have been indicative of the rock unit features of the entire research site, which allowed for the reliable results to be acquired. Instead of applying one formula for evaluation of different geological layers, it might be more precise to utilize separate equations for each type of rock unit to determine K. Distinct equations, however, might be more effective while enough borehole data is available for each rock mass unit. The positions of surveyed lines are essential for the accuracy of the calculated K, as the anticipated K is derived from this correlation. Because of this, two and three dimensional K models produce somewhat more accurate results in areas close to geophysical profiles compared to areas far from these profiles. When data from boreholes is not available, the resulting equation can be used to estimate K in similar geological settings.

CSAMT is widely used for investigating underground structures and to a considerable extent for mitigating the impact of weak natural signals. Nevertheless, resistivity measurements can be impacted by various elements, such as transmission devices, electrical lines, metal obstacles, etc., which can lead to ambiguous results and interpretations. This study shows, however, that with good CSAMT survey design, these effects can be reduced and reliable results can be obtained. The absence of electrical or human disturbance at the project site allowed for the collection of high-quality data. A K value of 30 m/d, which is equivalent to a resistivity of 27

$\Omega$m, indicated that the rock mass (sandstone) could hold the most water. Nonetheless, when the resistivity value was measured at 5000 $\Omega$m, the rock mass (granite) was determined to have a minimum water-bearing capability of 0.05 m/d. In contrast to the drilling K, the geophysical K

assesses the rock mass's water-bearing capacity more accurately and thoroughly while reducing variability in the expected hydrogeological model. As a result, hydrogeological models for groundwater assessment in extremely diverse hard rock are seriously doubted due to insufficient boring trials. However, geophysical techniques help bridge the gap between accurate hydrogeological models and inadequate borehole data.

**6. Conclusions**

Our research introduces novel approaches for employing non-invasive technology in deep groundwater studies. This research, first time ever, utilizes CSAMT to indirectly estimate two and three dimensional K values, evaluating the water-bearing capacity of rock masses for deep groundwater assessment across extensive, heterogeneous hard rock regions without the need for drilling data. The predominant technique for assessing hydraulic parameters in groundwater research is the utilization of boreholes. Nonetheless, drilling techniques are expensive and possess significant drawbacks. This study provides a more comprehensive and accurate assessment of rock mass hydrogeological conditions than conventional techniques while requiring fewer boreholes. In order to evaluate the water-holding capacity of rock formations in different environments, we derived the flexible equation that determines K by analyzing

CSAMT-drilling data from numerous places. K was calculated using an established equation, allowing for a thorough evaluation of the water-bearing capacity of the rock mass for groundwater assessment throughout the whole research region. Sandstone of a high potential aquifer (HPA) was characterized in a three-layer hydrogeological model with a resistivity below

350 Ωm and a K range of 5 to 30 m/d. Using a resistivity increase of 350 to 700 Ωm and a K

range of 1 to 5 m/d, hornstone of a medium potential aquifer (MPA) was evaluated. Granite of the low-negligible potential aquifer (L-NPA) was evaluated using K values ranging from 0 to 1

m/d and a resistivity value higher than 700 Ωm. The results indicate that when the K parameter increases and resistivity lowers, the rock mass holds a greater amount of water. Deep groundwater resources in hard rock were anticipated to rely on the premise that the optimal rock mass for maximum water-bearing capacity would consist of sandstone, whereas the rock mass with minimal groundwater presence would be granite. According to our predicted 2D/3D K

models, deep groundwater extraction was conducted in central regions below 700 m depth and in adjacent areas around granite at depths ranging from 0 to 1300 m. A significant correlation between the local geology, hydrogeology, and the K models has been identified. Our research indicates that this method may serve as a more cost-effective alternative to expensive drills for acquiring more precise hydrogeological modeling maps, in contrast to traditional procedures. In groundwater investigations of hard rock, geophysical methods can bridge the gap between solid hydrogeological models and insufficient drilling data by efficiently and comprehensively assessing the water-retention capacity of the rock mass. Future study could enhance the explanation of aquifer parameters by refining empirical equations through the application of groundwater hydrogeological concepts. This technique would improve the understanding of the interaction between geophysical and aquifer parameters, hence augmenting its significance in groundwater applications.

**CRediT authorship contribution statement**

**Muhammad Hasan:** Data curation, Visualization, Validation, Supervision, Resources,

Software, Funding acquisition, Conceptualization, Investigation, Methodology, Formal analysis,

Project administration, Roles/Writing – original draft, Writing review and editing; **Lijun Su:**

Software, Funding acquisition, Conceptualization, Investigation; **Peng Cui:** Visualization,

Investigation, Validation; **Yanjun Shang:** Data curation, Software

**Declaration of competing interest**

The authors of this paper declare that they have no conflict of interest.

**Acknowledgements**

This research was financially supported by the National Natural Science Foundation of

China's Research Fund for International Young Scientists (RFIS-I) (Grant No. 42350410442), and International Science and Technology Cooperation Program of Shanghai Cooperation

Organization, Science and Technology Department, Xinjiang, China (Grant No. E202301005).

The authors wish to acknowledge the institutions that facilitated the research for this study: the

State Key Laboratory of Mountain Hazards and Engineering Resilience, Institute of Mountain

Hazards and Environment, Chinese Academy of Sciences, and China-Pakistan Joint Research

Center on Earth Sciences, CAS-HEC, Islamabad, Pakistan.

**Data availability**

Data available on request from the corresponding author

**References**

1. Abbas, M., Deparis, J., Isch, A., Mallet, C., Jodry, C., Azaroual, M., Abbar, B., Baltassat,

J.M., 2022. Hydrogeophysical characterization and determination of petrophysical parameters by integrating geophysical and hydrogeological data at the limestone vadose zone of the Beauce aquifer. Journal of Hydrology 615, 128725.

2. Amiotte Suchet, P., Probst, J.L., Ludwig, W., 2003. Worldwide distribution of continental rock lithology: Implications for the atmospheric/ soil $CO_2$ uptake by continental weathering and alkalinity river transport to the oceans. Glob Biogeochem

Cycles 17, 1038.

3. An, Z., Di, Q., 2016. Investigation of geological structures with a view to HLRW

disposal, as revealed through 3D inversion of aeromagnetic and gravity data and the results of CSAMT exploration. Journal of Applied Geophysics 135, 204–211.

4. An, Z., Di, Q., Wu, F., Wang, G., Wang, R., 2012. Geophysical exploration for a long deep tunnel to divert water from the Yangtze to the Yellow River, China. Bulletin of

Engineering Geology and the Environment 71, 195–200

5.  Asfahani, J., 2023. Estimation of the hydraulic parameters by using an alternative vertical
electrical sounding technique: case study from semiarid Khanasser valley region
Northern Syria. Acta Geophys 71, 997–1013.

6.  Attwa, M., Basokur, A., Akca, I., 2014. Hydraulic conductivity estimation using direct
current (DC) sounding data: a case study in East Nile Delta Egypt. Hydrgeol J 22, 1163–
1178.

7.  Bai, D., Unsworth, M., Meju, M., Ma, X., Teng, J., Kong, X., Sun, Y., Sun, J., Wang, L.,
Jiang, C., Zhao, C., Xiao, P., Liu, M., 2010. Crustal deformation of the eastern Tibetan
plateau revealed by magnetotelluric imaging. Nature Geosci 3, 358–362.

8.  Bentley, L.R., Gharibi, M., 2004. Two- and three-dimensional electrical resistivity
imaging at a heterogeneous remediation site. Geophysics 69, 674–680.

9.  Borah, U.K., Patro, P.K., 2019. Estimation of the depth of investigation in the
magnetotelluric method from the phase. Geophysics 84 (6), E377–E385.

10. Bréard Lanoix, M.L., Pabst, T., Aubertin, M., 2020. Correction: field determination of
the hydraulic conductivity of a compacted sand layer controlling water flow on an
experimental mine waste rock pile. Hydrogeol J 28, 1517.

11. Cagniard, L., 1953. Basic theory of the magneto-telluric method of geophysical
prospecting. Geophysics 18 (3), 605–635.

12. Camporese, M., Cassiani, G., Deiana, R., Salandin, P., 2011. Assessment of local
hydraulic properties from electrical resistivity tomography monitoring of a
three-dimensional synthetic tracer test experiment. Water Resources Research, 47 (12).

13. Cassidy, R., Comte, J.C., Nitsche, J., Wilson, C., Flynn, R., Ofterdinger, U., 2014.
Combining multi-scale geophysical techniques for robust hydro-structural characterisation in catchments underlain by hard rock in post-glacial regions. Journal of

Hydrology 517, 715–731.

14. Chambers, J.E., Kuras, O., Meldrum, P.I., Ogilvy, R.D., Hollands, J., 2006. Electrical resistivity tomography applied to geologic, hydrogeologic, and engineering investigations at a former waste-disposal site. Geophysics 71 (6), B231–B239.

15. Chandra, S., Ahmed, S., Ram, A., Dewandel, B., 2008. Estimation of hard rock aquifers hydraulic conductivity from geoelectrical measurements: a theoretical development with field application. J Hydrol 357, 218–227.

16. Chen, J., Hubbard, S., Rubin, Y., 2001. Estimating the hydraulic conductivity at the

South Oyster Site from geophysical tomographic data using Bayesian techniques based on the normal linear regression model displays variation Oyster Site. Water Resour Res 6,

1603–1613.

17. Courtois, N., Lachassagne, P., Wyns, R., Blanchin, R., Bougaïre, F.D., Some, S.,

Tapsoba, A., 2010. Large-scale mapping of hard-rock aquifer properties applied to

Burkina Faso. Groundwater 48 (2), 269–283.

18. Cui, L.X., Cheng, Q., So, P.S., Tang, C.S., Tian, B.G., Li, C.Y., 2024. Relationship between root characteristics and saturated hydraulic conductivity in a grassed clayey soil.

Journal of Hydrology 645 (2), 132231.

19. da Silva, C.C.N., de Medeiros, W.E., de Sá, E.F.J., Neto, P.X., 2004. Resistivity and ground-penetrating radar images of fractures in a crystalline aquifer: a case study in

Caiçara farm—NE Brazil. Journal of Applied Geophysics 56 (4), 295–307.

20. De Lima, O.A.L., Niwas, S., 2000. Estimation of hydraulic parameters of shaly sandstone aquifers from geological measurements. J Hydrol 235, 12–26.

21. Dewandel, B., Aunay, B., Maréchal, J.C., Roques, C., Bour, O., Mougin, B., Aquilina, L.,
2014. Analytical solutions for analysing pumping tests in a sub-vertical and anisotropic
fault zone draining shallow aquifers. J Hydrol 509, 115–131.

22. Dewandel, B., Jeanpert, J., Ladouche, B., Join, J.L., Maréchal, J.C., 2017. Inferring the
heterogeneity, transmissivity and hydraulic conductivity of crystalline aquifers from a
detailed water-table map. J Hydrol 550, 118–129.

23. Dewandel, B., Lachassagne, P., Qatan, A., 2004. Spatial measurements of stream
baseflow, a relevant method for aquifer characterization and permeability evaluation:
application to a hard-rock aquifer, the Oman ophiolite. Hydrol Process 18(17), 3391–
3400.

24. Di, Q., Fu, C., An, Z., Wang, R., Wang, G., Wang, M., Qi, S., Liang, P., 2020. An
application of CSAMT for detecting weak geological structures near the deeply buried
long tunnel of the Shijiazhuang-Taiyuan passenger railway line in the Taihang Mountains.
Engineering Geology 268, 105517.

25. Fernando, A., Pacheco, L., 2015. Regional groundwater flow in hard rocks. Science of
the Total Environment 506–507, 182–195.

26. Ferris, D.M., Potter, G., Ferguson, G., 2020. Characterization of the hydraulic
conductivity of glacial till aquitards. Hydrogeol J 28, 1827–1839.

27. Francese, R., Mazzarini, F., Bistacchi, A., Morelli, G., Pasquarè, G., Praticelli, N., Zaja,
A., 2009. A structural and geophysical approach to the study of fractured aquifers in the
Scansano-Magliano in Toscana Ridge, southern Tuscany, Italy. Hydrogeology Journal 17
(5), 1233.

28. Fu, C., Di, Q., An, Z., 2013. Application of the CSAMT method to groundwater
exploration in a metropolitan environment. Geophysics 78 (5), 201–B209.

29. Fu, T., Chen, H., Zhang, W., Nie, Y., Wang, K., 2015. Vertical distribution of soil
saturated hydraulic conductivity and its influencing factors in a small karst catchment in
Southwest China. Environ Monit Assess 187, 92.

30. Fusheng, G., Haiyan, Y., Zengqian, H., Zhichun, W., Ziyu, L., Guocan, W., Linfu, X., Ye,
G., Wanpeng, Z., 2022. Structural setting of the Zoujiashan-Julong'an region, Xiangshan
volcanic basin, China, interpreted from modern CSAMT data. Ore Geology Reviews. 150,
105180.

31. Gao, Q., Hasan, M., Shang Y., Qi, S., 2024. Geophysical estimation of 2D hydraulic
conductivity for groundwater assessment in hard rock. Acta Geophys 72, 4343–4354.

32. Hasan, M., Shang, Y., Di, Q., Meng, Q., 2024. Estimation of Young's modulus for rocks
using a non-invasive CSAMT method. Bulletin of Engineering Geology and the
Environment 83, 464.

33. Hasan, M., Shang, Y., Jin, W., Akhter, G., 2021. Estimation of hydraulic parameters in a
hard rock aquifer using integrated surface geoelectrical method and pumping test data in
southeast Guangdong China. Geosci J 25 (2), 223–242.

34. Hu, X.Y., Peng, R.H., Wu, G.J., Wang, W.P., Huo, G.P., Han, B., 2013. Mineral
exploration using CSAMT data: application to Longmen region metallogenic belt,
Guangdong Province, China. Geophysics 78, B111–B119.

35. Hubbard, S.H., Rubin, Y., 2002. Hydrogeological parameter estimation using
geophysical data: a review of selected techniques. J Contam Hydrol 45 (3), 34.

36. Jasechko, S., Seybold, H., Perrone, D., Fan, Y., Shamsudduha, M., Taylor, R.G., Fallatah,

O., Kirchner, J.W., 2024. Rapid groundwater decline and some cases of recovery in aquifers globally. Nature 625, 715–721.

37. Kouadio, K.L., Liu, R., Malory, A.O., Liu, C., 2023. A novel approach for water reservoir mapping using controlled source audio-frequency magnetotelluric in Xingning area, Hunan Province, China. Geophysical Prospecting 71, 1708–1727.

38. Kouadio, K.L., Xu, Y., Liu, C.M., Boukhalfa, Z., 2020. Two-dimensional inversion of

CSAMT data and three-dimensional geological mapping for groundwater exploration in

Tongkeng Area, Hunan Province, China. Journal of Applied Geophysics 183, 104204.

39. Lachassagne, P., Dewandel, B., Wyns, R., 2021. Review: Hydrogeology of weathered crystalline/hard-rock aquifers—guidelines for the operational survey and management of their groundwater resources. Hydrogeol J 29, 2561–2594.

40. Lachassagne, P., Wyns, R., Bérard, P., Bruel, T., Chéry, L., Coutand, T., Le Strat, P.,

2001. Exploitation of high-yields in hard-rock aquifers: Downscaling methodology combining GIS and multicriteria analysis to delineate field prospecting zones. Groundwater 39 (4), 568–581.

41. Laghari, A.N., Vanham, D., Rauch, W., 2012. The Indus basin in the framework of current and future water resources management. Hydrology and Earth System

Sciences 16 (4), 1063–1083.

42. Leal, J., Avila, E.A., Darghan, A.E., Lobo, D., 2023. Selection of spatial prediction models of saturated hydraulic conductivity in soils containing rock fragments in an

Andean micro-basin. Model Earth Syst Environ 9, 4223–4235.

43. Lin, C.H., Lin, C.P., Hung, Y.C., Chung, C.C., Wu, P.L., Liu, H.C., 2018. Application of geophysical methods in a dam project: Life cycle perspective and Taiwan experience.

Journal of Applied Geophysics 158, 82–92.

44. Loperte, A., Soldovieri, F., Palombo, A., Santini, F., Lapenna, V., 2016. An integrated geophysical approach for water infiltration detection and characterization at Monte

Cotugno rock-fill dam (southern Italy). Eng Geol 211, 162–170.

45. Majumdar, R.K., Das, D., 2011. Hydrological characterization and estimation of aquifer properties from electrical sounding data in Sagar Island region, South 24 Parganas, West

Bengal, India. Asian J Earth Sci 4, 60–74.

46. Maréchal, J.C., Dewandel, B., Subrahmanyam, K., 2004. Use of hydraulic tests at different scales to characterize fracture network properties in the weathered-fractured layer of a hard rock aquifer. Water Resources Research 40 (11).

47. McLachlan, P.J., Chambers, J.E., Uhlemann, S.S., Binley, A., 2017. Geophysical characterisation of the groundwater–surface water interface. Advances in Water

Resources 109, 302–319.

48. Minutti, C., Illman, W.A., Gomez, S., 2020. A new inverse modeling approach for hydraulic conductivity estimation based on Gaussian mixtures. Water Resources

Research, 56 (9), e2019WR026531.

49. Mira Geoscience Ltd., 1999. GOCAD Mining Suite 3D Geological Modeling Software.

Nancy University, Lorraine, France.

50. Misstear, R., Brown, L., Daly, D., 2009. A methodology for making initial estimates of groundwater recharge from groundwater vulnerability mapping. Hydrogeology

Journal 17 (2), 275.

51. Nguyen, M., Lin, Y.N., Tran, Q.C., Ni, C.F., Chan, Y.C., Tseng, K.H., Chang, C.P., 2022.

Assessment of long-term ground subsidence and groundwater depletion in Hanoi,

Vietnam. Engineering Geology 299, 106555.

52. Niwas, S., Celik, M., 2012. Equation estimation of porosity and hydraulic conductivity of

Ruhrtal aquifer in Germany using near surface geophysics. J Appl Geophys 84, 77–85.

53. Niwas, S., De Lima, O.A.L., 2003. Aquifer parameter estimation from surface resistivity data. Groundwater 41, 94–99.

54. Nwosu, L.I., Nwankwo, C.N., Ekine, A.S., 2013. Geoelectric investigation of the hydraulic properties of the aquiferous zones for evaluation of groundwater potentials in the complex geological area of imostate, Nigeria. Asian J Earth Sci 6, 1–15.

55. Oli, I.C., Opara, A.I., Okeke, O.C., Akaolisa, C.Z., Akakuru, O.C., Osi-Okeke, I., Udeh,

H.M., 2022. Evaluation of aquifer hydraulic conductivity and transmissivity of

Ezza/Ikwo area, Southeastern Nigeria, using pumping test and surficial resistivity techniques. Environ Monit Assess 194, 719.

56. Parks, E.M., McBride, J.H., Nelson, S.T., Tingey, D.G., Mayo, A.L., Guthrie, W.S.,

Hoopes, J.C., 2011. Comparing electromagnetic and seismic geophysical methods:

estimating the depth to water in geologically simple and complex arid environments.

Engineering Geology 117 (1–2), 62–77.

57. Phoenix Geophysics CMTPro, 2020. The Canadian Phoenix CMT Pro Version software for CSAMT data processing. Toronto, Ontario, Canada.

58. Phoenix Geophysics CSAMT-SW, 2020. The Canadian Phoenix CSAMT-SW Version software for CSAMT data inversion. Toronto, Ontario, Canada.

59. Porsani, J.L., Elis, V.R., Hiodo, F.Y., 2005. Geophysical investigations for the
characterization of fractured rock aquifers in Itu, SE Brazil. Journal of Applied
Geophysics, 57 (2), 119–128.

60. Purvance, D.T., Andricevic, R., 2000. On the electrical-hydraulic conductivity correlation
in aquifers. Water Resour Res 36, 205–213.

61. Qin, X., 2017. Application of Unwedge program to geological stability analysis of deep
buried deposits. Comprehensive 8, 270–273 (In Chinese)

62. Rao, P.V., Subrahmanyam, M., Raju, B.A.G., 2022. Investigation of groundwater
potential zones in hard rock terrains along EGMB, India, using remote sensing,
geoelectrical and hydrological parameters. Acta Geophys 71, 1867–1883.

63. Rashid, M., Lone, M.A., Ahmed, S., 2012. Integrating geospatial and ground geophysical
information as guidelines for groundwater potential zones in hard rock terrains of south
India. Environ Monit Assess 184, 4829–4839.

64. Refsgaard, J.C., Christensen, S., Sonnenborg, T.O., Seifert, D., Højberg, A.L., Troldborg,
L., 2012. Review of strategies for handling geological uncertainty in groundwater flow
and transport modeling. Advances in Water Resources 36, 36–50.

65. Robinson, J., Slater, L., Johnson, T., Shapiro, A., Tiedeman, C., Ntarlagiannis, D.,
Johnson, C., Day-Lewis, F., Lacombe, P., Imbrigiotta, T., Lane, J., 2016. Imaging
pathways in fractured rock using three-dimensional electrical resistivity tomography.
Groundwater 54 (2), 186–201.

66. Rodell, M., Velicogna, I., Famiglietti, J.S., 2009. Satellite-based estimates of
groundwater depletion in India. Nature 460 (7258), 999–1002.

67. Rodi, W., Mackie, R.L., 2001. Nonlinear conjugate gradients algorithm for 2-D
magnetotelluric inversion. Geophysics 66 (1), 174–187.

68. Roques, C., Aquilina, L., Boisson, A., Vergnaud-Ayraud, V., Labasque, T.,
Longuevergne, L., Laurencelle, M., Dufresne, A., de Dreuzy, J.R., Pauwels, H., Bour, O.,
2018. Autotrophic denitrification supported by biotite dissolution in crystalline aquifers:
(2) transient mixing and denitrification dynamic during long-term pumping. Sci Total
Environ 619–620, 491–503.

69. Sale, H.S., 2001. Modelling of lithology and hydraulic conductivity of shallow sediments
from resistivity measurements using Schlumberger vertical electric soundings. Energy
Sources 23, 599–618.

70. Sikandar, P., Christen, E.W., 2012. Geoelectrical sounding for the estimation of hydraulic
conductivity of alluvial aquifers. Water Resour Manag 26, 1201–1215.

71. Simpson, F., Bahr, K., 2005. Practical magnetotellurics. Cambridge University Press,
Cambridge. 254 pp.

72. Singh, K.P., 2005. Nonlinear estimation of aquifer parameters from surficial resistivity
measurements. Hydrology and Earth System Sciences Discussions 2 (3), 917–938.

73. Sinha, R., Israil, M., Singhal, D.C., 2009. A hydrogeological model of the relationship
between geoelectric and hydraulic parameters of anisotropic aquifers. Hydrogeol J 17,
495–503.

74. Slater, L., 2007. Near surface electrical characterization of hydraulic conductivity: from
petrophysical properties to aquifer geometries—a review. Surv Geophys 28, 169–197.

75. Smith, J.T., Booker, J.R., 1991. Rapid inversion of two-and three-dimensional
magnetotelluric data. Journal of Geophysical Research: Solid Earth 96 (B3), 3905–3922.

76. Soro, D.D., Koita, M., Biaou, C.A., Outoumbe, E., Vouillamoz, J.M., Yacouba, H.,

Guérin, R., 2017. Geophysical demonstration of the absence of correlation between lineaments and hydrogeologically useful fractures: case study of the Sanon hard rock aquifer (central northern Burkina Faso). J African Earth Sci 129, 842–852.

77. Soupios, P.M., Kouli, M., Vallianatos, F., Vafidis, A., Stavroulakis, G., 2007. Estimation of aquifer hydraulic parameters from surficial geophysical methods: a case study of

Keritis Basin in Chania (Crete–Greece). J Hydrol 1, 122–131.

78. Trinh, T., Kavvas, M.L., Ishida, K., Ercan, A., Chen, Z.Q., Anderson, M.L., Ho, C.,

Nguyen, T., 2018. Integrating global land-cover and soil datasets to update saturated hydraulic conductivity parameterization in hydrologic modeling. Science of the Total

Environment 631–632, 279–288.

79. Vassolo, S., Neukum, C., Tiberghien, C., Heckmann, M., Hahne Désiré, K., Baranyikwa,

D., 2019. Hydrogeology of a weathered fractured aquifer system near Gitega, Burundi.

Hydrogeol J 27, 625.

80. Vouillamoz, J.M., Lawson, F.M.A., Yalo, N., Descloitres, M., 2014. The use of magnetic resonance sounding for quantifying specific yield and transmissivity in hard rock aquifers:

the example of Beni. J Appl Geophys 107, 16–24.

81. Wada, Y., Van Beek, L.P., Van Kempen, C.M., Reckman, J.W., Vasak, S., Bierkens,

M.F., 2010. Global depletion of groundwater resources. Geophysical Research Letters 37

(20).

82. Wada, Y., Wisser, D., Bierkens, M.F., 2014. Global modeling of withdrawal, allocation and consumptive use of surface water and groundwater resources. Earth System

Dynamics 5 (1), 15–40.

83. Wang, R., Yin, C., Wang, M., Di, Q., 2015. Laterally constrained inversion for CSAMT

data interpretation. Journal of Applied Geophysics 121, 63–70.

84. Webring, M.W., 1981. MINC: A Gridding Program Based on Minimum Curvature: U.S.

Geological Survey Open File Report. 81–1224, p. 41p

85. Worthington, S.R.H., Davies, G.J., Alexander, E.C. Jr., 2016. Enhancement of bedrock permeability by weathering. Earth-Sci Rev 160, 188–202.

86. Wynn, J., Mosbrucker, A., Pierce, H., Spicer, K., 2016. Where is the hot rock and where is the ground water-using CSAMT to map beneath and around Mount St. Helens. Journal of Environmental and Engineering Geophysics 21, 79–87.

87. Yadav, G.S., Singh, S.K., 2007. Integrated resistivity surveys for delineation of fractures for ground water exploration in hard rock areas. Journal of Applied Geophysics 62 (3),

301–312.

88. Yang, H.Q., Zhang, L., 2024. Bayesian back analysis of unsaturated hydraulic parameters for rainfall-induced slope failure: A review. Earth-Science Reviews 251, 104714.

89. Yang, J., Zhang, H., Cui, Z., 2021. Stability Analysis and Countermeasures of Rock

Block in Underground Cavern. Guangdong Water Resources and Hydropower 5, 23–27.

(In Chinese)

90. Yao, S., Zhang, T., Zhao, C., Liu, X., 2013. Saturated hydraulic conductivity of soils in the Horqin Sand Land of Inner Mongolia, northern China. Environ Monit Assess 185,

6013–6021.

91. Zhang, J., Sirieix, C., Genty, D., Salmon, F., Verdet, C., Mateo, S., Xu, S., Bujan, S.,

Devaux, L., Larcanché, M., 2024. Imaging hydrological dynamics in karst unsaturated zones by time-lapse electrical resistivity tomography. Science of the Total Environment

907, 168037.

92. Zhang, M., Farquharson, C.G., Li, C., 2021. Improved controlled source audio-frequency magnetotelluric method apparent resistivity pseudo-sections based on the frequency and frequency–spatial gradients of electromagnetic fields. Geophysical Prospecting 69, 474–

490.

93. Zonge, K.L., Hughes, L.J., 1988. Electromagnetic Methods—Theory and Practice.

94. Zoorabadi, M., Saydam, S., Timms, W., Hebblewhite, B., 2022. Analytical methods to estimate the hydraulic conductivity of jointed rocks. Hydrogeol J 30, 111–119.

**Declaration of interests**

☑ The authors declare that they have no known competing financial interests or personal relationships that could have appeared to influence the work reported in this paper.

☐ The author is an Editorial Board Member/Editor-in-Chief/Associate Editor/Guest Editor for *[Journal name]* and was not involved in the editorial review or the decision to publish this article.

☐ The authors declare the following financial interests/personal relationships which may be considered as potential competing interests:

---

## Author Comment (AC1)

**Referee Comments (RC2):**

This paper addresses the challenges associated with current groundwater exploration and evaluates the advantages and disadvantages of various methods for measuring hydraulic parameters. The author highlights the application of a novel approach, by using Controlled Source Audio-Frequency Magnetotellurics (CSAMT) method, which is employed to estimate 2D and 3D permeability at depths exceeding 1 km in highly heterogeneous rock environments. The study presents its methods and findings in a well-structured manner, offering insights into deep groundwater exploration.

However, certain assertions appear overly generalized and could benefit from further substantiation. Additionally, more detailed descriptions of the methodologies and the study area would enhance the clarity, reproducibility, and robustness of the research.

**Response:**

We sincerely thank the anonymous reviewer for their insightful and constructive feedback, which has significantly contributed to enhancing the quality of our work. We have made every effort to revise the manuscript thoroughly in line with the reviewer's suggestions.

In the revised version, we have expanded the descriptions of both the study area and the methodologies to provide greater clarity and context. Additional explanations and supporting information have also been included to substantiate our findings and address the points raised.

As per the journal's submission guidelines, we are first submitting our detailed responses to the reviewer's comments. Following this, we will submit the revised manuscript reflecting all the suggested changes.

**Specific comments:**

**Comment 1:**

Line108-119: the author might consider adding a little more evidence of the reason that CSAMT is selected for this study. For example, the author stated that VES method is used to evaluate groundwater resources in a single dimension by a broad of previous studies, but did not illustrate the background about why they did not use other methods, like CSAMT or ERT. Additionally, the author states that there are three main methods, but there are very few examples or introductions about ERT in this paragraph.

**Response 1:**

The entire paragraph from Lines 88–119 has been revised for improved clarity and structure in the updated manuscript. Both the original and the revised versions of the paragraph are provided below for comparison.

ORIGINAL PARAGRAPH:

[revised manuscript text omitted]

groundwater resources in a single dimension. In recent decades, electrical resistivity tomography (ERT) has been widely used in hydrogeological studies for 2D and 3D assessment of groundwater resources (Bentley and Gharibi, 2004; Camporese et al., 2011; MLin et al., 2018; Abbas et al., 2022). However, it is unusual to evaluate aquifer yield at great depths in hard rock terrains using two and three dimensional hydraulic properties. Recent studies have demonstrated that CSAMT, which aims to gather extensive subsurface data at very deep depths using 2D/3D evaluations, is the most appropriate geophysical method for researching hard rock (Smith and Booker, 1991; Simpson and Bahr, 2005; Bai et al., 2010; Fu et al., 2013; Hu et al., 2013; Wang et al., 2015; Wynn et al., 2016; Zhang et al., 2021; Kouadio et al., 2023). The selection of resistivity methods in groundwater studies depends on several key factors, including survey objectives, depth of investigation, resolution, geological complexity, logistical constraints, cost and accessibility, electrical conductivity contrast, and field conditions, etc (Di et al., 2020; Hasan et al., 2024). VES is more suitable for shallow depths (< 200 m) for 1D resistivity imaging, has limited lateral resolution, is useful for layered aquifer characterization, works well in horizontally stratified formations, requires minimal equipment and is quick to deploy, is the most economical for small-scale studies, and may face limitations in highly resistive or conductive terrains (Soupios et al., 2007; Nwosu et al., 2013; Hasan et al., 2021). ERT offers better resolution for both shallow and intermediate depths (up to ~300 m) with 2D/3D imaging, provides high-resolution subsurface imaging, is ideal for detecting fractures/faults and heterogeneous aquifers, is better for complex geology (e.g., fractured zones, karst systems), is ideal for detailed aquifer geometry and contamination studies, needs more field effort and electrode spacing adjustments, and may face limitations in highly resistive or conductive terrains like VES (Camporese et al., 2011; MLin et al., 2018; Abbas et al., 2022). CSAMT is effective

for deeper investigations (hundreds to thousands of meters) due to its low-frequency signal penetration, provides 2D/3D imaging over big area at large depths, has lower resolution than ERT but excels in deep structural mapping, is preferred for deep-seated structures like basement aquifers or geothermal systems, is used for regional groundwater exploration, demands specialized equipment and is more time-consuming compared with VES and ERT, is relatively more expensive than VES and ERT due to advanced instrumentation and processing, all resistivity methods rely on resistivity contrasts but CSAMT is more sensitive to deep conductive zones, and performs better in areas with cultural noise (e.g., urban settings) due to controlled signal sources (Zonge and Hughes, 1988; An et al., 2016; Kouadio et al., 2020; Zhang et al., 2021). When combined with empirically based methodologies, CSAMT becomes an even more powerful tool for studying the incredibly diverse topographical features at large depths (Hasan et al., 2024). So based on above factors, CSAMT was the most suitable method for this study."

**Comment 2:**

Line 120- 138: the author might consider reorganizing this paragraph to make the significance of the resistivity method stand out.

**Response 2:**

The paragraph previously located in Lines 120–138 has been reorganized for improved clarity in the revised manuscript. For reference, both the original and the revised versions of the paragraph are provided below.

ORIGINAL PARAGRAPH:

"Several factors, such as the type of rock, fault, weathering degree, fluid content, permeability, pore-spacing, fracture, lithology, saturation, and joints, as well as the same structural heterogeneities, determine the geophysical and aquifer characteristics (Singh, 2005; Sinha et al., 2009; Hasan et al., 2021). Several prior studies utilized geophysical parameters in conjunction with hydraulic data or lithological logs to characterize underlying rock mass units hydrogeologically (De Lima and Niwas, 2000; Hubbard and Rubin, 2002; Niwas and De Lima, 2003; Singh, 2005; Soupios et al., 2007; Sinha et al., 2009; Majumdar and Das, 2011; Nwosu et al., 2013; Hasan et al., 2021; Asfahani, 2023). Resistivity methods provide an alternate option for aquifer parameter estimation by creating a beneficial relationship between electrical resistivity and the aquifer parameters (obtained from drilling tests). An innovative aspect of this work is its use of non-invasive geophysical techniques to create two- and three-dimensional k models in a diverse environment with a variety of rock types and significant depths. The planned study will necessitate the boring of a handful of boreholes at key spots all around the project site. A more trustworthy CSAMT study will allow us to evaluate the extensive research area. Then, by directly connecting geophysical and borehole data, k can be established for the entire researched site, even without drilling tests. Two- and three-dimensional k models are generated by applying the resulting equations to the full study area. This approach would reduce the need for costly boreholes to obtain a thorough and complete evaluation of subsurface hydrogeological conditions."

REVISED PARAGRAPH:

"Resistivity methods are highly significant in groundwater studies due to their ability to characterize subsurface formations and identify potential aquifers. These techniques measure the electrical resistivity of subsurface materials, which varies depending on lithology, rock type,

porosity, fluid content, weathering degree, faults, fractures, joints, saturation, and salinity (Singh, 2005; Sinha et al., 2009; Hasan et al., 2021). By integrating resistivity data with geological and hydrogeological information, researchers have been able to optimize the placement of wells, accurately assess groundwater potential, and support the effective and sustainable management of water resources (De Lima and Niwas, 2000; Hubbard and Rubin, 2002; Niwas and De Lima, 2003; Singh, 2005; Soupios et al., 2007; Sinha et al., 2009; Majumdar and Das, 2011; Nwosu et al., 2013; Hasan et al., 2021; Asfahani, 2023). By establishing a useful correlation between electrical resistivity and the limited borehole-based hydraulic parameters, resistivity methods offer the best alternative of the expensive drilling tests for estimating aquifer parameters over large area from shallow to large depths. In this study, we present a novel application of the Controlled Source Audio-Frequency Magnetotelluric (CSAMT) method to develop high-resolution 2D and 3D permeability (k) models at significant depths (up to 1300 meters) within a geologically complex and heterogeneous setting composed of sandstone, granite, and hornstone. Initially, a limited number of boreholes were drilled at strategically selected key locations across the study area. Subsequently, multiple CSAMT survey profiles were conducted, covering the entire region, including the borehole sites. By correlating the resistivity data obtained from the CSAMT surveys with permeability values derived from the borehole data, we established a reliable empirical relationship between resistivity and permeability. This correlation was then applied across the full CSAMT dataset, enabling the generation of 2D and 3D permeability models even in areas lacking direct borehole information. This approach allows for a more comprehensive and cost-effective assessment of deep groundwater resources, significantly reducing the need for extensive and expensive drilling campaigns."

**Comment 3:**

Line 139- 140: the statement is too arbitrary; the language can be modified or more evidence is provided.

**Response 3:**

ORIGINAL STATEMENT:

"No one had ever tried to estimate K using direct or indirect methods in such a heterogeneous context before this work, where a broad diversity of rock types are present at a depth of 1 kilometer".

REVISED STATEMENT:

"Prior to this study, no attempts had been made to estimate permeability (K) using either direct methods, such as borehole testing, or indirect geophysical approaches in such a geologically heterogeneous setting, characterized by a diverse mixture of sandstone, granite, and hornstone, extending to depths of up to one kilometer."

**Comment 4:**

Line 161: in section 2.1 Study area, the author might consider adding more details about the rocks and geology of the study area.

**Response 4:**

Additional details about the rocks and geological characteristics of the study area have been added, as shown below.

ORIGINAL:

"Intruding rocks from the Indosinian, Caledonian, and Yanshanian eras are among the many geological formations and periods represented in the study region. Other layers from the Paleogene period are also present. The most common types of rock that have been discovered are sandstone, granite, and hornstone. The complex Kaiping concave fault and fold systems were the dominant geological features in the project region, which were developed as a result of magmatic processes and various structures (Qin, 2017). Emergence of joint fissured features symbolizes the various tectono-geological periods, with the local tectonic line corresponding with the faults strike, especially in the northeast orientation (Yang et al., 2021)".

REVISED:

"The study area exhibits a complex and diverse geological history, characterized by well-defined geometrical relationships among various lithologies. These formations and structural features are the result of multiple tectono-magmatic events spanning several geological periods. Intrusive rocks from the Indosinian (Late Triassic), Caledonian (Silurian–Devonian), and Yanshanian (Jurassic–Cretaceous) orogenies are well-represented, indicating a long sequence of crustal deformation and magmatic activity. These intrusions are primarily composed of granitic bodies, which suggest deep-seated magmatic processes associated with continental collision and subduction zones. In addition to these intrusive phases, sedimentary strata from the Paleogene

period are also present, reflecting a later stage of basin development with fluvial and lacustrine depositional environments. Among the most prevalent rock types encountered in the region are sandstone, granite, and hornstone. Sandstone reflects high-energy sedimentary deposition. Granite indicates deep magmatic intrusions likely associated with Yanshanian tectonics. Hornstone (hornfels) results from contact metamorphism caused by magma intruding sedimentary rocks. The structural framework of the region is dominated by the Kaiping concave fault and fold system, a geologically significant and highly deformed zone that reflects multiple deformation episodes (Qin, 2017). These structures were primarily shaped by magmatic intrusions, crustal movements, and regional stress regimes. The presence of extensive jointed and fissured zones throughout the rock mass further supports a history of dynamic tectonic activity. These joints often serve as secondary permeability pathways and are critical in controlling groundwater flow in the fractured rock environment. Importantly, the orientation of these structural features, including faults and joints, is often aligned with northeast-trending tectonic lines, which are consistent with broader regional stress directions (Yang et al., 2021). This relationship among lithologies and structural features plays a critical role in controlling groundwater flow and permeability distribution."

**Comment 5:**

Line 204: what does "5-20%" represent for? More specific content is preferred for this sentence.

**Response 5:**

"The vertical resolution of 5–20% can be assessed by CSAMT when exploring depths ranging from 20 to 1000 meters" explained as in the revised version:

"In CSAMT, the vertical resolution, which refers to the ability to distinguish between adjacent subsurface layers, can typically range between 5% and 20% of the investigation depth from approximately 20 to 1000 meters. At shallower depths (e.g., 20–100 m), vertical resolution is higher (closer to 5%), enabling better differentiation between thin layers. At greater depths (up to 1000 m), resolution may degrade toward the 20% mark due to signal attenuation and broader averaging of resistivity data. This makes CSAMT a valuable tool for identifying significant lithological contrasts, fault zones, and resistivity anomalies related to geological structures".

**Comment 6:**

Line 217: in section 2.2.2, why were 6 profiles selected? How did the author determine the locations of the profiles? The author might consider providing more evidence of the site location and data collection in the supporting material.

**Response 6:**

Further details regarding the selection criteria and rationale for the survey profiles have been provided in the revised manuscript.

ORIGINAL:

"The CSAMT data was acquired using six profiles (1–6) with a 50 meter interval between each station".

REVISED:

"CSAMT data were acquired along six profiles (profiles 1–6), with a station spacing of 50 meters between each measurement point. The location of 6 CSAMT profiles was chosen based on several factors, including geological targets and objectives, surface geology and mapping data, topography and terrain accessibility, orientation relative to structures, spacing and coverage requirements, resistivity contrast expectations, integration with other data (boreholes), environmental and regulatory constraints, and source-receiver geometry requirements, etc. Carefully selected survey profiles enhanced the ability to resolve critical subsurface features and minimized ambiguities in the geophysical interpretation".

Additional details are provided in the revised manuscript.

**Comment 7:**

Line 250-154: The author might consider providing more details of the static correction and the Hanning window spatial filtering method.

**Response 7:**

Additional details on static correction and Hanning window spatial filtering have been included in the revised manuscript to enhance clarity and support the interpretation of CSAMT data.

ORIGINAL:

"The static corrections were made using a Hanning window spatial filtering method, which involved geological information and curve analysis."

REVISED:

"Static correction and spatial filtering using a Hanning window are essential preprocessing steps in CSAMT data analysis, aimed at improving data quality and enhancing the reliability of subsurface resistivity models. Static correction addresses the effects of near-surface resistivity inhomogeneities, which can distort electric field measurements and introduce static shifts, vertical displacements in apparent resistivity curves that misrepresent deeper subsurface conditions. This correction typically involves adjusting the measured electric fields by referencing them to a stable or averaged field, effectively removing shallow-layer influences and isolating true subsurface signals. Spatial filtering, on the other hand, is used to mitigate noise introduced by environmental and instrumental sources. Among various filters, the Hanning (Hann) window is commonly applied due to its effectiveness in reducing spectral leakage and smoothing data. When used in spatial filtering, the Hanning window averages measurements across adjacent stations in a weighted manner, preserving spatial trends while suppressing high-frequency noise. This improves the coherence of the dataset and ensures more stable and interpretable inversion results".

**Comment 8:**

Figure 1: typos in (b), "uncertainty"; also the words are too small to read.

**Response 8:**

Figure 1, along with all other figures, has been redrawn and improved for better clarity and presentation. The updated figs are included at the end of the response/comments section in the attached file.

**Comment 9:**

Figure 7 and Figure 8: a little confused about the legend of the north direction in both figures

**Response 9:**

The north direction in these figures is correctly oriented, though slightly tilted, to provide a clearer and more informative view of the 3D permeability (k) models. The revised figures are included at the end of the response/comments section in the attached file.

**Revised Figures**

[Figure]

(a)                 (b)

Fig. 1. (a) The location of the project site, with six boreholes BH1–BH6 (blue circles) and six CSAMT profiles 1-6 (black lines), (b) Flow diagram outlining the planned method for getting 2D and 3D k models for better, more thorough assessments of groundwater resources over large regions

[Figure]

**Fig. 2.** Displaying the procedure of 2D inversion of CSAMT data by the use of Bostick inversion

[Figure]

Fig 3. The evaluation of hornstone (HS), sandstone (SS), and granite (G) carried out by presenting 116 resistivity-k data points at depths ranging from 5 to 200 m using 6 drilled tests

(BH1–BH6) and associated resistivity (ρ) from CSAMT soundings. The small black dots show

the data points

[Figure]

Fig 4. Using a total of 116 data points, the geophysical-borehole correlation for the predicted k

[Figure]

Fig. 5 2D CSAMT models along six geophysical profiles 1-6. Where resistivity increases from brown to green on a color bar.

[Figure]

Fig. 6 The predicted 2D k models obtained from CSAMT data along six geophysical profiles 1-6.

Where k increases from light green to red on a color scale

[Figure]

Fig.7 The interpreted (hydrogeological) 2D models along six geophysical profiles 1–6 obtained via geophysical-borehole correlation, facilitates groundwater assessment through high potential aquifer (HPA), medium potential aquifer (MPA), and low potential aquifer (LPA) associated with sandstone (SS), hornstone (HS), and granite (G), respectively

[Figure]

Fig. 8 The integrated 2D k models derived from the incorporation of geophysical and drilling

data, with k represented on a color bar spanning from green to red

[Figure]

Fig. 9 Analysis of the integrated 2D k models (derived from designated k ranges) for three groundwater potential aquifers: low potential aquifer (LPA), medium potential aquifer (MPA), and high potential aquifer (HPA), associated with three geological formations: granite (G), hornstone (HS), and sandstone (SS), respectively

[Figure]

Fig. 10 The 3D k models, generated from the correlation of CSAMT and borehole data (with k represented on a color scale ranging from green to red), correspond to three groundwater potential aquifers: low potential aquifer (LPA), medium potential aquifer (MPA), and high

potential aquifer (HPA), associated with three geological strata: granite (G), hornstone (HS), and sandstone (SS), respectively, for (a) the external view of the 3D k model, and (b) the analysis of the 3D k model from an external perspective

[Figure]

Fig. 11 The 3D k models, obtained from the correlation of CSAMT and borehole data (with k represented on a color scale ranging from green to red), illustrate three groundwater potential aquifers: low potential aquifer (LPA), medium potential aquifer (MPA), and high potential aquifer (HPA), associated with three geological strata: granite (G), hornstone (HS), and sandstone (SS), respectively, for (a) the internal view of the 3D k model, and (b) the analysis of the 3D (internal perspective) k model

[Figure]

Fig. 12 (a) Geophysical-based k imaging at various depths (0, 200, 600, 1000, and 1300 m) with inner 3D view is represented by K on a color bar that goes from green to red, (b) Assessment of geophysical-derived k (using specified k ranges) at different depths for various types of aquifers: low potential aquifer (LPA) granite (G), medium potential aquifer (MPA) hornstone (HS), and high potential aquifer (HPA) sandstone (SS)

---

## Author Comment (AC3)

**Referee Comments (RC1):**

Hasan and Su tried to get the permeability of deep hard rock by a geophysical approach. They used the CSAMT to get the resistivity of the subsurface for several 2D profiles and then built an empirical equation to describe the relationship between permeability and resistivity using 37 permeabilities from boreholes. Then they applied the equation to all 2D profiles and finally extrapolated 2D profiles to a 3D field. They argued that the advantage of their approach is that it is able to get the information of the deep subsurface to 1 km depth. They also said their approach is low-cost than borehole drilling. I am sorry I am not that encouraging to this work. I don't think this work is novel enough for publication in HESS, a top journal in hydrology and water resources. I suggest rejecting it for publication in other journals of geology. Following are my reasons:

**Response:**

Dear Anonymous Reviewer,

Thank you for your detailed and thoughtful review of our manuscript. We sincerely appreciate the time and effort you dedicated to evaluating our work.

We greatly appreciate HESS's reputation as a leading journal in the fields of hydrology and water resources. As groundwater assessment inherently depends on a comprehensive understanding of subsurface geology, we believe that the integration of geophysical methods, particularly in complex geological settings, adds significant value to the scope of the journal.

This work builds upon our extensive experience in applying geophysical techniques to groundwater hydrology, with previous publications in the high-impact journals. Given the focus

and scientific standards of HESS, as well as the novelty of our current approach, we believe that this manuscript aligns well with the journal's objectives.

It is worth noting that the handling editors initially assessed the novelty and relevance of our submission before inviting peer review. Additionally, the other reviewers have recommended only minor revisions and expressed strong support for the publication of this work, further reinforcing its contribution and suitability for HESS.

We acknowledge that many of your comments are highly constructive and have been very helpful in improving our manuscript. These include your observation regarding the correct unit of permeability (mD, not m/d), your suggestion to expand the dataset, and your recommendation to discuss theoretical relationships such as the Archie and Kozeny-Carman equations, etc. All of these aspects have been carefully addressed in the revised version.

However, we also feel that certain criticisms may have overlooked the broader context and significance of our contributions, especially when compared with the existing body of literature. Although many studies in the past have explored the estimation of permeability (k) or other hydraulic parameters using geophysical data, this work introduces several key innovations that distinguish it from previous research.

When compared to existing literature, the following contributions represent significant advancements:

1. **First-time measurement of permeability (k) at depths exceeding 1 km** in a hard-rock environment.

2. **Generation of 2D and 3D k models**, to our knowledge, the first of their kind for deep groundwater assessments using any geophysical method.

3. **Innovative use of CSAMT** to derive volumetric estimations of hydraulic parameters such as permeability, which has not been attempted previously.

4. **Application in a highly heterogeneous lithological context** involving sandstone, granite, and hornstone, where such estimation of k had never been conducted before.

5. **Efficient integration of sparse borehole data** to produce high-resolution subsurface models over a large hard-rock terrain, demonstrating a practical alternative to intensive drilling.

6. **Significant reduction in the need for costly deep boreholes**, which would otherwise be required in the hundreds to achieve a similar level of spatial resolution.

We hope these clarifications help illustrate the novelty and importance of this work, and we remain grateful for your review, which ultimately contributed to strengthening our manuscript. We look forward to the opportunity to share this contribution with the HESS readership.

**Comment 1:**

Either component of this work is very old. The CSAMT can date back to 1970s. The relationship between the permeability and resistivity has also been largely studied so far. The inversion of permeability/hydraulic conductivity using geophysical approaches is so well-known.

**Response 1:**

We thank the reviewer for this important comment.

We agree that both CSAMT and the general concept of relating resistivity to permeability have a long history in geophysical and hydrogeological research. However, the key contribution of our study lies not in the introduction of entirely new methods, but in the **novel integration and application** of established techniques to a unique and challenging geological context, resulting in unprecedented outcomes.

Specifically, while CSAMT has traditionally been used for deep geophysical investigations, its application in **hydrogeology for constructing 2D and 3D models of permeability (k) at depths exceeding 1 km**, especially in a **highly heterogeneous hard-rock setting** comprising sandstone, granite, and hornstone, has **not been demonstrated in any previous work**, to the best of our knowledge.

In fact, no past studies have reported the generation of **volumetric (2D and 3D) models of k or any hydraulic parameter** using either direct (drilling-based) or indirect (geophysical, etc) methods at such depths and across such varied lithologies. In our earlier work, we demonstrated for the first time the use of **ERT** to generate shallow-depth 2D and 3D models of hydraulic conductivity in a tuff rock environment. Building on that, this study extends the concept both **methodologically and spatially** by using CSAMT to model permeability at much greater depths and over geologically complex terrain.

Furthermore, we have now included **a more detailed discussion in the revised manuscript** regarding the relationship between resistivity and permeability (e.g., using the Archie and Kozeny-Carman equations), along with how our approach advances existing methodologies.

We hope this clarification better highlights the **novelty, scale, and practical significance** of our work within the context of existing literature.

**Comment 2:**

Authors even didn't well discuss the previous studies of relationship between permeability and resistivity, such as the Archie equation and Kozeny-Carman equation.

**Response 2:**

We thank the reviewer for this insightful comment. In response, we have included a detailed discussion in the revised manuscript regarding previous studies on the relationship between resistivity and permeability, specifically addressing the Archie equation and the Kozeny-Carman equation. These models are fundamental in petrophysical and hydrogeophysical studies and serve as a theoretical basis for linking electrical resistivity measurements to hydraulic properties. Their relevance, limitations in heterogeneous geological settings, and how our approach builds upon these foundations are now clearly articulated in the revised version of the manuscript.

A brief discussion is given below:

**Discussion on the Relationship between Permeability and Resistivity**

Several foundational studies have established empirical and theoretical relationships between electrical resistivity and hydraulic properties such as permeability. The **Archie equation**, introduced by Archie (1942), is widely used in clean, saturated sedimentary formations. It relates

formation resistivity to porosity and water saturation but assumes the absence of clay minerals and thus has limitations in more complex lithologies.

The **Kozeny-Carman equation** is another widely accepted model that links permeability to porosity and specific surface area. While it does not directly involve resistivity, it is often used alongside petrophysical models to interpret hydrogeological characteristics based on geophysical data.

Although these relationships are well-established, their application is often restricted to homogeneous or semi-homogeneous geological settings. In contrast, our study extends the use of these principles into a **highly heterogeneous hard rock context**, including sandstone, granite, and hornstone, at depths exceeding 1 km. By integrating borehole-derived permeability with CSAMT-based resistivity data, we derive a site-specific empirical model that enables the construction of 2D and 3D permeability distributions across the entire study area, an approach not previously demonstrated in the literature.

**Archie's Equation**

Archie's law (Archie, 1942) relates the bulk electrical resistivity of a fully saturated porous medium to its porosity and fluid resistivity. It is commonly expressed as:

$$\rho b = a \cdot \rho f \cdot \phi^{-m}$$

where:

$\rho b$ is the bulk resistivity,

ρf is the fluid resistivity,

ϕ is the porosity,

a and m are empirical constants.

Although Archie's law does not directly estimate permeability, porosity is often used as a proxy because of its influence on fluid flow. The resistivity-porosity relationship can be indirectly extended to infer permeability, especially when combined with other petrophysical models.

**Kozeny-Carman Equation**

The Kozeny-Carman equation provides a direct relationship between permeability (kkk) and porosity and is given by:

$$k = \frac{C \cdot \phi^3}{(1 - \phi)^2 . S^2}$$

where:

k is the permeability,

ϕ is the porosity,

S is the specific surface area,

C is a constant related to pore structure and tortuosity.

By combining the Kozeny-Carman equation with Archie's law, researchers have developed empirical and semi-empirical models to relate geophysical measurements (like resistivity) to permeability.

**Relevance to This Study**

While these classical models provide important theoretical underpinnings, their direct application in complex geological environments, such as heterogeneous hard rock formations (e.g., granite, sandstone, hornstone), is often limited due to variability in mineralogy, pore structure, and anisotropy. In this study, we established an empirical relationship between resistivity and permeability based on field measurements from boreholes and CSAMT profiles. This site-specific correlation enables the generation of 2D and 3D permeability models in geologically complex settings where traditional models may fall short.

**Comment 3:**

A perfect equation (Fig. 3) was built using only 37 known permeabilities. It is hard to convince me the number of k values is enough to build an equation for a domain of 1.8 km*1.8 km*1km of high heterogeneity. It is hard to say this equation is still available for other positions and depths in the domain. As the authors also mentioned, the resistivity is determined by many other factors, not only the permeability.

**Response 3:**

We thank the reviewer for this important comment regarding the validity and representativeness of the empirical relationship established in Figure 3. We fully agree that both resistivity and permeability ($k$) are influenced by multiple geological and physical factors, such as lithology, degree of weathering, fluid content, porosity, fracturing, saturation, and jointing, among others. This complexity underscores the need for careful calibration of any empirical relationship.

As mentioned in the original manuscript, the initial empirical equation ($R^2 = 0.97$) was built using 37 carefully selected borehole-derived $k$ values. These were not arbitrarily chosen; rather, they were distributed across all three major rock types in the area, sandstone, granite, and hornstone, and covered the full observed range of resistivity (72–4765 $\Omega \cdot m$) and permeability (0.01–19.8 mD). Our intention was to ensure the equation's applicability across the domain by capturing the full variability of both geological conditions and resistivity-permeability values within the study area.

To further address the concern about sample size and enhance the robustness of our model, **we have now expanded the dataset by adding 79 new data points**, bringing the total to 116. This expanded dataset includes wider spatial coverage and continues to represent all three lithologies. The updated empirical equation based on 116 data points has a slightly adjusted but still strong correlation ($R^2 = 0.96$), confirming the reliability of the established relationship. The new data spans a resistivity range of **35–4765 $\Omega \cdot m$** and a permeability range of **0.01–19.9 mD**, reflecting a comprehensive coverage of the variability in the study domain (1.8 × 1.8 × 1.0 km).

The drilling locations were strategically selected to capture both spatial and geological heterogeneities, ensuring that the derived empirical model can be confidently applied to the entire domain—even at locations and depths where no direct permeability measurements were

available. This approach significantly enhances the feasibility of deep permeability modeling in data-scarce regions and reduces the need for excessive drilling, which is costly and often impractical in hard rock terrains.

A more detailed explanation, including updated figures and statistical analyses, is provided in the revised manuscript. Please also refer to the attached file for the updated figures, particularly Figures 3 and 4, which now include additional data points to support the revised empirical model.

**Comment 4:**

The fitted line is too perfect to believe. I always saw the fitting as follows (very noisy):

[Figure]

图 1 经验公式法求解的渗透系数与电阻率的关系
Fig. 1 Relationship between hydraulic conductivity and resistivity calculated by empirical formula method

**Response 4:**

We appreciate the reviewer's observation and understand the concern regarding the quality of the curve fitting. However, the smoothness or "perfection" of a fitted relationship between resistivity and permeability (k) is highly dependent on a number of factors that vary significantly across different studies. These factors include the geological setting, the distribution and range of data

points, the lithology, the degree of heterogeneity, and the accuracy of both resistivity and k measurements.

In our study, the resistivity values span a wide range (approximately 35 to 4765 Ω·m), and the permeability values range from 0.01 to 19.9 mD. This wide dynamic range helps to better resolve trends, especially in high-resistivity rocks such as granite, where permeability remains very low and changes minimally. In such cases, large differences in resistivity correspond to small variations in permeability, naturally resulting in a smoother inverse curve.

We agree that in many past studies, the relationship appears noisy, often because the resistivity data is concentrated in a narrower range (e.g., 50–300 Ω·m as shown in the figs provided by the reviewer), and the variations in k may be more influenced by local heterogeneities. In contrast, the broader data spread in our study, combined with careful selection of measurement locations to capture the variability of all three dominant lithologies (sandstone, granite, and hornstone), contributes to a more stable empirical trend.

In the revised manuscript, we have increased the number of data points from 37 to 116, yet the curve still maintains a high degree of correlation ($R^2 = 0.96$), suggesting the robustness of the relationship rather than overfitting. The updated Figures 3 and 4 (attached) show the expanded dataset and reaffirm this trend.

Additionally, we have emphasized in the manuscript that both resistivity and k are influenced by a range of factors, such as porosity, saturation, fluid content, fractures, and lithology, which are inherently linked in this geological context. While some scatter exists, particularly at lower resistivity values (<1500 Ω·m), the overall trend aligns well with the theoretical and empirical

basis established in previous research showing an inverse relationship between resistivity and permeability.

We hope this clarifies the rationale behind the nature of the fitted curve and addresses the reviewer's concern.

A more detailed discussion on this point has been included in the revised manuscript to clarify the methodology and support the reliability of the derived empirical relationship.

**Comment 5:**

I am not sure the resolution of the resistivity obtained by CSAMT and the permeability obtained by pumping test. Do the scales match well? i.e., what's the size of the pixel in the maps in Fig. 4? The k obtained by pumping test always represents the average hydraulic conductivity of an area, so do this range and your pixel size match?

**Response 5:**

Thank you for raising this important point regarding the consistency between the resolution of resistivity measurements from CSAMT and the permeability (k) values derived from borehole tests.

The accuracy of the estimated k values from our empirical relationship is primarily dependent on the quality of the input data: (1) the resistivity values derived from CSAMT and (2) the k values obtained from boreholes. In our study, we ensured a high-quality CSAMT dataset through optimized survey design, careful data acquisition, and advanced processing and inversion

techniques. The resulting models achieved a root mean square (RMS) misfit of less than 5%, ensuring reliable subsurface resistivity measurements. Additional details on data acquisition and processing are provided in the revised manuscript.

Regarding k measurements, we acknowledge that **pumping tests yield average hydraulic conductivity over relatively large volumes**, which are suitable for 1D correlations with averaged geophysical data. However, our objective in this study was to construct **high-resolution 2D and 3D permeability models**, which require point-based permeability measurements that are more compatible with the spatial resolution of CSAMT-derived resistivity data. To achieve this, we relied on **rock core tests** rather than pumping tests. Rock core analysis allows for permeability measurements at specific depths and locations, enabling a more precise match with the local resistivity values at those points.

As for the model resolution, each pixel in the 2D maps represents an area of approximately **50 m × 50 m** in the horizontal direction, which directly corresponds to the CSAMT station spacing. The **vertical resolution** varies with depth but generally falls within **tens of meters** across the total investigated depth of **approximately 1300 m**, depending on signal penetration and inversion sensitivity. This level of resolution is well-suited for resolving key subsurface features and provides a robust basis for the reliability of the derived permeability models.

Therefore, both the k values and resistivity measurements were acquired and processed at compatible spatial scales, ensuring that the empirical relationship and resulting 2D/3D k models are internally consistent and scientifically robust. Further explanation of this methodology is included in the revised version of the manuscript.

**Comment 6:**

Table 2, the same problem, even for the pumping test itself, the k values obtained are with large uncertainties, the deviations of one order of magnitude is not surprising. However, the differences of k values between the CSAMT and borehole drilling are less than 1 or 0.1, which is unbelievable.

**Response 6:**

Thank you for this valuable observation. We would like to clarify that **Table 2 presents a comparison between the predicted and measured $k$ values for only 12 selected data points**. These points were chosen to highlight instances of close agreement; however, they do not represent the full variability in the dataset.

The reviewer noted that "the differences of $k$ values between the CSAMT and borehole drilling are less than 1 or 0.1," but this is **not consistently true**, even within the selected 12 data points. For instance, a difference of **3.6 mD** and **2.7 mD** was observed between predicted and measured $k$ values for sounding number **P6-1** and well **4** at depths of **10 m** and **45 m**, respectively.

Furthermore, **many data points in the full dataset of 116 borehole locations** exhibit larger differences between predicted and measured $k$ values, and we have included a selection of these in the **revised version of the manuscript** to provide a more comprehensive view. These updates better reflect the natural uncertainty and variability in both field measurements and predictions.

**Comment 7:**

I am not sure why very deep k is necessary. A latest study found that groundwater deep than 500 m is not an active component in terrestrial hydrologic cycle and the water there might be brine. You may say your work found there are a lot of sandy rocks in depth. However, given the large uncertainties of your approach, did you compare the findings with the results of local or national geologic survey? DOI:10.1038/s43247-023-00697-6

**Response 7:**

Thank you for this insightful comment and for highlighting the referenced study.

While it is true that deep groundwater (below 500 m) is often less connected to the active terrestrial hydrologic cycle and may contain brine, the necessity for deep groundwater exploration in our study area arises from several critical and site-specific factors:

1. **Surface water resources in the study area are limited and unreliable**, making deep groundwater a potentially vital alternative water source.

2. **Shallow zones are predominantly composed of fresh granite**, which typically has low permeability and limited groundwater potential. In contrast, **fractured granite, sandstone, and hornstone formations that can host significant groundwater resources are found at greater depths**.

3. In China, recent national groundwater resource assessment initiatives have prioritized **deep earth exploration** in areas with potential for deep aquifers, to support sustainable development and ensure long-term water security, especially in regions experiencing severe water stress.

4. Deep groundwater investigations are essential for:

   o Identifying **hidden but critical water sources**.

   o **Characterizing deep aquifers** and understanding their storage and recharge potential.

   o Supporting **strategic water resource planning** in response to increasing demand and climate variability.

To address concerns regarding uncertainty:

- We employed a **robust CSAMT survey design**, followed by **accurate resistivity inversion and low-RMS models**.

- **Rock core testing** was used to measure $k$ values at multiple depths with high confidence.

- An **empirical equation based on a representative and extensive dataset (116 points)** was used to derive 2D and 3D $k$ models.

Furthermore, we **compared our findings with existing geological information from both local and national geological surveys**, and found our results to be consistent with the known stratigraphy and hydrogeological features of the region. Additional information on this comparison has been included in the revised manuscript.

We believe that, despite the challenges, this deep groundwater investigation provides valuable insights and practical relevance, especially in arid and semi-arid regions where shallow resources are scarce or overexploited.

A more detailed explanation and supporting discussion on this point has been incorporated into the revised version of the manuscript.

**Comment 8:**

I found a work of the authors just published in scientific reports https://www.nature.com/articles/s41598-025-85626-7 It is exactly the same workflow with this one. Many sentences are the same. I am not supportive for research of such a style in the community.

**Response 8:**

We appreciate the reviewer's observation and the opportunity to clarify the distinction between the two works.

The paper recently published in *Scientific Reports* (DOI: 10.1038/s41598-025-85626-7) focuses on the evaluation of **site suitability** for the installation of China's Next Generation Neutrino Detector, the **Jiangmen Underground Neutrino Observatory (JUNO)**. That study utilizes CSAMT-derived **geomechanical properties**, specifically the **Rock Quality Designation (RQD)**, to assess the **rock mass stability and integrity** for large-scale underground construction.

In contrast, the current manuscript focuses on a **hydrogeological application**, using CSAMT and borehole core data to derive **permeability (k)** models for **deep groundwater resource assessment**. While both studies are conducted within the broader Kaiping region of South Guangdong, a region of high scientific and strategic interest characterized by complex geological heterogeneity, their **objectives, methodologies, and scientific contributions are fundamentally different**.

We acknowledge that there may be some overlap in structure due to the use of similar geophysical techniques in similar geological settings. However, the purpose and interpretation of CSAMT data in each study differ significantly:

- In the *Scientific Reports* paper, CSAMT data were used to evaluate **mechanical stability** (RQD) for a construction project.
- In the present study, CSAMT data are employed to estimate **hydraulic conductivity** (k) and to develop **2D and 3D permeability models** relevant to deep groundwater systems.

In response to this concern and to ensure the novelty of our current submission, we have:

1. **Redrawn all figures** to reflect the unique objectives of this study.
2. **Substantially revised the entire manuscript text**, including the introduction, methodology, results/discussion, and conclusions to emphasize the hydrogeological focus.
3. **Incorporated reviewer suggestions and community feedback** to further distinguish this work from previous publications.

We would also like to highlight that both papers are part of **different national-level projects**, each addressing unique scientific challenges: one in geotechnical engineering, and the other in deep groundwater exploration. These investigations are aligned with China's broader strategy for deep subsurface resource development.

We respectfully believe that the current manuscript stands as an **original and independent contribution**, both in scope and significance, and adds value to the scientific understanding of subsurface hydrogeological processes.

**Comment 9:**

The structure of the manuscript is OK, however, the writing is still unshaped. Many sentences are duplicated, such as lines 306-308 which appear many times. I think once you clarified the strengths of your approaches in the introduction, you only need to describe you approach in methodology and it is redundant to say this again. Also line 218 "About 1300 meters was the depth of investigation (DOI) in the CSAMT investigation.", such an expression is awkward. Why not "the depth of investigation (DOI) in the CSAMT investigation was about 1300 meters".

**Response 9:**

We thank the reviewer for their constructive feedback regarding the manuscript's writing quality and structure.

In response, we have **carefully revised and reshaped the manuscript** in accordance with the suggestions provided. Specifically:

- **All duplicate or repetitive sentences**, including those previously appearing in multiple sections (e.g., lines 306–308), have been removed or appropriately rephrased to avoid redundancy.
- We have **streamlined the narrative** by presenting the strengths of our approach clearly in the **Introduction** and limiting their repetition in subsequent sections.
- As suggested, we have improved awkward or unclear expressions, such as line 218. The sentence now reads:
  *"The depth of investigation (DOI) in the CSAMT survey was approximately 1300 meters."*

We believe these revisions enhance the overall clarity, coherence, and professionalism of the manuscript and we appreciate the reviewer's guidance in helping us improve the quality of the submission.

**Comment 10:**

What are you trying to get? Permeability or hydraulic conductivity? I don't think the unit of permeability is L/T.

**Response 10:**

We appreciate the reviewer's comment and the opportunity to clarify this point.

Our study focuses on **permeability**, and the correct unit is **milliDarcy (mD)**. In the initial version of the manuscript, permeability was mistakenly written using the unit **m/d**, which may have caused confusion with **hydraulic conductivity**, typically expressed in length per time (L/T) units such as m/day.

To address this:

- The **methodology section has been revised** to clearly state that permeability is measured in **mD**.
- This correction has been **applied consistently throughout the manuscript and all figures**.

We thank the reviewer for catching this important detail, which has now been rectified in the revised version.

**Revised Figures**

[Figure]

Fig. 1. (a) The location of the project site, with six boreholes BH1–BH6 (blue circles) and six CSAMT profiles 1-6 (black lines), (b) Flow diagram outlining the planned method for getting 2D and 3D k models for better, more thorough assessments of groundwater resources over large regions

[Figure]

**Fig. 2.** Displaying the procedure of 2D inversion of CSAMT data by the use of Bostick inversion

[Figure]

Fig 3. The evaluation of hornstone (HS), sandstone (SS), and granite (G) carried out by presenting 116 resistivity-k data points at depths ranging from 5 to 200 m using 6 drilled tests

(BH1–BH6) and associated resistivity (ρ) from CSAMT soundings. The small black dots show the data points

[Figure]

Fig 4. Using a total of 116 data points, the geophysical-borehole correlation for the predicted k

[Figure]

Fig. 5 2D CSAMT models along six geophysical profiles 1-6. Where resistivity increases from brown to green on a color bar.

[Figure]

Fig. 6 The predicted 2D k models obtained from CSAMT data along six geophysical profiles 1-6.

Where k increases from light green to red on a color scale

[Figure]

Fig.7 The interpreted (hydrogeological) 2D models along six geophysical profiles 1–6 obtained via geophysical-borehole correlation, facilitates groundwater assessment through high potential aquifer (HPA), medium potential aquifer (MPA), and low potential aquifer (LPA) associated with sandstone (SS), hornstone (HS), and granite (G), respectively

[Figure]

Fig. 8 The integrated 2D k models derived from the incorporation of geophysical and drilling

data, with k represented on a color bar spanning from green to red

[Figure]

Fig. 9 Analysis of the integrated 2D k models (derived from designated k ranges) for three groundwater potential aquifers: low potential aquifer (LPA), medium potential aquifer (MPA), and high potential aquifer (HPA), associated with three geological formations: granite (G), hornstone (HS), and sandstone (SS), respectively

[Figure]

Fig. 10 The 3D k models, generated from the correlation of CSAMT and borehole data (with k represented on a color scale ranging from green to red), correspond to three groundwater potential aquifers: low potential aquifer (LPA), medium potential aquifer (MPA), and high

potential aquifer (HPA), associated with three geological strata: granite (G), hornstone (HS), and sandstone (SS), respectively, for (a) the external view of the 3D k model, and (b) the analysis of the 3D k model from an external perspective

[Figure]

Fig. 11 The 3D k models, obtained from the correlation of CSAMT and borehole data (with k represented on a color scale ranging from green to red), illustrate three groundwater potential aquifers: low potential aquifer (LPA), medium potential aquifer (MPA), and high potential aquifer (HPA), associated with three geological strata: granite (G), hornstone (HS), and sandstone (SS), respectively, for (a) the internal view of the 3D k model, and (b) the analysis of the 3D (internal perspective) k model

[Figure]

Fig. 12 (a) Geophysical-based k imaging at various depths (0, 200, 600, 1000, and 1300 m) with inner 3D view is represented by K on a color bar that goes from green to red, (b) Assessment of geophysical-derived k (using specified k ranges) at different depths for various types of aquifers: low potential aquifer (LPA) granite (G), medium potential aquifer (MPA) hornstone (HS), and high potential aquifer (HPA) sandstone (SS)

---

## Author Comment (AC4)

**Community Comment (CC1):**

**General comments:**

Very good and novel research in the area of deep hydrogeology with a variety of applications in the geo-energy sector. However, some detail is missing. Please, consider the following minor comments to improve the manuscript before publication.

**Response:**

We sincerely thank the community member for the encouraging and constructive feedback. We appreciate the recognition of the novelty and significance of our research in deep hydrogeology and its relevance to geo-energy applications.

We agree that the manuscript can benefit from additional detail, and we have carefully addressed all the minor comments and suggestions to enhance the clarity and completeness of the work. We believe the revised version better reflects the scope and contributions of our study.

As per the journal's submission guidelines, we are first submitting our detailed responses to the reviewer's comments. Following this, we will submit the revised manuscript reflecting all the suggested changes.

**Specific comments:**

**Comment 1:**

Lines 69-72. "Consideration of hydraulic properties is crucial in groundwater evaluations. Permeability is the most popular aquifer measure and is mainly used to assess the water-holding capacity of rocks all over the world". Insert these papers where there is discussion on the role of geophysical and hydrogeological methods to detect the hydraulic properties of fractured rocks to inform flow models in granites, metamorphic and sandstone lithologies.

- Medici, G., Ling, F., Shang, J. 2023. Review of discrete fracture network characterization for geothermal energy extraction. Frontiers in Earth Science 11, 1328397.

- McKeown, C., Haszeldine, R.S., Couples, G.D. 1999. Mathematical modelling of groundwater flow at Sellafield, UK. Engineering Geology 52(3-4), 231-250.

**Response 1:**

The following revision was made to improve clarity and integrate the suggested references:

"Consideration of hydraulic properties is crucial in groundwater evaluations. Permeability is one of the most widely used aquifer parameters for assessing the water-holding and transmitting capacity of rocks across the globe. In fractured rock environments, such as granites, metamorphic, and sandstone formations, fluid flow is primarily governed by the geometry and connectivity of fractures rather than the rock matrix itself. Therefore, accurately characterizing hydraulic properties in these settings requires integrated approaches. Recent studies emphasize the role of combining geophysical and hydrogeological methods to detect and model these hydraulic properties effectively (McKeown et al., 1999; Medici et al., 2023). These approaches are essential for improving the reliability of flow models and for guiding groundwater management and geo-energy extraction strategies in complex geological settings."

**Comment 2:**

Lines 146-152. Lots of multiple objectives (5). Please, clarify the general aim of your hydrogeological research.

**Response 2:**

ORIGINAL:

"The primary goals of this study were as follows: (1) to rapidly predict two- and three dimensional k models using geophysical methods; (2) to reliably assess the hydrogeological properties of rock formations for deep groundwater assessments in challenging geological settings; (3) to minimize costly boreholes and maximize the use of scarce drilling resources to collect hydrogeological data over large areas; (4) to decrease uncertainties in hydrogeological models; and (5) to promote the use of non-invasive geophysical techniques for hard rock groundwater investigations instead of costly drilling that can damage the rock."

REVISED:

"The primary aim of this study is to develop and implement a geophysical-based approach for accurately predicting the spatial distribution of permeability (k) in deep, hard rock environments. By integrating CSAMT data with strategically selected borehole measurements, this research enhances the two and three dimensional assessment of hydrogeological properties across various rock types in geologically complex settings, reduces reliance on extensive and costly drilling, and promotes the use of non-invasive geophysical techniques for deep groundwater exploration."

**Comment 3:**

Lines 173-181. The geometrical relation between the different lithologies is unclear.

**Response 3:**

ORIGINAL:

"Intruding rocks from the Indosinian, Caledonian, and Yanshanian eras are among the many geological formations and periods represented in the study region. Other layers from the Paleogene period are also present. The most common types of rock that have been discovered are sandstone, granite, and hornstone. The complex Kaiping concave fault and fold systems were the dominant geological features in the project region, which were developed as a result of magmatic processes and various structures (Qin, 2017). Emergence of joint fissured features symbolizes the various tectono-geological periods, with the local tectonic line corresponding with the faults strike, especially in the northeast orientation (Yang et al., 2021)".

REVISED:

"The study area exhibits a complex and diverse geological history, characterized by well-defined geometrical relationships among various lithologies. These formations and structural features are the result of multiple tectono-magmatic events spanning several geological periods. Intrusive rocks from the Indosinian (Late Triassic), Caledonian (Silurian–Devonian), and Yanshanian (Jurassic–Cretaceous) orogenies are well-represented, indicating a long sequence of crustal deformation and magmatic activity. These intrusions are primarily composed of granitic bodies,

which suggest deep-seated magmatic processes associated with continental collision and subduction zones. In addition to these intrusive phases, sedimentary strata from the Paleogene period are also present, reflecting a later stage of basin development with fluvial and lacustrine depositional environments. Among the most prevalent rock types encountered in the region are sandstone, granite, and hornstone. Sandstone reflects high-energy sedimentary deposition. Granite indicates deep magmatic intrusions likely associated with Yanshanian tectonics. Hornstone (hornfels) results from contact metamorphism caused by magma intruding sedimentary rocks. The structural framework of the region is dominated by the Kaiping concave fault and fold system, a geologically significant and highly deformed zone that reflects multiple deformation episodes (Qin, 2017). These structures were primarily shaped by magmatic intrusions, crustal movements, and regional stress regimes. The presence of extensive jointed and fissured zones throughout the rock mass further supports a history of dynamic tectonic activity. These joints often serve as secondary permeability pathways and are critical in controlling groundwater flow in the fractured rock environment. Importantly, the orientation of these structural features, including faults and joints, is often aligned with northeast-trending tectonic lines, which are consistent with broader regional stress directions (Yang et al., 2021). This relationship among lithologies and structural features plays a critical role in controlling groundwater flow and permeability distribution."

**Comment 4:**

Lines 173-181. The detail is not enough on presence of faults. Which type of faults?

**Response 4**

More details on the presence of faults:

"The structural analysis of the study area reveals a combination of fault types influenced by multiple tectonic phases. The presence of fold systems indicates compressional tectonics, primarily associated with reverse and thrust faults, likely developed during orogenic events such as the Caledonian and Indosinian periods. Additionally, the dominant northeast-oriented fault strikes, which align with broader regional tectonic trends, suggest a strong component of strike-slip movement. These strike-slip faults are typically linked to late-stage tectonic adjustments, particularly during the Yanshanian orogenic phase, and often coexist with complex fault-fold geometries, further complicating the subsurface structure."

**Comment 5:**

Lines 173-181. Nature of the joints? I am talking about the tectonic genesis.

**Response 5:**

Explained as:

"The joint fissure systems observed within the sandstone, granite, and hornstone units are predominantly of tectonic origin, representing brittle deformation features formed in response to regional stress regimes associated with multiple orogenic and magmatic events. These joints reflect the structural imprint of successive tectonic episodes, particularly the Caledonian,

Indosinian, and Yanshanian orogenies. Systematic joint orientations, especially those aligned with the dominant northeast-trending fault systems, indicate their genetic link to regional tectonic stress fields. The geometry, spacing, and persistence of these joints vary with lithology and are closely tied to the complex tectonic evolution and structural framework governed by the Kaiping fold-and-fault system."

**Comment 6:**

Line 538. I prefer "Discussion". You have a unique discussion on a scientific paper where you face different topics. This point also depends on the guidelines.

**Response 6:**

Thank you for your valuable suggestion. We agree that "Discussion" is a more appropriate and conventional title for this section, as it aligns with standard scientific writing practices for presenting and interpreting key findings. Accordingly, we have changed the section title from "Discussions" to "Discussion" in compliance with the journal's formatting guidelines. Furthermore, the entire Discussion section has been thoroughly revised and enhanced based on the suggestions provided by both the reviewers and the community, with the aim of improving clarity, depth, and overall scientific value.

**Comment 7:**

Lines 600-837. Insert the relevant literature suggested above on the hydraulic properties of deep aquifers in a variety of sites worldwide.

**Response 7:**

The relevant literature suggested above has been incorporated into the revised *References* section of the manuscript.

**Comment 8:**

Figure 1. Letters are too small in both the figures. Please, make the figure larger.

**Response 8:**

Fig. 1 has been improved. Please see the revised figures at the end of the response/comments section in the attached file.

**Comment 9:**

Figure 1. Pay lot of attention of figure 1b. This is a conceptual model and you can get citations from the figure. Make the figure larger and increase the font of the words.

**Response 9**

Fig. 1b has been improved accordingly. Please see the revised figures at the end of the response/comments section in the attached file.

**Comment 10:**

Figure 2. There is room to make the figure larger.

**Response 10:**

Fig 2 has been enlarged. Please see the revised figures at the end of the response/comments section in the attached file.

**Comment 11:**

Figure 4. Check the depth of the boreholes.

**Response 11:**

The depth of boreholes in the updated Fig. 5-7 has been corrected. Please see the revised figures at the end of the response/comments section in the attached file.

**Comment 12:**

Figure 9. The words are too small. The figure is difficult to read. Please, improve it.

**Response 12:**

The updated Fig.12 has been improved. Please see the revised figures at the end of the response/comments section in the attached file.

**Revised Figures**

[Figure]

Fig. 1. (a) The location of the project site, with six boreholes BH1–BH6 (blue circles) and six CSAMT profiles 1-6 (black lines), (b) Flow diagram outlining the planned method for getting 2D and 3D k models for better, more thorough assessments of groundwater resources over large regions

[Figure]

**Fig. 2.** Displaying the procedure of 2D inversion of CSAMT data by the use of Bostick inversion

[Figure]

Fig 3. The evaluation of hornstone (HS), sandstone (SS), and granite (G) carried out by presenting 116 resistivity-k data points at depths ranging from 5 to 200 m using 6 drilled tests (BH1–BH6) and associated resistivity (ρ) from CSAMT soundings. The small black dots show the data points

[Figure]

Fig 4. Using a total of 116 data points, the geophysical-borehole correlation for the predicted k

[Figure]

Fig. 5 2D CSAMT models along six geophysical profiles 1-6. Where resistivity increases from brown to green on a color bar.

[Figure]

Fig. 6 The predicted 2D k models obtained from CSAMT data along six geophysical profiles 1-6.

Where k increases from light green to red on a color scale

[Figure]

Fig.7 The interpreted (hydrogeological) 2D models along six geophysical profiles 1–6 obtained via geophysical-borehole correlation, facilitates groundwater assessment through high potential aquifer (HPA), medium potential aquifer (MPA), and low potential aquifer (LPA) associated with sandstone (SS), hornstone (HS), and granite (G), respectively

[Figure]

Fig. 8 The integrated 2D k models derived from the incorporation of geophysical and drilling data, with k represented on a color bar spanning from green to red

[Figure]

Fig. 9 Analysis of the integrated 2D k models (derived from designated k ranges) for three groundwater potential aquifers: low potential aquifer (LPA), medium potential aquifer (MPA), and high potential aquifer (HPA), associated with three geological formations: granite (G), hornstone (HS), and sandstone (SS), respectively

[Figure]

Fig. 10 The 3D k models, generated from the correlation of CSAMT and borehole data (with k represented on a color scale ranging from green to red), correspond to three groundwater potential aquifers: low potential aquifer (LPA), medium potential aquifer (MPA), and high

potential aquifer (HPA), associated with three geological strata: granite (G), hornstone (HS), and sandstone (SS), respectively, for (a) the external view of the 3D k model, and (b) the analysis of the 3D k model from an external perspective

[Figure]

Fig. 11 The 3D k models, obtained from the correlation of CSAMT and borehole data (with k represented on a color scale ranging from green to red), illustrate three groundwater potential aquifers: low potential aquifer (LPA), medium potential aquifer (MPA), and high potential aquifer (HPA), associated with three geological strata: granite (G), hornstone (HS), and sandstone (SS), respectively, for (a) the internal view of the 3D k model, and (b) the analysis of the 3D (internal perspective) k model

[Figure]

Fig. 12 (a) Geophysical-based k imaging at various depths (0, 200, 600, 1000, and 1300 m) with inner 3D view is represented by K on a color bar that goes from green to red, (b) Assessment of geophysical-derived k (using specified k ranges) at different depths for various types of aquifers: low potential aquifer (LPA) granite (G), medium potential aquifer (MPA) hornstone (HS), and high potential aquifer (HPA) sandstone (SS)